# Prediction-Powered Adaptive Shrinkage Estimation

**Sida Li** [1]    **Nikolaos Ignatiadis** [1] [2]

## Abstract

Prediction-Powered Inference (PPI) is a powerful framework for enhancing statistical estimates by combining limited gold-standard data with machine learning (ML) predictions. While prior work has demonstrated PPI's benefits for individual statistical problems, modern applications require answering numerous parallel statistical questions. We introduce Prediction-Powered Adaptive Shrinkage (`PAS`), a method that bridges PPI with empirical Bayes shrinkage to improve estimation of multiple means. `PAS` debiases noisy ML predictions *within* each problem and then borrows strength *across* problems by using those same predictions as a reference point for shrinkage. The amount of shrinkage is determined by minimizing an unbiased estimate of risk, and we prove that this tuning strategy is asymptotically optimal. Experiments on both synthetic and real-world datasets show that `PAS` adapts to the reliability of the ML predictions and outperforms traditional and modern baselines in large-scale applications.

## 1. Introduction

A major obstacle in answering modern scientific questions is the scarcity of gold-standard data (Miao et al., 2024b). While advancements in data collection, such as large-scale astronomical surveys (York et al., 2000) and web crawling (Penedo et al., 2024), have led to an abundance of covariates (or features), scientific conclusions often rely on outcomes (or labels), which are often expensive and labor-intensive to obtain. The rapid development of machine learning (ML) algorithms has offered a path forward, with ML predictions increasingly used to supplement gold-standard outcomes and increase the statistical efficiency of subsequent analyses (Liang et al., 2007; Wang et al., 2020).

Prediction-Powered Inference (PPI) (Angelopoulos et al., 2023) addresses the scarcity issue by providing a framework for valid statistical analysis using predictions from black-box ML models. By combining ML-predicted and gold-standard outcomes, PPI and its variants (Angelopoulos et al., 2024; Zrnic & Candès, 2024; Zrnic, 2025) use the abundance of predictions to reduce variance while relying on the accuracy of labeled[1] data to control bias.

In this work, we adapt PPI to the estimation of multiple outcome means in compound estimation settings. Many applications of PPI naturally involve parallel statistical problems that can be solved simultaneously. For instance, several PPI methods (Angelopoulos et al., 2024; Fisch et al., 2024) have shown improvements in estimating the fraction of spiral galaxies using predictions on images from the Galaxy Zoo 2 dataset (Willett et al., 2013). While these methods focus on estimating a single overall fraction, a richer analysis emerges from partitioning galaxies based on metadata (such as celestial coordinates or pre-defined bins) and estimating the fraction of galaxies within each partition. This compound estimation approach enables more granular scientific inquiries that account for heterogeneity across galaxy clusters and spatial locations (Nair & Abraham, 2010).

We demonstrate, both theoretically and empirically, the benefits of solving multiple mean estimation problems simultaneously. Our approach builds on the empirical Bayes (EB) principle of sharing information *across* problems (Robbins, 1956; Efron, 2010) as exemplified by James-Stein shrinkage (James & Stein, 1961; Xie et al., 2012). The connection between modern and classical statistical ideas allows us to perform *within* problem PPI estimation in the first place, followed by a shrinkage step reusing the ML predictions in an adaptive way, which becomes possible through borrowing information *across* problems. Our contributions are:

1. We propose Prediction-Powered Adaptive Shrinkage (`PAS`) for compound mean estimation. `PAS` inherits the flexibility of PPI in working with *any* black-box predictive model and makes *minimal* distributional assumptions about the data. Its two-stage estimation process makes efficient use of the ML predictions as both a variance-reduction device and a shrinkage target.

---

[1]Data Science Institute, The University of Chicago [2]Department of Statistics, The University of Chicago. Correspondence to: Sida Li <listar2000@uchicago.edu>.

*Proceedings of the $42^{nd}$ International Conference on Machine Learning*, Vancouver, Canada. PMLR 267, 2025. Copyright 2025 by the author(s).

---

[1]Throughout the paper, we use the terms "labeled" and "gold-standard" interchangeably.

2. We develop a Correlation-Aware Unbiased Risk Estimate (CURE) for tuning the PAS estimator, establish asymptotic optimality of this tuning strategy, and derive an interpretation in terms of a Bayes oracle risk upper bound.

3. We conduct extensive experiments on both synthetic and real-world datasets. Our experiments demonstrate PAS's applicability to large-scale problems with deep learning models, showing improved estimation accuracy compared to other classical and modern baselines.

## 2. Preliminaries and Notation

### 2.1. Prediction-Powered Inference (PPI)

The PPI framework considers a setting where we have access to a small number of labeled data points $(X_i, Y_i)_{i=1}^n \in (\mathcal{X} \times \mathcal{Y})^n$ and a large number of unlabeled covariates $(\tilde{X}_i)_{i=1}^N \in (\mathcal{X})^N$, where $\mathcal{X}$ and $\mathcal{Y}$ represent the covariate and outcome space, respectively. The data points are drawn iid from a joint distribution $\mathbb{P}_{XY}$.[2] We are also given a black-box predictive model $f : \mathcal{X} \to \mathcal{Y}$ that is independent of the datasets (e.g., pre-trained on similar but unseen data). For mean estimation with $\mathcal{Y} \subset \mathbb{R}$, the goal is to leverage the predicted outcomes $f(X_i)$ to improve the estimation of $\theta := \mathbb{E}[Y_i]$. Some simple estimators take the form of the following aggregated (summary) statistics

$$\bar{Y} := \frac{1}{n} \sum_{i=1}^n Y_i, \qquad \tilde{Y} := \frac{1}{N} \sum_{i=1}^N \tilde{Y}_i,$$
$$\bar{Z}^f := \frac{1}{n} \sum_{i=1}^n f(X_i), \quad \tilde{Z}^f := \frac{1}{N} \sum_{i=1}^N f(\tilde{X}_i). \quad (1)$$

Above, $\bar{Y}$ is the classical estimator,[3] $\bar{Z}^f$, $\tilde{Z}^f$ are the prediction means on the labeled and unlabeled data, and $\tilde{Y}$ (grayed out) is unobserved. The vanilla PPI estimator is defined as,

$$\hat{\theta}^{\mathrm{PPI}} := \underbrace{\bar{Y}}_{\text{Baseline}} + \underbrace{(\tilde{Z}^f - \bar{Z}^f)}_{\text{Variance Reduction}} = \underbrace{\tilde{Z}^f}_{\text{Baseline}} + \underbrace{(\bar{Y} - \bar{Z}^f)}_{\text{Debiasing}}. \quad (2)$$

Both definitions represent $\hat{\theta}^{\mathrm{PPI}}$ in the form of a baseline estimator plus a correction term. In the first representation, the baseline estimator is the unbiased classical estimator $\bar{Y}$, while the correction term has expectation 0 and attempts to reduce the variance of $\bar{Y}$. In the second representation, the baseline estimator is the prediction mean on unlabeled data $\tilde{Z}^f$ (which in general may be biased for $\theta$), while the correction term removes the bias of $\tilde{Z}^f$ by estimating the bias of the ML model $f$ on the labeled dataset.

---

[2]To be concrete: $(X_i, Y_i) \overset{\text{iid}}{\sim} \mathbb{P}_{XY}$ and $(\tilde{X}_i, \tilde{Y}_i) \overset{\text{iid}}{\sim} \mathbb{P}_{XY}$ independently, but $\tilde{Y}_i$ is unobserved.

[3]From now on, we will use the term "classical estimator" to refer to the sample average of the labeled outcomes.

Writing $\hat{\theta}^{\mathrm{PPI}} = \frac{1}{N} \sum_{i=1}^N f(\tilde{X}_i) + \frac{1}{n} \sum_{i=1}^n (Y_i - f(X_i))$, we find that $\mathbb{E}[\hat{\theta}^{\mathrm{PPI}}] = \mathbb{E}[Y_i] = \theta$ and

$$\mathrm{Var}[\hat{\theta}^{\mathrm{PPI}}] = \frac{1}{N} \mathrm{Var}[f(\tilde{X}_i)] + \frac{1}{n} \mathrm{Var}[Y_i - f(X_i)], \quad (3)$$

that is, $\hat{\theta}^{\mathrm{PPI}}$ is unbiased for $\theta$ and its variance becomes smaller when the model predicts the true outcomes well. The mean squared error (MSE) of $\hat{\theta}^{\mathrm{PPI}}$ is equal to $\mathrm{Var}[\hat{\theta}^{\mathrm{PPI}}]$. Although we motivated $\hat{\theta}^{\mathrm{PPI}}$ in (2) as implementing a correction step on two possible baseline estimators ($\bar{Y}$ and $\tilde{Z}^f$), $\hat{\theta}^{\mathrm{PPI}}$ may have MSE for estimating $\theta$ that is arbitrarily worse than either of these baselines.

**Comparison to classical estimator $\bar{Y}$.** The classical estimator $\bar{Y}$ which only uses labeled data is unbiased for $\theta$ and has variance (and MSE) equal to $\frac{1}{n} \mathrm{Var}[Y_i]$.

**Power-Tuned PPI (PPI++).** To overcome the above limitation, Angelopoulos et al. (2024) introduce a power-tuning parameter $\lambda$ and define

$$\hat{\theta}_\lambda^{\mathrm{PPI}} := \bar{Y} + \lambda \left( \tilde{Z}^f - \bar{Z}^f \right), \quad (4)$$

which recovers the classical estimator when $\lambda = 0$ and the vanilla PPI estimator when $\lambda = 1$. For all values of $\lambda$, $\hat{\theta}_\lambda^{\mathrm{PPI}}$ is unbiased, so if we select the $\lambda$ that minimizes $\mathrm{Var}[\hat{\theta}_\lambda^{\mathrm{PPI}}]$, we can improve our estimator over both the classical estimator and vanilla PPI. Such an estimator is defined as the Power-Tuned (PT) PPI[4] estimator $\hat{\theta}^{\mathrm{PT}} := \hat{\theta}_{\lambda^*}^{\mathrm{PPI}}$, where we pick $\lambda^*$ that minimizes the variance (and thus the MSE) of $\hat{\theta}_\lambda^{\mathrm{PPI}}$. We will revisit PT as one of the building blocks of our proposed PAS estimator in Section 4.

**Comparison to $\tilde{Z}^f$.** Consider the ideal scenario for PPI with $N = \infty$ (that is, the unlabeled dataset is much larger than the labeled dataset) so that $\tilde{Z}^f \equiv \mathbb{E}[f(\tilde{X}_i)]$. Even then, the MSE of $\hat{\theta}^{\mathrm{PPI}}$ in (3) is always lower bounded[5] by $\frac{1}{n} \mathbb{E}[\mathrm{Var}[Y_i \mid X_i]]$ and the lower bound is attained by the perfect ML predictor $f(\cdot) \equiv \mathbb{E}[Y_i \mid X_i = \cdot]$. In words, if $Y_i$ is not perfectly predictable from $X_i$, then PPI applied to a labeled dataset of fixed size $n$ must have non-negligible MSE. By contrast, for $N = \infty$, the prediction mean of unlabeled data $\tilde{Z}^f$ has zero variance and MSE equal to the squared bias $(\mathbb{E}[f(X_i)] - \theta_i)^2$. Thus if the predictor satisfies a calibration-type property that $\mathbb{E}[f(X_i)] \approx \mathbb{E}[Y_i]$ (which is implied by, but much weaker than the requirement $f(X_i) \approx Y_i$), then the MSE of $\tilde{Z}^f$ could be nearly 0. By contrast, PPI (and PPI++) can only partially capitalize on such a predictor $f(\cdot)$.

While PPI and PPI++ are constrained by their reliance on unbiased estimators, we show that the compound estima-

---

[4]We use the term "PPI++" for the broader framework, while "PT" refers to the specific estimator.

[5]The same lower bound also applies to power-tuned PPI $\hat{\theta}^{\mathrm{PT}}$.

tion setting (Section 2.2) enables a different approach. By carefully navigating the bias-variance tradeoff through information sharing *across* parallel estimation problems, we can provably match the performance of both $\bar{Y}$ and $\tilde{Z}^f$.

## 2.2. The Compound Mean Estimation Setting

In this section, we introduce the problem setting that PAS is designed to address—estimating the mean of $m > 1$ parallel problems with a single black-box predictive model $f$.[6] For the $j$-th problem, where $j \in [m] := \{1, \ldots, m\}$, we observe a labeled dataset $(X_{ij}, Y_{ij})_{i=1}^{n_j}$ with $n_j \in \mathbb{N}$ samples and an unlabeled dataset $(\tilde{X}_{ij})_{i=1}^{N_j}$ with $N_j \in \mathbb{N}$ samples. We start with modeling heterogeneity across problems.

**Assumption 2.1** (Prior). There exist problem-specific unobserved latent variables $\eta_j$ with

$$\eta_j \overset{\text{iid}}{\sim} \mathbb{P}_\eta, \ j \in [m], \ \text{and} \ \boldsymbol{\eta} := (\eta_1, ..., \eta_m)^\mathsf{T}, \quad (5)$$

where $\mathbb{P}_\eta$ is an unknown probability measure. The latent variable $\eta_j$ fully specifies the distribution of the $j$-th labeled and unlabeled dataset. We use the notation $\mathbb{E}_{\eta_j}[\cdot]$ (resp. $\mathbb{E}_{\boldsymbol{\eta}}[\cdot]$) to denote the expectation conditional on $\eta_j$ (resp. $\boldsymbol{\eta}$), while $\mathbb{E}_{\mathbb{P}_\eta}[\cdot]$ denotes an expectation also integrating out $\mathbb{P}_\eta$.

We do not place any restriction over the unknown prior $\mathbb{P}_\eta$. Assumption 2.1 posits exchangeability across problems, which enables information sharing, without restricting heterogeneity (Ignatiadis et al., 2023). In our setting, we are specifically interested in the means

$$\theta_j := \mathbb{E}_{\eta_j}[Y_{ij}], \ j \in [m], \ \text{and} \ \boldsymbol{\theta} := (\theta_1, \ldots, \theta_m)^\mathsf{T}. \quad (6)$$

Our next assumption specifies that we only model the first two moments of the joint distribution between the outcomes and the predictions. The upshots of such modeling are that the exact form of the observation distribution is neither assumed nor required in our arguments, and that our approach will be directly applicable to settings where the covariate space $\mathcal{X}$ is high-dimensional or structured.

**Assumption 2.2** (Sampling). For each problem $j \in [m]$, we assume that the joint distribution of $(f(X_{ij}), Y_{ij})$ has finite second moments conditional on $\eta_j$ and for $i \in [n_j]$

$$\begin{bmatrix} f(X_{ij}) \\ Y_{ij} \end{bmatrix} \Big| \eta_j \overset{\text{iid}}{\sim} \mathbb{F}_j \left( \begin{bmatrix} \mu_j \\ \theta_j \end{bmatrix}, \begin{bmatrix} \tau_j^2 & \rho_j \tau_j \sigma_j \\ \rho_j \tau_j \sigma_j & \sigma_j^2 \end{bmatrix} \right), \quad (7)$$

where $\mathbb{F}_j, \mu_j, \theta_j, \rho_j, \sigma_j^2, \tau_j^2$ are functions of $\eta_j$. Conditional on $\eta_j$, the unlabeled predictions $f(\tilde{X}_{i'j})$, $i' \in [N_j]$, are also

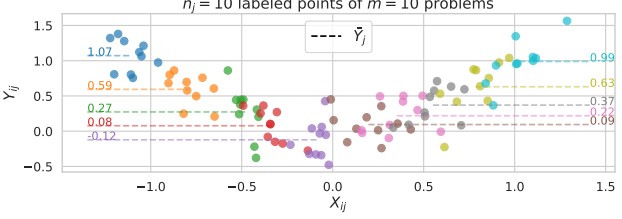

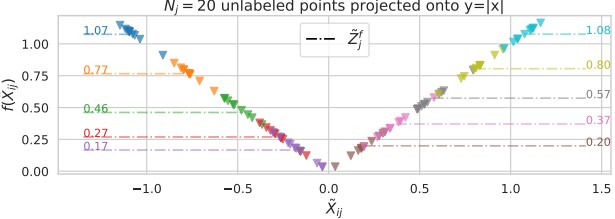

*Figure 1.* We instantiate the model described in Example 2.3 with $m = 10$ problems, each has $n_j = 10$ labeled and $N_j = 20$ unlabeled data (we use different colors for all 10 problems). **(Top)** Labeled data $(X_{ij}, Y_{ij})_{j=1}^{n_j}$ with the classical estimator $\bar{Y}_j$ shown for each problem. **(Bottom)** We apply a flawed predictor $f(x) = |x|$ to the unlabeled covariates and visualize $(X_{ij}, f(X_{ij}))_{j=1}^{N_j}$ as well as the prediction mean $\tilde{Z}_j^f$.

iid, independent of the labeled dataset and identically distributed with $f(X_{ij})$. In the notation of (7), $\mathbb{F}_j$ represents an unspecified distribution satisfying the moment constraints in (6) and

$$\mathbb{E}_{\eta_j}[f(X_{ij})] = \mu_j, \qquad \text{Var}_{\eta_j}[f(X_{ij})] = \tau_j^2,$$
$$\text{Corr}_{\eta_j}[f(X_{ij}), Y_{ij}] = \rho_j, \quad \text{Var}_{\eta_j}[Y_{ij}] = \sigma_j^2.$$

We further denote $\gamma_j := \text{Cov}_{\eta_j}[f(X_{ij}), Y_{ij}] = \rho_j \tau_j \sigma_j$.

Similar to Eq. (1), we define the aggregated statistics $\bar{Y}_j, \bar{Z}_j^f, \tilde{Z}_j^f$ for each $j \in [m]$. To facilitate exposition, following prior work,[7] we treat the second moments $\tau_j^2, \sigma_j^2, \gamma_j$ as known until the end of Section 5. In Section 6, we extend our methods to allow for unknown $\tau_j^2, \sigma_j^2$, and $\gamma_j$.

We next introduce a synthetic model that will serve both as a running example and as part of our numerical study.

**Example 2.3** (Synthetic model). For each problem $j$, let $\eta_j \sim \mathcal{U}[-1, 1]$. We think of $\eta_j$ as both indexing the problems and generating heterogeneity across problems. The $j$-th dataset is generated via (with constants set to $c = 0.05, \psi = 0.1$),

$$X_{ij} \overset{\text{iid}}{\sim} \mathcal{N}(\eta_j, \psi^2), \quad Y_{ij} | X_{ij} \overset{\text{ind}}{\sim} \mathcal{N}(2\eta_j X_{ij} - \eta_j^2, c). \quad (8)$$

In Figure 1, we visualize realizations from this model with $m = 10$ problems, $n_j = 10$ labeled observations, and $N_j = 20$ unlabeled observations for each problem. We apply a flawed predictor $f(x) = |x|$. The classical estimator $\bar{Y}_j$ and the prediction mean $\tilde{Z}_j^f$ deviate from each

---

[6]Our proposal also accommodates using separate predictors $\{f_j\}_{j=1}^m$ for each problem. To streamline exposition, we focus on the practical scenario where a single (large) model (e.g., a large language or vision model) can handle multiple tasks simultaneously (Radford et al., 2019; He et al., 2022).

[7]For EB, examples include Xie et al. (2012); Soloff et al. (2024); for PPI, see recent works like Fisch et al. (2024).

other. Nevertheless, $\tilde{Z}_j^f$ contains information that can help us improve upon $\bar{Y}_j$ as an estimator of $\theta_j$ by learning from within problem (PPI, PPI++, this work) and across problem (this work) structure. We emphasize that, as specified in (7), our approach only requires modeling the first and second moments of the joint distribution of $(f(X_{ij}), Y_{ij})$. For instance, in this synthetic model, $\theta_j = \eta_j^2$ and $\sigma_j^2 = 4\eta_j^2\psi^2 + c$, while $\mu_j$, $\tau_j^2$ and $\gamma_j$ also admit closed-form expressions in terms of $\eta_j$ when the predictor takes the form $f(x) = |x|$ or $f(x) = x^2$ (see Appendix E.1).

To conclude this section, we define the compound risk (Robbins, 1951; Jiang & Zhang, 2009) for any estimator $\hat{\boldsymbol{\theta}} = (\hat{\theta}_1, \ldots, \hat{\theta}_m)^{\intercal}$ as the expected squared error loss averaged over problems,

$$\mathcal{R}_m(\hat{\boldsymbol{\theta}}, \boldsymbol{\theta}) := \mathbb{E}_{\boldsymbol{\eta}}\left[\ell_m(\hat{\boldsymbol{\theta}}, \boldsymbol{\theta})\right], \tag{9}$$

$$\text{with} \quad \ell_m(\hat{\boldsymbol{\theta}}, \boldsymbol{\theta}) := \frac{1}{m}\sum_{j=1}^{m}(\hat{\theta}_j - \theta_j)^2. \tag{10}$$

The Bayes risk, which we also refer to simply as mean squared error (MSE), further integrates over randomness in the unknown prior $\mathbb{P}_\eta$ in (5),

$$\mathcal{B}_m^{\mathbb{P}_\eta}(\hat{\boldsymbol{\theta}}) := \mathbb{E}_{\mathbb{P}_\eta}\left[\mathcal{R}_m(\hat{\boldsymbol{\theta}}, \boldsymbol{\theta})\right]. \tag{11}$$

## 3. Statistical Guiding Principles & Prior Work

In this section, we illustrate both the statistical guiding principles of our approach and some connections to prior work[8] through the following stylized Gaussian model:

Sampling: $\hat{\theta}^{\text{cl}} = \theta + (\xi + \varepsilon)$, $\xi \sim \mathcal{N}(0, \sigma_\xi^2)$, $\varepsilon \sim \mathcal{N}(0, \sigma_\varepsilon^2)$.

Prior: $\theta \sim \mathcal{N}(0, \sigma_\theta^2)$, $\phi \sim \mathcal{N}(0, \sigma_\phi^2)$, $\text{Corr}[\theta, \phi] = \rho$.

In our stylized model, we assume that $(\theta, \phi, \varepsilon, \xi)$ are jointly normal and that all their pairwise correlations are zero with the exception of $\text{Corr}[\theta, \phi] = \rho \neq 0$. We write $\sigma_{\theta|\phi}^2 := \text{Var}[\theta \mid \phi] = (1 - \rho^2)\sigma_\theta^2 < \sigma_\theta^2$.

We think of $\hat{\theta}^{\text{cl}}$ as the baseline classical statistical estimator of a quantity $\theta$ that we seek to estimate with small MSE. In our stylized Gaussian model, $\hat{\theta}^{\text{cl}}$ is unbiased for $\theta$ and has noise contribution $\xi + \varepsilon$, so that $\mathbb{E}[(\hat{\theta}^{\text{cl}} - \theta)^2] = \text{Var}_\theta[\hat{\theta}^{\text{cl}}] = \sigma_\xi^2 + \sigma_\varepsilon^2$. We describe three high-level strategies used to improve the MSE of $\hat{\theta}^{\text{cl}}$. These strategies are not tied in any way to the stylized model; nevertheless, the stylized model enables us to give precise expressions for the risk reductions possible, see Table 1.

**Variance reduction (VR).** An important statistical idea is to improve $\hat{\theta}^{\text{cl}}$ via obtaining further information to intercept some of its noise, say $\xi$, and replacing $\hat{\theta}^{\text{cl}}$ by $\hat{\theta}^{\text{cl}} - \xi$

---

[8]We provide further connections in Appendix A.1.

*Table 1.* Estimator comparison in the stylized model of Section 3.

| Estimator | MSE | VR | P | CP |
|-----------|-----|----|----|----|
| $\hat{\theta}^{\text{cl}}$ | $\sigma_\xi^2 + \sigma_\varepsilon^2$ | ✗ | ✗ | ✗ |
| $\hat{\theta}^{\text{cl}} - \xi$ | $\sigma_\varepsilon^2$ | ✓ | ✗ | ✗ |
| $\mathbb{E}[\theta \mid \hat{\theta}^{\text{cl}}]$ | $\frac{(\sigma_\xi^2+\sigma_\varepsilon^2)\sigma_\theta^2}{(\sigma_\xi^2+\sigma_\varepsilon^2)+\sigma_\theta^2}$ | ✗ | ✓ | ✗ |
| $\mathbb{E}[\theta \mid \hat{\theta}^{\text{cl}}, \phi]$ | $\frac{(\sigma_\xi^2+\sigma_\varepsilon^2)\sigma_{\theta|\phi}^2}{(\sigma_\xi^2+\sigma_\varepsilon^2)+\sigma_{\theta|\phi}^2}$ | ✗ | ✓ | ✓ |
| $\mathbb{E}[\theta \mid \hat{\theta}^{\text{cl}} - \xi]$ | $\frac{\sigma_\varepsilon^2\sigma_\theta^2}{\sigma_\varepsilon^2+\sigma_\theta^2}$ | ✓ | ✓ | ✗ |
| $\mathbb{E}[\theta \mid \hat{\theta}^{\text{cl}} - \xi, \phi]$ | $\frac{\sigma_\varepsilon^2\sigma_{\theta|\phi}^2}{\sigma_\varepsilon^2+\sigma_{\theta|\phi}^2}$ | ✓ | ✓ | ✓ |

VR: Variance Reduction, P: Prior Information, CP: Contextual Prior Information.

which has MSE $\sigma_\varepsilon^2$ and remains unbiased for $\theta$. This idea lies at the heart of approaches such as control variates in simulation (Lavenberg & Welch, 1981; Hickernell et al., 2005), variance reduction in randomized controlled experiments via covariate adjustment (Lin, 2013) and by utilizing pre-experiment data (Deng et al., 2013, CUPED), as well as model-assisted estimation in survey sampling (Cochran, 1977; Breidt & Opsomer, 2017). It is also the idea powering PPI and related methods: the unlabeled dataset and the predictive model are used to intercept some of the noise in the classical statistical estimator $\hat{\theta}^{\text{cl}} \triangleq \bar{Y}$; compare to Eq. (2) with $\xi \triangleq \bar{Z}^f - \tilde{Z}^f$. We refer to Ji et al. (2025) and Gronsbell et al. (2025) for informative discussions of how PPI relates to traditional ideas in semi-parametric inference as in e.g., Robins et al. (1994).

**Prior information (P) via empirical Bayes (EB).** In the Bayesian approach we seek to improve upon $\hat{\theta}^{\text{cl}}$ by using the prior information that $\theta \sim \mathcal{N}(0, \sigma_\theta^2)$. The Bayes estimator,

$$\mathbb{E}[\theta \mid \hat{\theta}^{\text{cl}}] = \frac{\sigma_\theta^2}{\sigma_\xi^2 + \sigma_\varepsilon^2 + \sigma_\theta^2}\hat{\theta}^{\text{cl}},$$

reduces variance by shrinking $\hat{\theta}^{\text{cl}}$ toward 0 (at the cost of introducing some bias). When $\sigma_\theta^2$ is small, the MSE of $\mathbb{E}[\theta \mid \hat{\theta}^{\text{cl}}]$ can be substantially smaller than that of $\hat{\theta}^{\text{cl}}$.

Now suppose that the variance of the prior, $\sigma_\theta^2$, is unknown but we observe data from multiple related problems generated from the same model and indexed by $j \in [m]$, say, $\theta_j \overset{\text{iid}}{\sim} \mathcal{N}(0, \sigma_\theta^2)$ and $\hat{\theta}_j^{\text{cl}} \overset{\text{ind}}{\sim} \mathcal{N}(\theta_j, \sigma_\xi^2 + \sigma_\varepsilon^2)$. Then an EB analysis can mimic the MSE of the oracle Bayesian that has full knowledge of the prior. To wit, we can estimate $\sigma_\theta^2$ as

$$\hat{\sigma}_\theta^2 = \left\{\frac{1}{m-2}\sum_{j=1}^{m}(\hat{\theta}_j^{\text{cl}})^2\right\} - (\sigma_\xi^2 + \sigma_\varepsilon^2),$$

and then consider a plug-in approximation of the Bayes rule, $\hat{\theta}_j^{\text{JS}} = \hat{\mathbb{E}}[\theta_j \mid \hat{\theta}_j^{\text{cl}}] := \{\hat{\sigma}_\theta^2 / (\sigma_\xi^2 + \sigma_\varepsilon^2 + \hat{\sigma}_\theta^2)\}\hat{\theta}^{\text{cl}}$. The resulting estimator is the celebrated James-Stein estimator (James

& Stein, 1961; Efron & Morris, 1973), whose risk is very close to the Bayes risk under the hierarchical model for large $m$ (Efron, 2010, equation (1.25)). The James-Stein estimator also always dominates the classical estimator under a frequentist evaluation of compound risk in (9) under the assumption that $\hat{\theta}_j^{\text{cl}} \overset{\text{ind}}{\sim} \mathcal{N}(\theta_j, \sigma_\xi^2 + \sigma_\varepsilon^2)$ and $m \geq 3$:

$$\mathcal{R}_m(\hat{\boldsymbol{\theta}}^{\text{JS}}, \boldsymbol{\theta}) < \mathcal{R}_m(\hat{\boldsymbol{\theta}}^{\text{cl}}, \boldsymbol{\theta}) \ \text{ for all } \ \boldsymbol{\theta} \in \mathbb{R}^m.$$

**Contextual prior information (CP) via EB.** Instead of using the same prior for each problem, we may try to sharpen the prior and increase its relevance (Efron, 2011) by using further contextual information $\phi$. In the stylized example, as seen in Table 1, such an approach reduces the variance of the prior from $\sigma_\theta^2$ to $\sigma_{\theta|\phi}^2 < \sigma_\theta^2$ with corresponding MSE reduction of the Bayes estimator $\mathbb{E}[\theta \mid \hat{\theta}^{\text{cl}}, \phi]$. With multiple related problems, such a strategy can be instantiated via EB shrinkage toward an informative but biased predictor (Fay III & Herriot, 1979; Green & Strawderman, 1991; Mukhopadhyay & Maiti, 2004; Kou & Yang, 2017; Rosenman et al., 2023). The strategy of this form that is closest to our proposal is the covariate-powered EB approach of Ignatiadis & Wager (2019), recently applied to large language model evaluation by Fogliato et al. (2024). Therein (following the notation of Section 2.2), the analyst has access to classical estimators $\bar{Y}_j$, $j \in [m]$, and problem-specific covariates $W_j$ and seeks to shrink $\bar{Y}_j$ toward ML models that predict $\bar{Y}_j$ from $W_j$. By contrast, in our setting we have observation-level covariates $X_{ij}$ and the ML model operates on these covariates. In principle one could simultaneously use both types of covariates: problem-specific and observation-specific.

**Combine variance reduction (VR) and prior information (P).** One can shrink the variance reduced estimator $\hat{\theta}^{\text{cl}} - \xi$ toward 0 via $\mathbb{E}[\theta \mid \hat{\theta}^{\text{cl}} - \xi] = \{\sigma_\theta^2/(\sigma_\varepsilon^2 + \sigma_\theta^2)\}(\hat{\theta}^{\text{cl}} - \xi)$. In the context of PPI, variance reduction and prior information (with a more heavy-tailed prior) are used by Cortinovis & Caron (2025) within the Frequentist-Assisted by Bayes (FAB) framework of Yu & Hoff (2018). Cortinovis & Caron (2025) only consider a single problem and do not pursue an empirical Bayes approach.

**Combine P, CP, and VR together.** Finally, in our stylized example, we can get the smallest MSE (last row of Table 1) by using both variance reduction, shrinkage, and a contextual prior. In that case, the Bayes estimator $\mathbb{E}[\theta \mid \hat{\theta}^{\text{cl}} - \xi, \phi]$ takes the form,

$$\frac{\sigma_{\theta|\phi}^2}{\sigma_\varepsilon^2 + \sigma_{\theta|\phi}^2}(\hat{\theta}^{\text{cl}} - \xi) + \frac{\sigma_\varepsilon^2}{\sigma_\varepsilon^2 + \sigma_{\theta|\phi}^2}\mathbb{E}[\theta \mid \phi]. \quad (12)$$

EB ideas can be used to mimic the estimator above and provide the starting point for the proposal we describe next.

# 4. Prediction-Powered Adaptive Shrinkage

On a high level, PAS aims to provide a lightweight approach that outperforms both baselines in (2) and PPI/PPI++ in terms of MSE when estimating multiple means. PAS also aims at minimal modeling requirements and assumptions.

The stylized example from Section 3 serves as a guiding analogy. We seek to benefit from ML predictions in two ways: first by variance reduction (acting akin to $\xi$ in the stylized example), and second by increasing prior relevance (acting as a proxy for $\phi$). We implement both steps to adapt to the unknown data-generating process in an assumption-lean way using *within*-problem information for the first step (Section 4.1) and *across*-problem information for the second step (Section 4.2), drawing on ideas from the EB literature.

## 4.1. The Within Problem Power-Tuning Stage

Extending the notation from (4) to each problem $j$ provides us with a class of unbiased estimators $\hat{\theta}_{j,\lambda}^{\text{PPI}} := \bar{Y}_j + \lambda(\tilde{Z}_j^f - \bar{Z}_j^f)$, $\lambda \in \mathbb{R}$. Calculating the variance gives

$$\text{Var}_{\eta_j}\left[\hat{\theta}_{j,\lambda}^{\text{PPI}}\right] = \frac{\sigma_j^2}{n_j} + \overbrace{\frac{n_j + N_j}{n_j N_j}\lambda^2\tau_j^2 - \frac{2}{n_j}\lambda\gamma_j}^{=: \delta_j(\lambda)}.$$

Note that the classical estimator has risk $\sigma_j^2/n_j$ and gets outperformed whenever $\delta_j(\lambda) < 0$. We can further analytically solve for the optimal $\lambda$, which yields

$$\lambda_j^* := \arg\min_\lambda \delta_j(\lambda) = \left(\frac{N_j}{n_j + N_j}\right)\frac{\gamma_j}{\tau_j^2}, \quad (13)$$

and the Power-Tuned (PT) estimator $\hat{\theta}_j^{\text{PT}} := \hat{\theta}_{j,\lambda_j^*}^{\text{PPI}}$ with

$$\tilde{\sigma}_j^2 := \text{Var}_{\eta_j}\left[\hat{\theta}_j^{\text{PT}}\right] = \frac{\sigma_j^2}{n_j} - \frac{N_j}{n_j(n_j + N_j)}\frac{\gamma_j^2}{\tau_j^2}. \quad (14)$$

The formulation of the above PT estimators is well understood in the single problem setting (Angelopoulos et al., 2024; Miao et al., 2024a). In PAS, we execute this stage separately for each problem, as the optimal power-tuning parameter is problem-dependent and varies case by case.

## 4.2. The Across Problem Adaptive Shrinkage Stage

The PT estimator derived in Section 4.1 already possesses many appealing properties: it is unbiased and has lower variance than both the classical estimator and vanilla PPI. However, as our setting involves working with many parallel problems together, there is the opportunity of further MSE reduction by introducing bias in a targeted way.[9] Concretely, based on the PT estimator obtained in

---

[9]See Appendix A.2.1 for some explanations about why we can improve MSE by borrowing information across problems.

Section 4.1, we consider a class of shrinkage estimators $\hat{\boldsymbol{\theta}}_\omega^{\mathrm{PAS}} := (\hat{\theta}_{1,\omega}^{\mathrm{PAS}}, \ldots, \hat{\theta}_{m,\omega}^{\mathrm{PAS}})^\intercal$, where for any $\omega \geq 0$,

$$\hat{\theta}_{j,\omega}^{\mathrm{PAS}} := \omega_j \hat{\theta}_j^{\mathrm{PT}} + (1 - \omega_j) \tilde{Z}_j^f,$$
$$\text{with} \quad \omega_j \equiv \omega_j(\omega) := \frac{\omega}{\omega + \tilde{\sigma}_j^2}. \tag{15}$$

The motivation is to formally match the form of the Bayes estimator with variance reduction and contextual prior information in (12) with the following (approximate) analogies:[10]

$$\hat{\theta}^{\mathrm{cl}} - \xi \longleftrightarrow \hat{\theta}_j^{\mathrm{PT}}, \quad \mathbb{E}[\theta \mid \phi] \longleftrightarrow \tilde{Z}_j^f,$$
$$\sigma_\varepsilon^2 \longleftrightarrow \tilde{\sigma}_j^2, \quad \boxed{\sigma_{\theta|\phi}^2 \longleftrightarrow \omega}. \tag{16}$$

The highlighted $\omega$ is a global shrinkage parameter that acts as follows:

(i) Fixing $\omega$, any problem whose PT estimator has higher variance possesses smaller $\omega_j$ and shrinks more toward $\tilde{Z}_j^f$; a smaller variance increases $\omega_j$ and makes the final estimator closer to $\hat{\theta}_j^{\mathrm{PT}}$.

(ii) Fixing all the problems, increasing $\omega$ has an overall effect of recovering $\hat{\theta}_j^{\mathrm{PT}}$ for all $j$ (full recovery when $\omega \to \infty$), and setting $\omega = 0$ recovers $\tilde{Z}_j^f$.

Points (i) and (ii) establish the conceptual importance of $\omega$. If we could choose $\omega$ in an optimal way, that is,

$$\omega^* \in \operatorname*{arg\,min}_{\omega \geq 0} \left\{ \mathcal{R}_m\left(\hat{\boldsymbol{\theta}}_\omega^{\mathrm{PAS}}, \boldsymbol{\theta}\right) \right\},$$

then the resulting estimator $\hat{\boldsymbol{\theta}}_{\omega^*}^{\mathrm{PAS}}$ would satisfy all our desiderata. While this construction is not feasible since the compound risk function in (9) depends on the unknown $\boldsymbol{\eta}, \boldsymbol{\theta}$, we can make progress by pursuing a classical statistical idea: we can develop an unbiased estimate of the compound risk (Mallows, 1973; Stein, 1981; Efron, 2004) and then use it as a surrogate for tuning $\omega$.

To this end, we define the Correlation-aware Unbiased Risk Estimate (CURE),

$$\mathrm{CURE}\left(\hat{\boldsymbol{\theta}}_\omega^{\mathrm{PAS}}\right) := \frac{1}{m} \sum_{j=1}^m \Big[ (2\omega_j - 1)\tilde{\sigma}_j^2 + 2(1 - \omega_j)\tilde{\gamma}_j$$
$$+ (1 - \omega_j)^2 \big(\hat{\theta}_{j,\omega_j}^{\mathrm{PT}} - \tilde{Z}_j^f\big)^2 \Big].$$

Both the formula and our nomenclature ("correlation-aware") highlight the fact that we must account for the potentially non-zero covariance between shrinkage source $\hat{\theta}_j^{\mathrm{PT}}$ and target $\tilde{Z}_j^f$, which can be explicitly written down as

$$\tilde{\gamma}_j := \mathrm{Cov}_{\eta_j}[\hat{\theta}_j^{\mathrm{PT}}, \tilde{Z}_j^f] = \lambda_j^* \mathrm{Var}_{\eta_j}[\tilde{Z}_j^f] = \frac{\gamma_j}{n_j + N_j}. \tag{17}$$

[10]We comment more on these analogies in Appendix A.2.2.

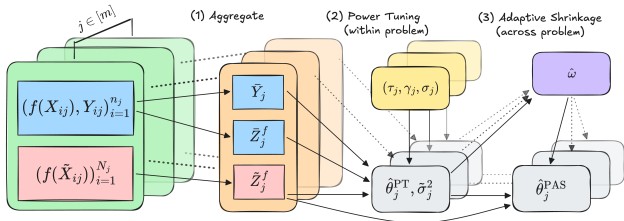

*Figure 2.* A flowchart illustration of the PAS method. See Algorithm 1 for a pseudo-code implementation.

---

**Algorithm 1** Prediction-Powered Adaptive Shrinkage

**Require:** $(X_{ij}, Y_{ij})_{i=1}^{n_j}$, $(\tilde{X}_{ij})_{i=1}^{N_j}$, $\gamma_j, \tau_j, \sigma_j$ for $j \in [m]$, predictive model $f$
1: **for** $j = 1$ to $m$ **do**
2:    ▷ Step 1: Apply predictor (Eq. (1))
3:    $\bar{Y}_j, \bar{Z}_j^f, \tilde{Z}_j^f = \mathtt{get\_means}((X_{ij}, Y_{ij})_{i=1}^{n_j}, (\tilde{X}_{ij})_{i=1}^{N_j}, f)$
4:    ▷ Step 2: Power tuning (Eq. (13))
5:    $\lambda_j^* = \mathtt{get\_pt\_param}(\gamma_j, \tau_j, n_j, N_j)$
6:    $\hat{\theta}_j^{\mathrm{PT}} = \bar{Y}_j + \lambda_j^*(\tilde{Z}_j^f - \bar{Z}_j^f)$
7:    $\tilde{\sigma}_j^2 = \mathtt{get\_pt\_var}(\hat{\theta}_j^{\mathrm{PT}})$ ▷ (Eq. (14))
8: **end for**
9: ▷ Step 3: Adaptive shrinkage (Eq. (18))
10: $\hat{\omega} = \mathtt{get\_shrink\_param}((\hat{\theta}_j^{\mathrm{PT}})_{j=1}^m, (\tilde{Z}_j^f)_{j=1}^m, (\tilde{\sigma}_j^2)_{j=1}^m)$
11: **for** $j = 1$ to $m$ **do**
12:    $\hat{\omega}_j = \hat{\omega}/(\hat{\omega} + \tilde{\sigma}_j^2)$
13:    $\hat{\theta}_j^{\mathrm{PAS}} = \hat{\omega}_j \hat{\theta}_j^{\mathrm{PT}} + (1 - \hat{\omega}_j)\tilde{Z}_j^f$
14: **end for**
15: **return** $\{\hat{\theta}_j^{\mathrm{PAS}}\}_{j=1}^m$

---

**Theorem 4.1.** *Under Assumption 2.2, CURE is an unbiased estimator of the compound risk defined in (9), that is, for all $\omega \geq 0$ and all $\boldsymbol{\eta}$,*

$$\mathbb{E}_{\boldsymbol{\eta}}\left[\mathrm{CURE}\left(\hat{\boldsymbol{\theta}}_\omega^{\mathrm{PAS}}\right)\right] = \mathcal{R}_m\left(\hat{\boldsymbol{\theta}}_\omega^{\mathrm{PAS}}, \boldsymbol{\theta}\right).$$

See Appendices B and F.1 for the proof and motivation. With Theorem 4.1 in hand, we now have a systematic strategy of picking $\omega$ by minimizing CURE, following the paradigm of tuning parameter selection via minimization of an unbiased risk estimate (as advocated by, e.g. Li (1985); Donoho & Johnstone (1995); Xie et al. (2012); Candès et al. (2013); Ignatiadis & Wager (2019); Ghosh et al. (2025)):[11]

$$\hat{\omega} \in \operatorname*{arg\,min}_{\omega \geq 0} \mathrm{CURE}\left(\hat{\boldsymbol{\theta}}_\omega^{\mathrm{PAS}}\right). \tag{18}$$

Even though $\hat{\omega}$ does not admit a closed-form expression, the one-dimensional minimization can be efficiently carried out numerically (e.g., grid search). The final PAS estimator is:

[11]The connection to EB is the following. Xie et al. (2012) and Tibshirani & Rosset (2019) explain that James-Stein-type estimators may be derived by tuning $\sigma_\theta^2$ (in Section 3) via minimization of Stein's (1981) unbiased risk estimate (SURE).

$$\hat{\theta}_j^{\mathrm{PAS}} := \hat{\theta}_{j,\hat{\omega}}^{\mathrm{PAS}} = \frac{\hat{\omega}}{\hat{\omega} + \tilde{\sigma}_j^2} \hat{\theta}_j^{\mathrm{PT}} + \frac{\tilde{\sigma}_j^2}{\hat{\omega} + \tilde{\sigma}_j^2} \tilde{Z}_j^f.$$

Figure 2 visualizes the full method for constructing the `PAS` estimator—from applying the predictor and obtaining aggregated statistics to going through the two stages described in Section 4.1 and this section. A pseudo-code implementation is also presented in Algorithm 1.

To illustrate the flexibility and adaptivity of `PAS`, we briefly revisit the synthetic model in Example 2.3, whose special structure allows us to visualize how the power-tuned and adaptive shrinkage parameters vary across problems and different predictors. In Figure 3, we consider $m = 200$ problems and two predictors: a good predictor $f_1(x) = x^2$ and a flawed predictor $f_2(x) = |x|$. The model setup in (8) is such that the magnitude of $\mathrm{Cov}_{\eta_j}[X_j, Y_j]$ relative to $\mathrm{Var}_{\eta_j}[Y_j]$ is much larger for problems with $\eta_j$ closer to the origin. Therefore, for both predictors, we see a dip in $\lambda_j^*$ near the middle (top panel), which shows that `PAS` adapts to the level of difficulty of each problem when deciding how much power-tuning to apply. On the other hand (bottom panel), the overall shrinkage effect is much stronger (smaller $\hat{\omega}_j$ for all $j$) with $f_1$ than with $f_2$, which demonstrates `PAS`'s ability to adapt to the predictor's quality across problems—while still allowing each problem to have its own shrinkage level. Numerical results are postponed to Section 7.

# 5. Asymptotic Optimality

In (18), we proposed selecting $\hat{\omega}$ by optimizing an unbiased surrogate of true risk. In this section, we justify this procedure theoretically. Our first result establishes that CURE approximates the true loss (whose expectation is the compound risk in (9)) uniformly in $\omega$ as we consider more and more problems.

**Proposition 5.1.** *Suppose the datasets are generated according to Assumptions 2.1 and 2.2 and further assume that $\mathbb{E}_{\mathbb{P}_\eta}[Y_{ij}^4] < \infty$, $\mathbb{E}_{\mathbb{P}_\eta}[f(X_{ij})^4] < \infty$. Then,*

$$\mathbb{E}_{\mathbb{P}_\eta} \left[ \sup_{\omega \geq 0} \left| \mathrm{CURE}\left( \hat{\theta}_\omega^{\mathrm{PAS}} \right) - \ell_m\left( \hat{\theta}_\omega^{\mathrm{PAS}}, \theta \right) \right| \right] = o(1),$$

*where $o(1)$ denotes a term that converges to 0 as $m \to \infty$.*

A principal consequence of Proposition 5.1 is that `PAS` with the data-driven choice of $\hat{\omega}$ in (18) has asymptotically smaller Bayes MSE (defined in (11)) than any of the estimators in (15), i.e., it has smaller MSE than both baselines in (2) as well as the PPI and PT estimators.

**Theorem 5.2.** *Under the assumptions of Proposition 5.1,*

$$\mathcal{B}_m^{\mathbb{P}_\eta}\left( \hat{\theta}_{\hat{\omega}}^{\mathrm{PAS}} \right) \leq \inf_{\omega \geq 0} \left\{ \mathcal{B}_m^{\mathbb{P}_\eta}\left( \hat{\theta}_\omega^{\mathrm{PAS}} \right) \right\} + o(1) \text{ as } m \to \infty,$$

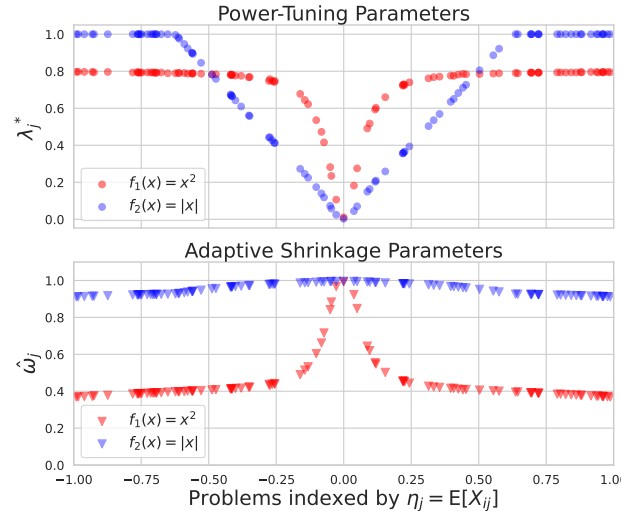

**Figure 3.** The power-tuned and adaptive shrinkage parameters, $\lambda_j^*$ and $\hat{\omega}_j$ across $m = 200$ problems in Example 2.3. On the $x$-axis, we identify the problem by its $\eta_j$ so the trend is more visible.

*so* $\mathcal{B}_m^{\mathbb{P}_\eta}\left( \hat{\theta}_{\hat{\omega}}^{\mathrm{PAS}} \right) \leq \min \left\{ \mathcal{B}_m^{\mathbb{P}_\eta}\left( \tilde{\mathbf{Z}}^f \right), \mathcal{B}_m^{\mathbb{P}_\eta}\left( \hat{\theta}^{\mathrm{PT}} \right) \right\} + o(1).$

Our next proposition connects Theorem 5.2 with the lowest possible MSE in the stylized Gaussian example of Section 3 (last row of Table 1).

**Proposition 5.3.** *In addition to the assumptions of Proposition 5.1, further assume that $N_j = \infty$ and that there exist $n \in \mathbb{N}$, $\tilde{\sigma}^2 > 0$ such that $n_j = n$ and $\tilde{\sigma}_j^2 = \tilde{\sigma}^2$ for all $j$ almost surely. Let $\beta^2 := \mathbb{E}_{\mathbb{P}_\eta}[(\tilde{Z}_j^f - \theta_j)^2]$ (which does not depend on $j$ as we are integrating over $\mathbb{P}_\eta$). Then,*

$$\mathcal{B}_m^{\mathbb{P}_\eta}\left( \hat{\theta}_{\hat{\omega}}^{\mathrm{PAS}} \right) \leq \frac{\tilde{\sigma}^2 \beta^2}{\tilde{\sigma}^2 + \beta^2} + o(1) \text{ as } m \to \infty.$$

To interpret the result, it is instructive to compare the asymptotic upper bound on the MSE of `PAS` with the MSE in the last line of Table 1, i.e., with $(\sigma_\varepsilon^2 \sigma_{\theta|\phi}^2)/(\sigma_\varepsilon^2 + \sigma_{\theta|\phi}^2)$. Observe that $\tilde{\sigma}^2$ plays the role of $\sigma_\varepsilon^2$ (as already anticipated in (16)) which is smaller than the variance of the classical estimator (due to power tuning). Meanwhile, $\beta^2$ plays the role of $\sigma_{\theta|\phi}^2$. If the baseline $\tilde{Z}_j^f$ (that is, the mean of the ML predictions on the unlabeled datasets) is doing a good job of predicting $\theta_j$, then $\beta^2$ will be small, and so `PAS` may have MSE substantially smaller than that of PT. On the other hand, even if $\beta^2$ is large (that is, even if the ML model is very biased), `PAS` asymptotically still has MSE less than or equal to $\tilde{\sigma}^2$, the MSE of PT. We emphasize that the role of Proposition 5.3 is to provide intuition at the expense of strong assumptions.[12] By contrast, the statement of Theorem 5.2 does not restrict heterogeneity (e.g., heteroscedasticity) across problems and allows for varying, finite unlabeled and labeled sample sizes.

---

[12]We further elaborate on this proposition in Appendix A.2.3.

## 6. PAS with Unknown Second Moments

So far we have assumed that the second moments $\sigma_j^2, \tau_j^2, \gamma_j$ in (7) are known. In practice, e.g., in our numerical experiments with real-world datasets, we use sample-based estimates $\hat{\sigma}_j^2, \hat{\tau}_j^2$ and $\hat{\gamma}_j$ instead (Appendix C.1). This approach works well empirically, even for small $n_j$, but is not covered by the theory above. To address this shortcoming, we develop variants UniPT (Appendix C.2) and UniPAS (Appendix C.3) that provide asymptotic guarantees (as $m \to \infty$) without requiring known or consistently estimable second moments ($n_j, N_j$ remain bounded). Briefly, UniPT targets an optimal single power-tuning parameter $\lambda$ across all problems, and does so consistently as $m \to \infty$. UniPAS then builds upon UniPT, applying adaptive shrinkage similar to PAS, but with mechanisms to handle the unknown moments. These extensions are justified by theoretical results akin to those for PAS (cf. Theorem 5.2). In our experiments below, UniPAS is competitive with PAS.

## 7. Experiments

We apply the proposed PAS estimator and conduct extensive experiments in both the synthetic model proposed in Example 2.3 and two real-world datasets.

**All Estimators.** We compare the PAS estimator against both classical and modern baseline estimators:

(i.) the classical estimator;

(ii.) the prediction mean on unlabeled data;

(iii.) the vanilla PPI estimator (Angelopoulos et al., 2023);

(iv.) the PT estimator (Angelopoulos et al., 2024, PPI++);

(v.) the "shrink-classical" estimator that directly shrinks the classical estimator toward the prediction mean;

(vi.) the "shrink-average" estimator that shrinks $\hat{\theta}_j^{\mathrm{PT}}$ toward the PT group mean across all problems, i.e., toward $\bar{\theta}^{\mathrm{PT}} := \frac{1}{m} \sum_{j=1}^{m} \hat{\theta}_j^{\mathrm{PT}}$.

(vii.) the UniPT and UniPAS estimators introduced in Section 6 and detailed in Appendix C.

We include the estimators in (vii.) only for the real-world experiments, as they are specifically designed for settings where the second moments are unknown. See Appendix D for detailed formulations and implementations for (v.) & (vi.) (the latter of which is inspired by the SURE-grand mean estimator of Xie et al. (2012)). The comparisons with (iv.) and (v.) also directly serve as ablation studies of the two stages in constructing the PAS estimator.

**Metrics.** We report the mean squared error (MSE) ($\pm$ 1 standard error) of each estimator $\hat{\theta}$ by averaging

| Estimator | MSE $f_1$ ($\times 10^{-3}$) | MSE $f_2$ ($\times 10^{-3}$) |
|---|---|---|
| Classical | $3.142 \pm 0.033$ | $3.142 \pm 0.033$ |
| Prediction Avg | $0.273 \pm 0.004$ | $34.335 \pm 0.147$ |
| PPI | $2.689 \pm 0.027$ | $2.756 \pm 0.027$ |
| PT | $2.642 \pm 0.027$ | $2.659 \pm 0.026$ |
| Shrink Classical | $0.273 \pm 0.003$ | $3.817 \pm 0.042$ |
| Shrink Avg | $2.486 \pm 0.026$ | $2.575 \pm 0.026$ |
| PAS **(ours)** | $\mathbf{0.272 \pm 0.003}$ | $\mathbf{2.496 \pm 0.025}$ |

*Table 2.* MSE ($\pm$ standard error) of different estimators under the synthetic model with predictors $f_1(x) = x^2$ and $f_2(x) = |x|$.

$\frac{1}{m} \sum_{j=1}^{m} (\hat{\theta}_j - \theta_j)^2$ across $K = 200$ Monte Carlo replicates. In the synthetic model, we sample $\eta_j$ (and thus $\theta_j$) from the known prior $\mathbb{P}_\eta$. For the real-world datasets, since $\mathbb{P}_\eta$ is unknown, we follow the standard evaluation strategy in the PPI literature: we start with a large labeled dataset and use it to compute a pseudo-ground truth for each mean $\theta_j$. Then in each Monte Carlo replicate, we randomly split the data points of each problem into labeled/unlabeled partitions (where we choose a 20/80 split ratio).[13]

For real-world datasets, we introduce a second metric to assess whether improvements in MSE are driven by a few difficult problems rather than consistent performance gains: the percentage of problems improved relative to the classical estimator (abbreviated as "Improved % ↑"). This metric is defined as $\frac{1}{m} \sum_{j=1}^{m} \mathbf{1}[(\hat{\theta}_j - \theta_j)^2 < (\hat{\theta}_j^{\mathrm{Classical}} - \theta_j)^2] \times (100\%)$. Larger values of this metric are preferable.

### 7.1. Synthetic Model

This is the synthetic model from Example 2.3, where we choose $m = 200$, $n_j = 20$, and $N_j = 80$ for all $j$. Since we have already visualized the model and the parameters of the PAS estimator in previous sections, we simply report the numerical results (for the good predictor $f_1$ and the flawed predictor $f_2$) in Section 7.1.

For both predictors, we see that PAS outperforms all the baselines. With a good predictor $f_1$, both the prediction mean and the shrinkage estimator closely track PAS; in contrast, the PPI and PT estimators fail to fully leverage the accurate predictions, as their design enforces unbiasedness. The situation reverses for the less reliable predictor $f_2$: the prediction mean and the shrinkage estimator have high MSE, while estimators with built-in debiasing mechanisms demonstrate greater resilience. PAS adapts effectively across these extremes, making it a handy choice for a wide range of problems and predictors.

### 7.2. Real-World Datasets

We next evaluate PAS on two large-scale real-world datasets, highlighting its ability to leverage state-of-the-art deep learn-

---

[13]In Appendix E, we vary this ratio from 1% to 40%, and provide more details on the benchmarking procedure for MSE.

*Table 3.* Results aggregated over $K = 200$ replicates on three real-world datasets: Amazon review ratings with `BERT-base` and `BERT-tuned` predictors, and spiral galaxy fractions with `ResNet50` predictor. The bottom two rows are the `UniPT` and `UniPAS` estimators detailed in Appendix C as variants of PT and PAS. Metrics are reported with $\pm 1$ standard error.

| Estimator | Amazon (base $f$) | | Amazon (tuned $f$) | | Galaxy | |
|---|---|---|---|---|---|---|
| | MSE ($\times 10^{-3}$) | % Improved ↑ | MSE ($\times 10^{-3}$) | % Improved ↑ | MSE ($\times 10^{-3}$) | % Improved ↑ |
| Classical | $24.305 \pm 0.189$ | baseline | $24.305 \pm 0.189$ | baseline | $2.073 \pm 0.028$ | baseline |
| Prediction Avg | $41.332 \pm 0.050$ | $30.7 \pm 0.2$ | $3.945 \pm 0.011$ | $75.4 \pm 0.2$ | $7.195 \pm 0.008$ | $17.0 \pm 0.2$ |
| PPI | $11.063 \pm 0.085$ | $62.4 \pm 0.2$ | $7.565 \pm 0.066$ | $70.4 \pm 0.2$ | $1.149 \pm 0.017$ | $59.4 \pm 0.3$ |
| PT | $10.633 \pm 0.089$ | $70.3 \pm 0.2$ | $6.289 \pm 0.050$ | $76.0 \pm 0.2$ | $1.026 \pm 0.015$ | $67.7 \pm 0.3$ |
| Shrink Classical | $15.995 \pm 0.121$ | $56.4 \pm 0.3$ | $3.828 \pm 0.039$ | $78.9 \pm 0.2$ | $1.522 \pm 0.016$ | $48.8 \pm 0.4$ |
| Shrink Avg | $9.276 \pm 0.078$ | $70.4 \pm 0.2$ | $6.280 \pm 0.058$ | $77.1 \pm 0.2$ | $0.976 \pm 0.014$ | $\mathbf{68.9 \pm 0.3}$ |
| PAS (ours) | $\mathbf{8.517 \pm 0.071}$ | $\mathbf{71.4 \pm 0.2}$ | $\mathbf{3.287 \pm 0.024}$ | $\mathbf{80.8 \pm 0.2}$ | $\mathbf{0.893 \pm 0.011}$ | $67.3 \pm 0.4$ |
| UniPT (ours) | $10.272 \pm 0.084$ | $70.0 \pm 0.2$ | $6.489 \pm 0.053$ | $76.2 \pm 0.2$ | $1.017 \pm 0.015$ | $67.7 \pm 0.3$ |
| UniPAS (ours) | $8.879 \pm 0.073$ | $69.5 \pm 0.2$ | $3.356 \pm 0.031$ | $77.6 \pm 0.2$ | $0.909 \pm 0.011$ | $66.7 \pm 0.3$ |

ing models in different settings.[14]

**Amazon Review Ratings (SNAP, 2014).** Many commercial and scientific studies involve collecting a large corpus of text and estimating an average score (rating, polarity, etc.) from it. A practitioner would often combine limited expert annotations with massive automatic evaluations from ML models (Baly et al., 2020; Egami et al., 2023; Fan et al., 2024; Mozer & Miratrix, 2025). To emulate this setup, we consider mean rating estimation problems using the Amazon Fine Food Review dataset from Kaggle, where we artificially hide the labels in a random subset of the full data to serve as the unlabeled partition. Concretely, we estimate the average rating for the top $m = 200$ products with the most reviews (from ~200 to ~900). For the $i$-th review of the $j$-th product, the covariate $X_{ij}$ consists of the review's title and text concatenated, while the outcome $Y_{ij}$ is the star rating in $\{1, \ldots, 5\}$. We employ two black-box predictors: (1) `BERT-base`, a language model without fine-tuning (Devlin et al., 2019) and (2) `BERT-tuned` which is the same model but fine-tuned on a held-out set of reviews from other products. Neither of them are trained on the reviews of the 200 products we evaluate.

**Spiral Galaxy Fractions (Willett et al., 2013).** The Galaxy Zoo 2 project contains the classification results of galaxy images from the the Sloan Digital Sky Survey (York et al., 2000, SDSS). We are interested in estimating the fraction of galaxies that are classified as "spiral," i.e., have at least one spiral arm. The covariates $X_{ij}$ in this applications are images (we provide some examples in Figure 5 of the appendix). Existing PPI papers have focused on estimating the overall fraction; we demonstrate how this dataset's metadata structure enables compound estimation of spiral fractions across distinct galaxy subgroups. We first use a pre-defined partition of the galaxies into 100 subgroups that

is based on the metadata attribute `WVT_BIN`. Second, we estimate the fraction of spiral galaxies in all of the galaxy subgroups simultaneously. The predictor is a `ResNet50` network trained on a held-out set with around 50k images.

For both datasets, we randomly split the data for each problem (a food product or galaxy subgroup) into a labeled and unlabeled partition with a 20/80 ratio. We repeat the random splitting $K = 200$ times and report metrics averaged over all splits. Here we summarize the results in Table 3:

**Amazon Review:** similar to the trend in the synthetic model, the more accurate `BERT-tuned` model enables stronger shrinkage for `PAS` while the biased `BERT-base` predictions necessitate less shrinkage. Our `PAS` estimator adapts to both predictors and outperforms other baselines. `PAS` has the lowest MSE and highest % Improved ↑.

**Galaxy Zoo 2:** the predictions from `ResNet50` are suboptimal, so the variance-reduction from power tuning dominates any benefit from shrinkage. `PAS` achieves the lowest MSE among all estimators, and improves individual estimates at a level on par with the shrink-average estimator.

## 8. Conclusion

This paper introduces `PAS`, a novel method for compound mean estimation that effectively combines PPI and EB principles. We motivate the problem through the lens of variance reduction and contextual prior information—then demonstrate how `PAS` achieves both goals, in theory and in practice. Our paper differs from many other PPI-related works in its focus on estimation, so a natural next step is to develop average coverage controlling intervals for the means centered around `PAS`. To this end, it may be fruitful to build on the robust empirical Bayes confidence intervals of Armstrong et al. (2022). Modern scientific inquiries increasingly demand the simultaneous analysis of multiple related problems. The framework developed in this paper represents a promising direction for such settings.

---

[14]We include only the essential setup below and defer additional data and model details (e.g., hyper-parameters, preprocessing) to Appendix E.

## Acknowledgments

We thank Dan Kluger and Qingqi Zhang for thoughtful feedback on an earlier version of our manuscript. We also thank Claire Donnat, Yichen Ji, Lihua Lei, Aaron Schein, and Tijana Zrnic for helpful discussions. We are grateful to Bennett Hunter and the compute cluster at the Data Science Institute for supporting our computational resources. We thank all four reviewers for their constructive feedback. N.I. gratefully acknowledges support from NSF (DMS 2443410).

## Impact Statement

By developing a method to improve the efficiency and accuracy of mean estimation using machine learning predictions, this research has the potential to enhance data analysis across various domains where labeled data is limited but predictive models are available.

We acknowledge that advancements in machine learning can have broader societal consequences. However, the ethical considerations directly arising from this methodological contribution are those typically associated with the general advancement of statistical methods and machine learning. We do not foresee specific negative ethical impacts unique to this work that require further detailed discussion.

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

# ♣ Appendix: Table of Contents

## A. Further Connections and Intuitions Behind PAS

### A.1. Further Connections to Related Works

In this part, we elaborate on two important connections to existing work.

**Stratified PPI (StratPPI; Fisch et al. (2024)).** Fisch et al. (2024) also consider a setting with stratification of observations into subproblems. However, their overall aim is still to estimate a single parameter, rather than multiple parameters, as we do. Thus, in solving individual problems, for instance, they still adhere to the requirement of unbiasedness. As an example, let us revisit the Galaxy Zoo 2 application. Therein, we mentioned the following distinction between our work and previous methods in the PPI literature:

- Estimate the fraction $\theta$ of spiral galaxies across the whole universe (Angelopoulos et al., 2023).

- (Our work) Estimate the fraction $\theta_j$ of spiral galaxies within the $j$-th cluster of galaxies for $j \in [m]$.

Now suppose that galaxies in the whole universe can be partitioned into the $m$ clusters above. Then the goal of StratPPI is:

*Estimate the fraction $\theta$ of spiral galaxies across the whole universe by proceeding with an intermediate step that involves estimating $\theta_1, \ldots, \theta_m$.*

We continue to make this connection explicit in our mean estimation setting. Suppose (dropping subscripts for convenience) that we start with covariate-outcome pairs $(X, Y) \in \mathcal{X} \times \mathcal{Y}$ distributed as

$$(X, Y) \overset{\text{iid}}{\sim} \mathbb{P}.$$

Moreover suppose that there exist pairwise disjoint strata $\mathcal{A}_1, \ldots, \mathcal{A}_m$ whose union is the full covariate space $\mathcal{X}$, that is, $\mathcal{X} = \dot\bigcup_{j=1}^{m} \mathcal{A}_j$. Then define,

$$w_j := \mathbb{P}[X \in \mathcal{A}_j], \qquad \theta_j := \mathbb{P}[Y \mid X \in \mathcal{A}_j], \qquad j \in [m],$$

and observe the key equality for $\theta := \mathbb{E}[Y]$,

$$\theta = \sum_{j=1}^{m} w_j \theta_j.$$

StratPPI assumes that the probabilities $w_j$ are known exactly and then samples $n_j$ labeled as well as $N_j$ unlabeled observations from the conditional distribution $\mathbb{P}[\cdot \mid X \in \mathcal{A}_j]$.[15] These samples are then used alongside PPI++ to estimate $\theta_j$

---

[15]Part of the contribution of StratPPI includes a strategy for allocating resources and choosing $n_j$ and $N_j$ for different problems with a view toward minimizing the variance of the final estimator of $\theta$.

by $\hat{\theta}_j^{\mathrm{PT}}$, almost verbatim to the approach we described in Section 4.1. Finally, StratPPI estimates $\theta$ via:

$$\hat{\theta}^{\mathrm{SPPI}} := \sum_{j=1}^{m} w_j \hat{\theta}_j^{\mathrm{PT}}.$$

To summarize, the settings of StratPPI and our paper are similar, but the goals and methods differ. StratPPI seeks an unbiased estimate of a single parameter $\theta$, while we seek to estimate a parameter vector $(\theta_1, \ldots, \theta_m)$ as well as possible in mean squared error, while allowing for the possibility of bias. Moreover, the asymptotic regimes in these two papers are different: StratPPI keeps $m$ fixed and takes $n_j, N_j \to \infty$, while we assume that the number of problems grows ($m \to \infty$) while $n_j, N_j$ are allowed to remain bounded.

**PPI++ (Angelopoulos et al., 2024) with multivariate mean estimand.** Angelopoulos et al. (2024) (and other papers in the PPI literature) consider statistical problems going beyond the estimation of a univariate mean. For instance, the results of Angelopoulos et al. (2024) also accommodate multivariate mean estimands. Here we explain how one can frame our compound mean estimation setting into the estimation of a *single but multivariate mean estimand*, which then fits in the more generalized formulation for power tuning considered in PPI++.

If we assume $n_j = n$ and $N_j = N$ for all $j$, we can collate the responses and predictions from all problems into $(f(X_{i.}), Y_{i.})_{i=1}^{n}$ and $(f(\tilde{X}_{i.}))_{i=1}^{N}$, where

$$Y_{i.} = (Y_{i1}, ..., Y_{im})^{\mathsf{T}}, \quad f(X_{i.}) = (f(X_{i1}), ..., f(X_{im}))^{\mathsf{T}}, \quad f(\tilde{X}_{i.}) = (f(\tilde{X}_{i1}), ..., f(\tilde{X}_{im}))^{\mathsf{T}}.$$

By independence across problems, we have the covariance structures $\mathrm{Var}[f(\tilde{X}_{i.})] = \mathrm{diag}((\tau_j^2)_{j=1}^{m})$ and $\mathrm{Cov}[f(X_{i.}), Y_{i.}] = \mathrm{diag}((\gamma_j)_{j=1}^{m})$ where $\mathrm{diag}(\cdot)$ constructs a diagonal matrix with its arguments on the diagonal. Angelopoulos et al. (2024) then consider a class of estimators indexed by a single weighting parameter $\lambda$ that is applied to the entire vector:

$$\hat{\theta}_\lambda^{\mathrm{PPI}} := \frac{1}{n} \sum_{i=1}^{n} Y_{i.} + \lambda \left( \frac{1}{N} \sum_{i=1}^{N} f(\tilde{X}_{i.}) - \frac{1}{n} \sum_{i=1}^{n} f(X_{i.}) \right).$$

The optimal $\lambda^*$ is chosen by minimizing the trace of an asymptotic covariance $\Sigma^\lambda$. For the mean estimation problem, $\Sigma^\lambda$ simplifies to the covariance of $\hat{\theta}_\lambda^{\mathrm{PPI}}$ and can be exactly calculated in finite-sample setting (without relying on $n, N \to \infty$). Minimizing $\Sigma^\lambda$ over $\lambda$ thus admits a simple closed-from solution:

$$\lambda^* := \frac{n^{-1}\mathrm{Tr}(\mathrm{Cov}[f(X_{i.}), Y_{i.}])}{\frac{n+N}{nN}\mathrm{Tr}(\mathrm{Var}[f(\tilde{X}_{i.})])} = \frac{n^{-1}\sum_{j=1}^{m}\gamma_j}{\frac{n+N}{nN}\sum_{j=1}^{m}\tau_j^2} \tag{19}$$

Going back to our original formulation with multiple problems, what we are doing here is to perform power tuning *across problems* to minimize the sum of MSEs—which equals the trace of the covariance matrix $\mathrm{Tr}(\mathrm{Var}[f(\tilde{X}_{i.})])$ in the PPI++ formulation—over a single (univariate) $\lambda$. Finally, if $n_j$ and $N_j$ vary across $j$, the above PPI++ formulation fails to hold, but the idea to power tune all problems together carries over to our multiple problem setting. The univariate optimal tuning parameter takes the same form as Equation (19):

$$\lambda^* := \frac{\sum_{j=1}^{m} n_j^{-1}\gamma_j}{\sum_{j=1}^{m} \frac{n_j+N_j}{n_j N_j}\tau_j^2}.$$

In fact, this choice of $\lambda^*$ leads to a new estimator, denoted as `UniPT`, that will be further explored in Appendix C.

## A.2. Further Intuitions Behind the Design of `PAS`

### A.2.1. WHY DOES SHARING INFORMATION ACROSS PROBLEMS HELP?

Here we provide an explanation of the fundamental statistical difference between estimating a single mean versus estimating a lot of means ($m \to \infty$) with an eye toward providing intuition of how we can share information across problems. We emphasize that this is only meant for intuition; the empirical Bayes literature, starting with James & Stein (1961), provides a more nuanced picture.

Imagine a simplified setting with only a single problem ($m = 1$, as in the standard PPI setting). We further assume that $N_j = \infty$, so that $\tilde{Z}_1^f \equiv \mathbb{E}_{\eta_1}[f(\tilde{X}_1)]$ and we also assume that $\tilde{\sigma}_1^2$ is known. Moreover, suppose we want to choose between the PT estimator $\hat{\theta}_1^{\text{PT}}$ and the prediction mean $\tilde{Z}_1^f$ by comparing their MSEs in a data-driven way. We know that the PT estimator is unbiased and its MSE is $\tilde{\sigma}_1^2$, but we cannot estimate $\mathbb{E}[(\theta_1 - \tilde{Z}_1^f)^2]$ accurately since we only have a single $\theta_1$ (a single problem). At best, we can use $(\hat{\theta}_1^{\text{PT}} - \tilde{Z}_1^f)^2 - \tilde{\sigma}_1^2$ as an unbiased estimate of this quantity since $\mathbb{E}_{\eta_1}[(\hat{\theta}_1^{\text{PT}} - \tilde{Z}_1^f)^2] - \tilde{\sigma}_1^2 = \mathbb{E}_{\eta_1}[(\theta_1 - \tilde{Z}_1^f)^2]$.

Now suppose we have multiple problems, then we can more precisely learn how good the prediction mean $\tilde{Z}_j^f$ is for estimating $\theta_j$ by repeating the above estimation procedure for each problem and averaging the results. This average turns out to be not only unbiased but also consistent. To wit, as $m \to \infty$, by the law of large numbers,

$$\frac{1}{m} \sum_{j=1}^{m} \left( (\hat{\theta}_j^{\text{PT}} - \tilde{Z}_j^f)^2 - \tilde{\sigma}_j^2 \right) \xrightarrow{\mathbb{P}} \mathbb{E}_{\mathbb{P}_\eta}[(\theta_j - \tilde{Z}_j^f)^2], \tag{20}$$

where we emphasize that the mean squared on the right-hand side also integrates with respect to the meta-distribution $\mathbb{P}_\eta$ that determines the distribution of the $\theta_j$ (cf. Assumption 2.1). This illustrates a mechanism for sharing information across problems: in a first step, we can consistently estimate the average squared difference between the true parameters $\theta_j$ and the prediction means $\tilde{Z}_j^f$ (i.e., how good the ML predictor is on average) by aggregating information from many parallel problems. Then, based on the first step, we can choose whether to use $\tilde{Z}^f$ or $\hat{\theta}^{\text{PT}}$ according to their MSE.

Our actual implementation is more involved but embodies the same fundamental principle: we construct a consistent estimate for the MSE of a family of shrinkage estimators. While other factors such as heterogeneity in sample sizes ($n_j, N_j$) introduce further complexities, the core idea of sharing information here carries over to the overall design and justification of PAS.

### A.2.2. ANALOGY BETWEEN $Z_j^f$ AND $\mathbb{E}[\theta \mid \psi]$

In Equation (16), we informally draw an analogy between the prediction mean $Z_j^f$ in the compound PPI problem and the posterior mean $\mathbb{E}[\theta \mid \psi]$ in the stylized Gaussian model. Our goal here is to provide a heuristic motivation for the one-dimensional parameterized family of weights $\omega_j$ in Equation (15). We emphasize that the ultimate success of this parameterization choice is judged by the empirical results.

Our heuristic motivation building on the stylized example of Section 3 is as follows. We seek to estimate $\theta$ by taking a convex combination $\hat{\theta}^{cl} - \xi$ and a shrinkage target $s$,

$$w(\hat{\theta}^{cl} - \xi) + (1 - w)s,$$

for $w \in [0, 1]$. Our main point is that for several choices of shrinkage target $s$, the best weight $w$ can be written in the form

$$w = \frac{\omega}{\omega + \sigma_\varepsilon^2}, \tag{21}$$

for some $\omega \geq 0$.[16]

- The Bayes estimator with contextual prior in Equation (12) uses $s = \mathbb{E}[\theta \mid \phi]$ and weight $w$ as in (21) with

$$\omega = \mathbb{E}[(\theta - \mathbb{E}[\theta \mid \phi])^2] = \mathbb{E}[\text{Var}[\theta \mid \phi]] = \sigma_{\theta|\phi}^2.$$

- The Bayes estimator without contextual prior uses $s = \mathbb{E}[\theta] = 0$ and weight $w$ as in (21) with

$$\omega = \mathbb{E}[(\theta - \mathbb{E}[\theta])^2] = \text{Var}(\theta) = \sigma_\theta^2.$$

---

[16]At this stage, there is 1-1 mapping between $\omega \geq 0$ and $w \in [0, 1]$. However, for PAS, $\sigma_\varepsilon^2$ corresponds to the variance of the $j$-th power tuning estimator and will be different from problem to problem. Then it will be convenient to parameterize all the problems by the same parameter $\omega \geq 0$.

- Suppose now that we ask for the best convex combination (not necessarily a Bayes predictor) between $\hat{\theta}^{cl} - \xi$ and $s = h(\phi)$ where $h(\cdot)$ is some fixed function. Then, the weight $w$ minimizing MSE can be shown to take the form in (21) with

$$\omega = \mathbb{E}[(\theta - h(\phi))^2] = \mathbb{E}[\mathrm{Var}[\theta \mid \phi]] + \mathbb{E}[(h(\phi) - \mathbb{E}[\theta \mid \phi])^2].$$

The above expression is interesting as it forces us to inflate "$\omega$", i.e., to shrink less toward $h(\phi)$ in a way that depends on how close $h(\phi)$ is to $\mathbb{E}[\theta \mid \phi]$. See Ignatiadis & Wager (2019, last paragraph of Section 3) and Appendix A.2.3 below for further discussion of this point.

The takeaway is that for lots of possible predictors, the optimal weights have the same parameterized form up to the single parameter $\omega$ that varies according to the quality of the predictor. This motivates our one-dimensional family of weights. Once this family has been motivated, we learn $\omega$ in Equation (15) in a way that does not depend on the above analogy at all by minimizing CURE.

### A.2.3. ELABORATION ON PROPOSITION 5.3

Suppose (as in Proposition 5.3) that $n_j$ is the same across all problems, $N_j = \infty$, and that second moments $\rho_j, \tau_j, \sigma_j$ are identical across all problems. Then we could ask: *what is the best convex combination between $\hat{\theta}_j^{PT}$ and $\tilde{Z}_j^f$ in the following sense:*

$$\omega_j^* \in \underset{\omega_j \geq 0}{\arg\min} \, \mathbb{E}_{\mathbb{P}_\eta}\left[\{\theta_j - (\omega_j \hat{\theta}_j^{\mathrm{PT}} + (1 - \omega_j)\tilde{Z}_j^f)\}^2\right].$$

By direct calculation (note that the right-hand side is a convex quadratic in $\omega_j$) we find that:

$$\omega_j^* = \frac{\mathbb{E}_{\mathbb{P}_\eta}[(\theta_j - \tilde{Z}_j^f)^2]}{\mathbb{E}_{\mathbb{P}_\eta}[(\theta_j - \tilde{Z}_j^f)^2] + \tilde{\sigma}^2}.$$

This implies the following intuitive result: the larger the MSE $\mathbb{E}_{\mathbb{P}_\eta}[(\theta_j - \tilde{Z}_j^f)^2]$, the less weight we should assign to $\tilde{Z}_j^f$. If we evaluate the MSE at this optimal $\omega_j^*$, we recover precisely the upper bound of Proposition 5.3.

## B. The Correlation-Aware Unbiased Risk Estimate

**Theorem B.1.** *Let $X, Y$ be two random variables satisfying $\mathbb{E}_\theta[X] = \theta$, $\mathrm{Var}_\theta[X] = \sigma^2$, $\mathrm{Cov}_\theta[X, Y] = \gamma$, and the second moment of $Y$ exists.*[17] *Consider estimating $\theta$ with the shrinkage estimator $\hat{\theta}_c = cX + (1 - c)Y$ with $c \in [0, 1]$. Assuming that $\sigma^2$ and $\gamma$ are known, the following estimator*

$$\mathrm{CURE}(\hat{\theta}_c) := (2c - 1)\sigma^2 + 2(1 - c)\gamma + \{(1 - c)(X - Y)\}^2, \tag{22}$$

*defined as the $\underline{C}$orrelation-aware $\underline{U}$nbiased $\underline{R}$isk $\underline{E}$stimate, is an unbiased estimator for the risk of $\hat{\theta}_c$ under quadratic loss. That is, letting $R(\hat{\theta}_c, \theta) := \mathbb{E}_\theta[(\hat{\theta}_c - \theta)^2]$, it holds that:*

$$\mathbb{E}_\theta\left[\mathrm{CURE}(\hat{\theta}_c)\right] = R(\hat{\theta}_c, \theta).$$

*Proof.* First, expand the risk:

$$
\begin{aligned}
R(\hat{\theta}_c, \theta) = \mathbb{E}_\theta[(\hat{\theta}_c - \theta)^2] &= \mathbb{E}_\theta[(cX + (1 - c)Y - \theta)^2] \\
&= \mathrm{Var}_\theta[cX + (1 - c)Y] + (\mathbb{E}_\theta[cX + (1 - c)Y] - \theta)^2 \\
&= c^2\sigma^2 + (1 - c)^2\mathrm{Var}_\theta[Y] + 2c(1 - c)\gamma + [(1 - c)(\mathbb{E}_\theta[Y] - \theta)]^2.
\end{aligned}
$$

---

[17]We redefine certain variables for generality of this result beyond the setting in Assumption 2.2. In this theorem, $\theta$ plays the role of $\eta$ in the main text, i.e. all the other parameters are deterministic given $\theta$.

Then, taking the expectation of $\text{CURE}(\hat{\theta}_c)$:

$$\mathbb{E}_\theta\left[\text{CURE}(\hat{\theta}_c)\right] = \underbrace{(2c-1)\sigma^2 + 2(1-c)\gamma}_{\mathbb{I}} + \mathbb{E}_\theta\left[\{(1-c)(X-Y)\}^2\right], \tag{23}$$

where the last term is

$$\begin{aligned}
\mathbb{E}_\theta\left[\{(1-c)(X-Y)\}^2\right] &= (1-c)^2\left[(\mathbb{E}_\theta[X-Y])^2 + \text{Var}_\theta[X-Y]\right] \\
&= (1-c)^2\left[(\mathbb{E}_\theta[Y]-\theta)^2 + \sigma^2 + \text{Var}_\theta[Y] - 2\gamma\right] \\
&= [(1-c)(\mathbb{E}_\theta[Y]-\theta)]^2 + \underbrace{(1-c)^2(\sigma^2 + \text{Var}_\theta[Y] - 2\gamma)}_{\mathbb{III}}.
\end{aligned}$$

With a little algebra, we observe

$$\begin{aligned}
\mathbb{I} + \mathbb{III} &= (2c-1)\sigma^2 + 2(1-c)\gamma + (1-c)^2(\sigma^2 + \text{Var}_\theta[Y] - 2\gamma) \\
&= c^2\sigma^2 + (1-c)^2\text{Var}_\theta[Y] + 2c(1-c)\gamma.
\end{aligned}$$

Thus, a term-by-term matching confirms $\mathbb{E}_\theta[\text{CURE}(\hat{\theta}_c)] = R(\hat{\theta}_c, \theta)$. □

**Remark B.2** (Connection to SURE). Stein's Unbiased Risk Estimate (SURE) was proposed in Charles Stein's seminal work (1981) to study the quadratic risk in Gaussian sequence models. As a simple special case of SURE, let $Z \sim \mathcal{N}(\theta, \sigma^2)$ and let $h : \mathbb{R} \to \mathbb{R}$ be an absolutely continuous function and $\mathbb{E}_\theta[|h'(Z)|] < \infty$, then SURE is defined as

$$\text{SURE}(h) := (h(Z) - Z)^2 + 2\sigma^2 h'(Z) - \sigma^2,$$

with the property that $\mathbb{E}_\theta[\text{SURE}(h)] = R(h(Z), \theta) = \mathbb{E}_\theta[(h(Z) - \theta)^2]$. A proof of this argument relies on Stein's lemma, an identity specific to Gaussian random variables (Stein, 1981). Now consider the specific linear shrinkage estimator $h_c(Z) := cZ + (1-c)Y$, with $c \in [0,1]$ and $Y \in \mathbb{R}$ being fixed (that is, $Y$ is a constant, or $Y$ is independent of $Z$ and we condition on $Y$). Then SURE takes the following form:

$$\begin{aligned}
\text{SURE}(h_c) &= (h_c(Z) - Z)^2 + 2\sigma^2 h_c'(Z) - \sigma^2 \\
&= (cZ + (1-c)Y - Z)^2 + 2c\sigma^2 - \sigma^2 \\
&= [(1-c)(Y - Z)]^2 + (2c-1)\sigma^2 \\
&\overset{(\star)}{=} \text{CURE}(h_c(Z)),
\end{aligned}$$

where in $(\star)$ we used the fact that in this case (with $Y$ fixed or $Y$ independent of $Z$), it holds that $\gamma = 0$ so that the definition of CURE in (22) simplifies. This explains how CURE defined in B.1 is connected to SURE.

We make one last remark: The derivation of SURE itself requires Gaussianity. However, for linear shrinkage rules as $h_c(Z)$, SURE only depends on the first two moments of the distribution of $Z$ and thus is an unbiased estimator of quadratic risk under substantial generality as long as $\mathbb{E}_\theta[Z] = \theta$ and $\text{Var}_\theta[Z] = \sigma^2$. This remark has been made by previous authors, e.g., Kou & Yang (2017); Ignatiadis & Wager (2019) and is important for the assumption-lean validity of PAS.

## C. Details on PAS with Unknown Second Moments

In this appendix we explain how to apply PAS when second moments are unknown. In Appendix C.1 we describe sample-based estimators of the second moments. We also develop UniPT (Appendix C.2) and UniPAS (Appendix C.3), two new estimators that extend our framework to scenarios where second moments are unknown and must be estimated from data. We derive these methods and their theoretical guarantees within the PAS asymptotic regime, where the number of problems $m \to \infty$ while individual sample sizes $n_j, N_j$ remain bounded. Specifically, UniPT introduces a global power-tuning strategy, and UniPAS builds upon it to perform adaptive shrinkage, both without requiring knowledge of true second-moment parameters.

**Notations.** Throughout this section and the proof details in Appendix F, we use $\xrightarrow{P}$ for convergence in probability and $\xrightarrow{L^p}$ for $L^p$ convergence of random variables. We use the "Little-o" notation $o(1)$ for any term that vanishes to zero as $m \to \infty$. Similarly, for a sequence of random variables $X_m$, we use $X_m = o_P(1)$ if $X_m$ converges to zero in probability, and $X_m = O_P(1)$ if $X_m$ is bounded in probability. All stochastic order relations are understood to hold as $m \to \infty$.

## C.1. Sample-based Estimators for $\sigma_j^2, \tau_j^2, \gamma_j$

We first write down the expressions of the unbiased sample-based estimators for $\sigma_j^2, \tau_j^2$ and $\gamma_j$, assuming that $n_j \geq 2$.

$$\hat{\sigma}_j^2 := \frac{1}{n_j - 1} \sum_{i=1}^{n_j} (Y_{ij} - \bar{Y}_j)^2, \quad \hat{\tau}_j^2 := \frac{1}{n_j + N_j - 1} \left( \sum_{i=1}^{n_j} (f(X_{ij}) - \bar{Z}_j^{N+n})^2 + \sum_{i=1}^{N_j} (f(\tilde{X}_{ij}) - \bar{Z}_j^{N+n})^2 \right),$$

$$\hat{\gamma}_j := \frac{1}{n_j - 1} \sum_{i=1}^{n_j} (Y_{ij} - \bar{Y}_j)(f(X_{ij}) - \bar{Z}_j^{N+n}), \quad \text{where} \quad \bar{Z}_j^{N+n} := \frac{1}{n_j + N_j} \left( \sum_{i=1}^{n_j} f(X_{ij}) + \sum_{i=1}^{N_j} f(\tilde{X}_{ij}) \right).$$

These sample-based estimators serve a dual role. Firstly, they are utilized in the practical implementation of the PT and `PAS` estimators for our numerical experiments on real-world datasets, where true second moments are unavailable. Secondly, they form the basis for defining the `UniPT` and `UniPAS` estimators, which we introduce subsequently.

## C.2. UniPT: Power-tuning Across Problems with Estimated Second Moments

Under this new setup, the first step is to derive a new variant of the PT estimator without the known second moments. It turns out that performing power-tuning across problems using the same $\lambda$ for all problems (which we show in Appendix A.1 to be closely related to the multivariate estimation problem in PPI++) leads to a promising alternative.

**Definition C.1** (Univariate Power Tuning (`UniPT`)). We consider a family of estimators for each problem $j \in [m]$:

$$\hat{\theta}_{j,\lambda} = \bar{Y}_j + \lambda(\bar{Z}_j^f - \tilde{Z}_j^f),$$

where all problems are controlled by a single, global power-tuning parameter $\lambda \in \mathbb{R}$. Our goal is to find the $\lambda$ that minimizes the sum of variances across all problems, $\sum_{j=1}^m \mathrm{Var}_{\eta_j}[\hat{\theta}_{j,\lambda}]$. The variance for each problem $j$ is:

$$\mathrm{Var}_{\eta_j}[\hat{\theta}_{j,\lambda}] = \frac{\sigma_j^2}{n_j} + \lambda^2 \left( \frac{1}{n_j} + \frac{1}{N_j} \right) \tau_j^2 - \frac{2\lambda}{n_j} \gamma_j,$$

where $\sigma_j^2 = \mathrm{Var}_{\eta_j}[Y_{ij}]$, $\tau_j^2 = \mathrm{Var}_{\eta_j}[f(X_{ij})]$, and $\gamma_j = \mathrm{Cov}_{\eta_j}[Y_{ij}, f(X_{ij})]$. Minimizing $\sum_{j=1}^m \mathrm{Var}_{\eta_j}[\hat{\theta}_{j,\lambda}]$ with respect to $\lambda$ yields the theoretically optimal global parameter for the given set of $m$ problems:

$$\lambda_m^* := \frac{\sum_{j=1}^m n_j^{-1} \gamma_j}{\sum_{j=1}^m \left( \frac{1}{n_j} + \frac{1}{N_j} \right) \tau_j^2} = \frac{\sum_{j=1}^m n_j^{-1} \gamma_j}{\sum_{j=1}^m \frac{n_j + N_j}{n_j N_j} \tau_j^2}.$$

In practice, since $\gamma_j$ and $\tau_j^2$ are now assumed unknown, we replace them with their sample-based unbiased estimators $\hat{\gamma}_j$ and $\hat{\tau}_j^2$ (computed in Appendix C.1) to obtain the estimated global power-tuning parameter:

$$\hat{\lambda} := \frac{\sum_{j=1}^m n_j^{-1} \hat{\gamma}_j}{\sum_{j=1}^m \frac{n_j + N_j}{n_j N_j} \hat{\tau}_j^2}.$$

We clip $\hat{\lambda}$ to the interval $[0, 1]$. Let this be $\hat{\lambda}_{\text{clip}} = \text{clip}(\hat{\lambda}, [0, 1])$. Similarly, we denote the clipped version of $\lambda_m^*$ as $\lambda_{\text{clip},m}^* := \text{clip}(\lambda_m^*, [0, 1])$. The `UniPT` estimator for problem $j$ is then:

$$\hat{\theta}_j^{\text{UPT}} = \bar{Y}_j + \hat{\lambda}_{\text{clip}}(\bar{Z}_j^f - \tilde{Z}_j^f).$$

The `UniPT` estimator, by using a single data-driven $\hat{\lambda}$, offers a practical way to perform power tuning when per-problem moments are unknown. This approach is justified by the following theoretical results, which hold as $m \to \infty$.

**Proposition C.2** (Asymptotic Consistency of Clipped Global Tuning Parameter). *Assume that:*

1. *Sample sizes $n_j, N_j$ are bounded ($2 \leq n_{\min} \leq n_j \leq n_{\max} < \infty$, $1 \leq N_{\min} \leq N_j \leq N_{\max} < \infty$).*

2. *$\mathbb{E}_{\mathbb{P}_\eta}[\mathrm{Var}_{\eta_j}[\hat{\gamma}_j]] < \infty$ and $\mathbb{E}_{\mathbb{P}_\eta}[\mathrm{Var}_{\eta_j}[\hat{\tau}_j^2]] < \infty$.*

3. $\mathbb{E}_{\mathbb{P}_\eta}[\gamma_j^2] < \infty$, $\mathbb{E}_{\mathbb{P}_\eta}[(\tau_j^2)^2] < \infty$, $\mathbb{E}_{\mathbb{P}_\eta}[(\sigma_j^2)^2] < \infty$.[18]

4. *(Denominator bounded away from 0) there exists some $\varepsilon > 0$ such that*

$$\lim_{m \to \infty} \mathbb{P}\left[\left|\frac{1}{m}\sum_{j=1}^m \frac{n_j + N_j}{n_j N_j}\tau_j^2\right| > \varepsilon\right] = 1$$

*Then, the clipped estimated global tuning parameter $\hat{\lambda}_{\text{clip}}$ converges in $L^2$ to the clipped theoretical optimal global tuning parameter $\lambda^*_{\text{clip},m}$:*

$$\hat{\lambda}_{\text{clip}} - \lambda^*_{\text{clip},m} \xrightarrow{L^2} 0 \quad as \ m \to \infty.$$

The proof is deferred to Appendix F.5.

This consistency ensures that $\hat{\lambda}_{\text{clip}}$ effectively targets the best single power-tuning parameter for the collection of $m$ problems. Building on this, we can state a result regarding the asymptotic variance of the `UniPT` estimator.

**Theorem C.3** (Asymptotic Variance Optimality of UniPT). *Under the assumptions of Proposition C.2, the sum of variances of the `UniPT` estimators, $\sum_{j=1}^m \text{Var}_{\eta_j}[\hat{\theta}_j^{\text{UPT}}]$, asymptotically achieves the minimum possible sum of variances within the class of estimators $\mathcal{C} = \{(\hat{\theta}_{j,\lambda})_{j=1}^m \mid \hat{\theta}_{j,\lambda} = \bar{Y}_j + \lambda(\bar{Z}_j^f - \bar{Z}_j'^f), \lambda \in [0,1]\}$, in the sense that:*

$$\frac{1}{m}\sum_{j=1}^m \text{Var}_{\eta_j}[\hat{\theta}_j^{\text{UPT}}] - \min_{\lambda' \in [0,1]}\frac{1}{m}\sum_{j=1}^m \text{Var}_{\eta_j}[\hat{\theta}_{j,\lambda'}] \xrightarrow{P} 0 \quad as \ m \to \infty,$$

*where the left-hand side is still a random variable with randomness from drawing $\eta_j \overset{iid}{\sim} \mathbb{P}_\eta$.*

The proof is deferred to Appendix F.6.

### C.3. UniPAS: CURE and Adaptive Shrinkage with Estimated Second Moments

Once we obtain the `UniPT` estimator, which is asymptotically optimal within a class of unbiased estimators, our next goal is to imitate the steps in Section 4.2 to apply shrinkage across problems. To do so, we must first revisit the formulation of CURE and see how it depends on the now unknown second-moment parameters. By definition:

$$\text{CURE}\left(\hat{\boldsymbol{\theta}}_\omega^{\text{PAS}}\right) := \frac{1}{m}\sum_{j=1}^m \left[(2\omega_j - 1)\tilde{\sigma}_j^2 + 2(1-\omega_j)\tilde{\gamma}_j + (1-\omega_j)^2(\hat{\theta}_{j,\omega_j}^{\text{PT}} - \tilde{Z}_j^f)^2\right], \quad \omega_j = \frac{\omega}{\omega + \tilde{\sigma}_j^2}. \quad (24)$$

So there are three places where CURE makes use of $\tilde{\sigma}_j^2, \tilde{\gamma}_j$, which then depend on second moments of the data generating process: (1) in the definition of PT estimator, (2) in the definition of CURE itself and (3) in determining the localized shrinkage level $\omega_j$. We thus make the following modifications to CURE using the sample-based estimators.

1. We first replace the PT estimator (shrinkage source) to the `UniPT` estimator. Additionally, $\tilde{\sigma}_j^2$ is now the variance of $\hat{\theta}_{j,\lambda^*_{\text{clip},m}}$ (as defined in Definition C.1) and $\tilde{\tau}_j$ is its covariance with $\tilde{Z}_j^f$. We can explicitly write them down as

$$\tilde{\sigma}_j^2 := \frac{\sigma_j^2}{n_j} + \frac{N_j + n_j}{N_j n_j}(\lambda^*_{\text{clip},m})^2\tau_j^2 - \frac{2}{n_j}\lambda^*_{\text{clip},m}\gamma_j, \quad \tilde{\gamma}_j := \lambda^*_{\text{clip},m}\frac{\tau_j^2}{N_j}.$$

2. For $\tilde{\sigma}_j^2$ and $\tilde{\gamma}_j$ in the definition of CURE, we replace them directly with the sample-based estimators

$$\dot{\sigma}_j^2 := \frac{\hat{\sigma}_j^2}{n_j} + \frac{N_j + n_j}{N_j n_j}\hat{\lambda}_{\text{clip}}^2\hat{\tau}_j^2 - \frac{2}{n_j}\hat{\lambda}_{\text{clip}}\hat{\gamma}_j, \quad \dot{\gamma}_j := \hat{\lambda}_{\text{clip}}\frac{\hat{\tau}_j^2}{N_j}$$

where $\hat{\sigma}_j^2, \hat{\tau}_j^2, \hat{\gamma}_j$ are defined in Appendix C.1.

---

[18]Note that this condition is implied by the finite fourth-moment assumption in Proposition 5.1.

3. For $\tilde{\sigma}_j^2$ in the definition of $\omega_j$, we replace it with the following averaging estimators:

$$\check{\sigma}_j^2 := \frac{\bar{\sigma}^2}{n_j} + \frac{N_j + n_j}{N_j n_j} \hat{\lambda}_{\text{clip}}^2 \bar{\tau}^2 - \frac{2}{n_j} \hat{\lambda}_{\text{clip}} \bar{\gamma}, \quad \text{where} \tag{25}$$

$$\bar{\sigma}^2 := \frac{1}{m} \sum_{j=1}^m \hat{\sigma}_j^2, \quad \bar{\tau}^2 := \frac{1}{m} \sum_{j=1}^m \hat{\tau}_j^2, \quad \bar{\gamma} := \frac{1}{m} \sum_{j=1}^m \hat{\gamma}_j$$

are the average sample co-(variances) across all $m$ problems.[19]

With these modifications, we define a new class of shrinkage estimators based on $\hat{\theta}_j^{\text{UPT}}$. For any global shrinkage parameter $\omega \geq 0$, the problem-specific shrinkage weight is now defined as:

$$\hat{\omega}_j := \hat{\omega}_j(\omega) = \frac{\omega}{\omega + \check{\sigma}_j^2}.$$

The corresponding family of shrinkage estimators for problem $j$ is:

$$\hat{\theta}_{j,\omega}^{\text{UPAS}} := \hat{\omega}_j \hat{\theta}_j^{\text{UPT}} + (1 - \hat{\omega}_j) \tilde{Z}_j^f.$$

We then define the modified CURE, denoted $\widehat{\text{CURE}}$, by taking into account all the changes above.[20]

$$\widehat{\text{CURE}} \left( \hat{\boldsymbol{\theta}}_\omega^{\text{UPAS}} \right) := \frac{1}{m} \sum_{j=1}^m \left[ (2\hat{\omega}_j - 1)\check{\sigma}_j^2 + 2(1 - \hat{\omega}_j)\dot{\gamma}_j + (1 - \hat{\omega}_j)^2 \left( \hat{\theta}_j^{\text{UPT}} - \tilde{Z}_j^f \right)^2 \right]. \tag{26}$$

Finally, the `UniPAS` estimator is obtained by selecting the $\omega$ that minimizes this $\widehat{\text{CURE}}$:

**Definition C.4** (Univariate Prediction-Powered Adaptive Shrinkage (`UniPAS`)). The `UniPAS` estimator for problem $j$ is $\hat{\theta}_j^{\text{UPAS}} := \hat{\theta}_{j,\hat{\omega}}^{\text{UPAS}}$, where

$$\hat{\omega} := \arg\min_{\omega \geq 0} \widehat{\text{CURE}} \left( \hat{\boldsymbol{\theta}}_\omega^{\text{UPAS}} \right).$$

This `UniPAS` estimator is fully data-driven and does not rely on knowledge of the true second-moment parameters. The pseudo-code for the full `UniPAS` algorithm is given below.

The fully data-driven construction of `UniPAS` is supported by the following theoretical guarantee:

**Proposition C.5** (Asymptotic Consistency of $\widehat{\text{CURE}}$ for `UniPAS`). *On top of the assumptions in Proposition C.2 and Assumption 2.2, if we further require that $\inf_{j \in [m], m \in \mathbb{N}} \mathring{\sigma}_{j,m}^2 \geq \delta > 0$ for some fixed $\delta$, where*

$$\mathring{\sigma}_j^2 \equiv \mathring{\sigma}_{j,m}^2 := \frac{\mu_{\sigma^2}}{n_j} + \frac{N_j + n_j}{N_j n_j} (\lambda_{\text{clip},m}^*)^2 \mu_{\tau^2} - \frac{2}{n_j} \lambda_{\text{clip},m}^* \mu_\gamma \tag{27}$$

*with $\mu_{\sigma^2} := \mathbb{E}_{\mathbb{P}_\eta}[\sigma_j^2]$, $\mu_{\tau^2} := \mathbb{E}_{\mathbb{P}_\eta}[\tau_j^2]$, $\mu_\gamma := \mathbb{E}_{\mathbb{P}_\eta}[\gamma_j]$. Then, $\widehat{\text{CURE}} \left( \hat{\boldsymbol{\theta}}_\omega^{\text{UPAS}} \right)$ is an asymptotically consistent estimator for the true loss $\ell_m(\hat{\boldsymbol{\theta}}_\omega^{\text{UPAS}}, \boldsymbol{\theta})$ of the `UniPAS` estimator (which uses weights $\hat{\omega}_j = \omega/(\omega + \check{\sigma}_j^2)$), in the sense that:*

$$\mathbb{E}_{\mathbb{P}_\eta} \left[ \sup_{\omega \geq 0} |\widehat{\text{CURE}} \left( \hat{\boldsymbol{\theta}}_\omega^{\text{UPAS}} \right) - \ell_m(\hat{\boldsymbol{\theta}}_\omega^{\text{UPAS}}, \boldsymbol{\theta})| \right] \xrightarrow{m \to \infty} 0.$$

We defer the proof to Appendix F.7.

This consistency result ensures that minimizing $\widehat{\text{CURE}}$ is asymptotically equivalent to minimizing the true MSE of the `UniPAS` estimator. This leads to the following optimality guarantee:

---

[19]The rationale is as follows: In defining $\omega_j$ we pretend that $\sigma_j^2$, $\tau_j^2$ and $\gamma_j$ are the same across all problems and so can be estimated consistently. We emphasize that our theoretical results do not require that these second moments be identical (i.e., it is only a working modeling assumption).

[20]We use the hat notation for $\widehat{\text{CURE}}$ to make it explicit that $\widehat{\text{CURE}}$ is not an unbiased estimator of risk in finite samples. However, we will show that it is a consistent estimator of risk asymptotically as $m \to \infty$.

---

**Algorithm 2** `UniPAS`

---

**Require:** $(X_{ij}, Y_{ij})_{i=1}^{n_j}, (\tilde{X}_{ij})_{i=1}^{N_j}$ for $j \in [m]$, predictive model $f$

1: **for** $j = 1$ to $m$ **do**
2:    ▷ Step 1: Apply predictor (Eq. (1)) to get aggregated statistics and sample-based estimators for second moments
3:    $\bar{Y}_j, \bar{Z}_j^f, \tilde{Z}_j^f = \texttt{get\_means}((X_{ij}, Y_{ij})_{i=1}^{n_j}, (\tilde{X}_{ij})_{i=1}^{N_j}, f)$
4:    $\hat{\sigma}_j^2, \hat{\gamma}_j, \hat{\tau}_j^2 = \texttt{get\_sample\_variances}((X_{ij}, Y_{ij})_{i=1}^{n_j}, (\tilde{X}_{ij})_{i=1}^{N_j}, f)$
5:    ▷ Step 2: Univariate power tuning (Appendix C.2)
6:    $\hat{\lambda}_{\text{clip}} = \text{clip}\left( \dfrac{\sum_{j=1}^m n_j^{-1} \hat{\gamma}_j}{\sum_{j=1}^m \frac{N_j + n_j}{N_j n_j} \hat{\tau}_j^2}, [0, 1] \right)$
7:    $\hat{\theta}_j^{\text{UPT}} = \bar{Y}_j + \hat{\lambda}_{\text{clip}}(\tilde{Z}_j^f - \bar{Z}_j^f)$
8:    ▷ Step 3: Construct sample-based estimators of $\text{Var}[\hat{\theta}_j^{\text{UPT}}]$ and $\text{Cov}[\hat{\theta}_j^{\text{UPT}}, \tilde{Z}_j^f]$
9:    $\dot{\sigma}_j^2 = \dfrac{\hat{\sigma}_j^2}{n_j} + \dfrac{N_j + n_j}{N_j n_j} \hat{\lambda}_{\text{clip}}^2 \hat{\tau}_j^2 - \dfrac{2}{n_j} \hat{\lambda}_{\text{clip}} \hat{\gamma}_j$
10:    $\dot{\gamma}_j = \hat{\lambda} \dfrac{\hat{\tau}_j^2}{N_j}$
11: **end for**
12: ▷ Step 4: Adaptive shrinkage
13: $\bar{\sigma}^2 = \frac{1}{m} \sum_{j=1}^m \hat{\sigma}_j^2, \ \bar{\tau}^2 = \frac{1}{m} \sum_{j=1}^m \hat{\tau}_j^2, \ \bar{\gamma} = \frac{1}{m} \sum_{j=1}^m \hat{\gamma}_j$
14: **for** $j = 1$ to $m$ **do**
15:    ▷ Calculate the averaging variance estimator in Equation (25)
16:    $\check{\sigma}_j^2 := \dfrac{\bar{\sigma}^2}{n_j} + \dfrac{N_j + n_j}{N_j n_j} \hat{\lambda}_{\text{clip}}^2 \bar{\tau}^2 - \dfrac{2}{n_j} \hat{\lambda}_{\text{clip}} \bar{\gamma}$
17: **end for**
18: ▷ Now for each choice of $\omega \geq 0$, the shrinkage coefficient is determined by $\frac{\omega}{\omega + \check{\sigma}^2}$
19: ▷ The CURE calculation in Equation (26), which still depends on the sample-based estimators $\dot{\sigma}_j^2, \dot{\gamma}_j$
20: $\hat{\omega} = \texttt{get\_shrink\_param}((\hat{\theta}_j^{\text{UPT}})_{j=1}^m, (\tilde{Z}_j^f)_{j=1}^m, (\check{\sigma}_j^2)_{j=1}^m, (\dot{\sigma}_j^2)_{j=1}^m, (\dot{\gamma}_j)_{j=1}^m)$
21: **for** $j = 1$ to $m$ **do**
22:    $\hat{\theta}_j^{\text{UPAS}} = \hat{\omega} \hat{\theta}_j^{\text{UPT}} + (1 - \hat{\omega}) \tilde{Z}_j^f$
23: **end for**
24: **return** $\{\hat{\theta}_j^{\text{UPAS}}\}_{j=1}^m$

---

**Theorem C.6** (Asymptotic Bayes Risk Optimality of `UniPAS`). *Under the assumptions of Theorem C.5, the `UniPAS` estimator $\hat{\boldsymbol{\theta}}^{\text{UPAS}} = \hat{\boldsymbol{\theta}}_{\hat{\omega}}^{\text{UPAS}}$ satisfies:*

$$\mathcal{B}_m^{\mathbb{P}_\eta}\left( \hat{\boldsymbol{\theta}}^{\text{UPAS}} \right) \leq \inf_{\omega \geq 0} \left\{ \mathcal{B}_m^{\mathbb{P}_\eta}\left( \hat{\boldsymbol{\theta}}_\omega^{\text{UPAS}} \right) \right\} + o(1) \quad as \ \ m \to \infty,$$

*and so*

$$\mathcal{B}_m^{\mathbb{P}_\eta}\left( \hat{\boldsymbol{\theta}}^{\text{UPAS}} \right) \leq \min \left\{ \mathcal{B}_m^{\mathbb{P}_\eta}\left( \tilde{\boldsymbol{Z}}^f \right), \mathcal{B}_m^{\mathbb{P}_\eta}\left( \hat{\boldsymbol{\theta}}^{\text{UPT}} \right) \right\} + o(1) \quad as \ \ m \to \infty.$$

The proof of Theorem C.6, which follows directly from Proposition C.5, mirrors the argument in in Appendix F.3 used to derive Theorem 5.2 from Proposition 5.1.

We note that the conclusion here is slightly weaker than that of Theorem 5.2 for `PAS`. Theorem 5.2 guarantees that `PAS` asymptotically has risk less or equal to that of PT with optimal per-problem choice of the power tuning parameter. By contrast, Theorem C.6 along with Theorem C.3 shows that `UniPAS` always has risk less or equal to that of power tuning that uses the same power tuning parameter for all problems. The main upshot of Theorem C.6 is that `UniPAS` does not require knowledge of second moments.

## D. Other Baseline Shrinkage Estimators

### D.1. "Shrink-classical" (Shrinkage) Baseline

The "shrink-classical" estimator applies shrinkage directly to the classical estimator $\bar{Y}_j$, using the prediction mean $\tilde{Z}_j^f$ as a shrinkage target *without* first applying power-tuned PPI. We include this baseline to isolate the benefits of power tuning

from the `PAS` estimator as an ablation study.

**Formulation.** The "shrink-classical" estimator for problem $j$ takes the form:

$$\hat{\theta}_{j,\omega}^{\text{Shrink}} := \omega_j \bar{Y}_j + (1 - \omega_j)\tilde{Z}_j^f,$$

$$\text{where} \quad \omega_j := \omega/(\omega + \tilde{\sigma}_j^2), \quad \tilde{\sigma}_j^2 := \text{Var}_{\eta_j}[\bar{Y}_j] = \sigma_j^2/n_j.$$

Here $\omega \geq 0$ is a global shrinkage parameter analogous to Section 4.2. The key difference from `PAS` is that we shrink the classical estimator $\bar{Y}_j$ (which is independent of $\tilde{Z}_j^f$) rather than the power-tuned estimator $\hat{\theta}_j^{\text{PT}}$ (which is correlated with $\tilde{Z}_j^f$).

**Optimizing $\omega$ via CURE.** Since $\bar{Y}_j$ and $\tilde{Z}_j^f$ are independent, Theorem 4.1 simplifies. Let $\tilde{\gamma}_j = \text{Cov}[\bar{Y}_j, \tilde{Z}_j^f] = 0$ and $\tilde{\sigma}_j^2 = \sigma_j^2/n_j$. CURE simplifies as:

$$\text{CURE}(\hat{\theta}_{j,\omega}^{\text{Shrink}}) = (2\omega_j - 1)\tilde{\sigma}_j^2 + \left[(1 - \omega_j)(\bar{Y}_j - \tilde{Z}_j^f)\right]^2.$$

This follows from Theorem 4.1 by setting $\tilde{\gamma}_j = 0$. The global shrinkage parameter $\omega$ is selected by minimizing CURE across all $m$ problems.

$$\hat{\theta}_j^{\text{Shrink}} := \hat{\omega}_j \bar{Y}_j + (1 - \hat{\omega}_j)\tilde{Z}_j^f, \quad \hat{\omega}_j = \hat{\omega}/(\hat{\omega} + \tilde{\sigma}_j^2),$$

$$\text{where} \quad \hat{\omega} \in \underset{\omega \geq 0}{\arg\min} \frac{1}{m} \sum_{j=1}^m \text{CURE}(\hat{\theta}_{j,\omega}^{\text{Shrink}}). \tag{28}$$

The optimal $\hat{\omega}$ does not admit a closed-form expression, but we can compute it numerically by grid search. Below we provide the pseudo-code for implementing the "shrink-classical" estimator.

---

**Algorithm 3** "Shrink-classical" Estimator

---

**Require:** $\{(X_{ij}, Y_{ij})_{i=1}^{n_j}\}, \{\tilde{X}_{ij}\}_{i=1}^{N_j}$ for $j \in [m]$, variance parameters $\{\sigma_j^2\}_{j=1}^m$, predictive model $f$
1: **for** $j = 1$ to $m$ **do**
2: $\quad \bar{Y}_j, \tilde{Z}_j^f = \textbf{get\_means}((X_{ij}, Y_{ij})_{i=1}^{n_j}, (\tilde{X}_{ij})_{i=1}^{N_j}, f)$
3: $\quad \tilde{\sigma}_j^2 \leftarrow \sigma_j^2/n_j \quad \triangleright$ variance of $\bar{Y}_j$
4: **end for**
5: $\hat{\omega} = \textbf{get\_shrink\_param}((\bar{Y}_j)_{j=1}^m, (\tilde{Z}_j^f)_{j=1}^m, (\tilde{\sigma}_j^2)_{j=1}^m) \triangleright$ use Eq. (28)
6: **for** $j = 1$ to $m$ **do**
7: $\quad \hat{\omega}_j = \hat{\omega}/(\hat{\omega} + \tilde{\sigma}_j^2)$
8: $\quad \hat{\theta}_j^{\text{Shrink}} = \hat{\omega}_j \bar{Y}_j + (1 - \hat{\omega}_j) \tilde{Z}_j^f$
9: **end for**
10: **return** $\{\hat{\theta}_j^{\text{Shrink}}\}_{j=1}^m$

---

### D.2. "Shrink-average" Baseline

The "shrink-average" estimator represents an alternative, perhaps more classical, shrinkage approach that attempts to further improve upon the unbiased PT estimators. While `PAS` reuses the prediction means on unlabeled data as shrinkage targets, here we consider shrinking the PT estimators across all problems to a shared location, namely their group mean

$$\bar{\theta}^{\text{PT}} := \frac{1}{m} \sum_{j=1}^m \hat{\theta}_j^{\text{PT}}.$$

**Formulation.** The "shrink-average" estimator for problem $j$ takes the form:

$$\hat{\theta}_{j,\omega}^{\text{Avg}} := \omega_j \hat{\theta}_j^{\text{PT}} + (1 - \omega_j)\bar{\theta}^{\text{PT}},$$

$$\text{where} \quad \omega_j := \omega/(\omega + \tilde{\sigma}_j^2), \quad \tilde{\sigma}_j^2 := \text{Var}_{\eta_j}[\hat{\theta}_j^{\text{PT}}].$$

---

**Algorithm 4** "Shrink-average" Estimator

---

**Require:** $(X_{ij}, Y_{ij})_{i=1}^{n_j}, (\tilde{X}_{ij})_{i=1}^{N_j}, \gamma_j, \tau_j, \sigma_j$ for $j \in [m]$, predictive model $f$
1: **for** $j = 1$ to $m$ **do**
2:     ▷ Step 1: Apply predictor (Eq. (1))
3:     $\bar{Y}_j, \bar{Z}_j^f, \tilde{Z}_j^f = \mathbf{get\_means}((X_{ij}, Y_{ij})_{i=1}^{n_j}, (\tilde{X}_{ij})_{i=1}^{N_j}, f)$
4:     ▷ Step 2: Power tuning (Eq. (13))
5:     $\lambda_j^* = \mathbf{get\_pt\_param}(\gamma_j, \tau_j, n_j, N_j)$
6:     $\hat{\theta}_j^{\mathrm{PT}} = \bar{Y}_j + \lambda_j^*(\tilde{Z}_j^f - \bar{Z}_j^f)$
7:     $\tilde{\sigma}_j^2 = \mathbf{get\_pt\_var}(\hat{\theta}_j^{\mathrm{PT}})$ ▷ (Eq. (14))
8: **end for**
9: $\bar{\theta}^{\mathrm{PT}} = m^{-1} \sum_{j=1}^m \hat{\theta}_j^{\mathrm{PT}}$
10: ▷ Step 3: Adaptive shrinkage toward group mean (Eq. (29))
11: $\hat{\omega} = \mathbf{get\_shrink\_param}((\hat{\theta}_j^{\mathrm{PT}})_{j=1}^m, \bar{\theta}^{\mathrm{PT}}, (\tilde{\sigma}_j^2)_{j=1}^m)$
12: **for** $j = 1$ to $m$ **do**
13:     $\hat{\omega}_j = \hat{\omega}/(\hat{\omega} + \tilde{\sigma}_j^2)$
14:     $\hat{\theta}_j^{\mathrm{Avg}} = \hat{\omega}_j \hat{\theta}_j^{\mathrm{PT}} + (1 - \hat{\omega}_j)\bar{\theta}^{\mathrm{PT}}$
15: **end for**
16: **return** $\{\hat{\theta}_j^{\mathrm{Avg}}\}_{j=1}^m$

---

**Optimizing $\omega$ via SURE.** Xie et al. (2012) proposed the following unbiased risk estimate to optimize $\omega$ for this estimator. Note that even though the group mean is also correlated with each PT estimator, we still denote the following SURE instead of CURE following the nomenclature in Xie et al. (2012).

$$\hat{\omega} \in \arg\min_{\omega \geq 0} \frac{1}{m} \sum_{j=1}^m \mathrm{SURE}(\hat{\theta}_{j,\omega}^{\mathrm{Shrink}}) \tag{29}$$

$$\mathrm{SURE}(\hat{\theta}_{j,\omega}^{\mathrm{Shrink}}) := \left[ (1 - \omega_j)(\hat{\theta}_j^{\mathrm{PT}} - \bar{\theta}^{\mathrm{PT}}) \right]^2 + (1 - \omega_j)(\omega + (2/m - 1)\tilde{\sigma}_j^2).$$

We refer to Algorithm 4 for a full pseudo-code implementation of "shrink-average" estimator.

## E. Experiment Details

### E.1. Synthetic Model

**Motivation.** In Example 2.3, we described the following data generation process (copied from Eq. (8))

$$\eta_j \sim \mathcal{U}[-1, 1], \quad j = 1, \dots, m,$$
$$X_{ij} \sim \mathcal{N}(\eta_j, \psi^2), \quad Y_{ij}|X_{ij} \sim \mathcal{N}(2\eta_j X_{ij} - \eta_j^2, c), \quad i = 1, \dots, n_j,$$

and the same for $(\tilde{X}_{ij}, \tilde{Y}_{ij})$. $\psi$ and $c$ are two hyperparameters that we chose to be 0.1 and 0.05, respectively. The (marginal) mean and variance of $Y_{ij}$ are

$$\theta_j := \mathbb{E}_{\eta_j}[Y_{ij}] = \eta_j^2, \quad \sigma_j^2 := \mathrm{Var}_{\eta_j}[Y_{ij}] = 4\eta_j^2 \psi^2 + c.$$

To understand the motivation behind this setup, we can further inspect the covariance between $X_{ij}$ and $Y_{ij}$, which can be verified to be $\mathrm{Cov}_{\eta_j}[X_{ij}, Y_{ij}] = 2\eta_j \psi^2$. Therefore, if we consider the ratio between the absolute covariance and the variance (of $Y_{ij}$) as a characterization of the "inherent predictability" of a problem, we see that

$$\frac{|\mathrm{Cov}_{\eta_j}[X_{ij}, Y_{ij}]|}{\mathrm{Var}_{\eta_j}[Y_{ij}]} = \frac{2|\eta_j|\psi^2}{4\eta_j^2 \psi^2 + c}$$

which has its minimum when $\eta_j = 0$ and increases monotonically in $|\eta_j|$ for $|\eta_j| \in [0, 1]$ given our specific choices of $\psi$ and $c$ (see Figure 4). In other words, problems with $\eta_j$ close to the origin have a lower "predictability;" whereas when $\eta_j$ moves away from zero, the problems become easier to solve. This quantitatively reflects the pattern we see in Figure 3, where we display the power-tuning parameters as a function of $\eta_j$.

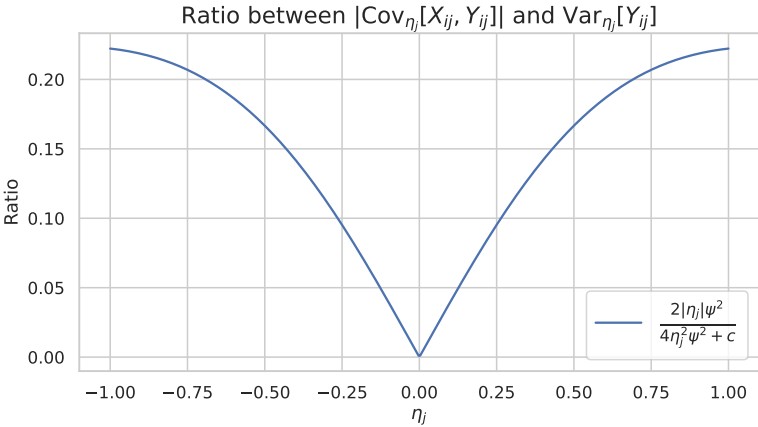

*Figure 4.* The ratio between $|\text{Cov}_{\eta_j}[X_{ij}, Y_{ij}]|$ and $\text{Var}_{\eta_j}[Y_{ij}]$ as a function of $\eta_j$. The constants are set to $\psi = 0.1$ and $c = 0.05$.

**Expressions for $\theta_j, \mu_j, \sigma_j^2, \tau_j^2, \gamma_j$ when $f(x) = |x|$.** When we work with the synthetic model using the flawed predictor $f(x) = |x|$, we can match the form of our dataset with the general setting in Assumption 2.2 by identifying closed-form expressions for the model parameters $\theta_j, \mu_j, \sigma_j^2, \tau_j^2, \gamma_j$.

$$\theta_j = \eta_j^2, \quad \sigma_j^2 = 4\eta_j^2\psi^2 + c,$$

$$\gamma_j = 2\eta_j\psi^2\sqrt{\frac{2}{\pi}}e^{-\eta_j^2/(2\psi^2)}, \quad \mu_j = \sqrt{\frac{2}{\pi}}\psi\exp\left(-\frac{\eta_j}{2\psi^2}\right) + \eta_j\left[\Phi\left(\frac{\eta_j}{\psi}\right) - \frac{1}{2}\right],$$

$$\tau_j^2 = \eta_j^2 + \psi^2 - \left[\sqrt{\frac{2\psi^2}{\pi}}\exp\left(-\frac{\eta_j^2}{2\psi^2}\right) + \eta_j\left(2\Phi\left(\frac{\eta_j}{\psi}\right) - 1\right)\right]^2,$$

where $\Phi(\cdot)$ denotes the standard normal distribution function.

**Expressions for $\theta_j, \mu_j, \sigma_j^2, \tau_j^2, \gamma_j$ when $f(x) = x^2$.** Similar closed-form expressions can be derived when we use the other predictor $f(x) = x^2$. Note that $\theta_j$ and $\sigma_j^2$ remain the same.

$$\theta_j = \eta_j^2, \quad \sigma_j^2 = 4\eta_j^2\psi^2 + c,$$

$$\gamma_j = 4\eta_j^2\psi^2, \quad \mu_j = \eta_j^2 + \psi^2, \quad \tau_j^2 = 2\psi^4 + 4\eta_j^2\psi^2.$$

In experiments involving the synthetic model with both predictors, we are able to leverage these closed-form expressions and supplement the ground-truth parameters to our datasets.

**Interpretation of MSE.** In the synthetic experiments, since we have access to the true prior for $\eta_j$ (therefore for $\theta_j$) and resample them for each problem across $K$ trials, the MSE we obtained in Section 7.1 is an unbiased estimate of the *Bayes Risk* defined in Eq. (11).

### E.2. Amazon Review Ratings Dataset

**Dataset & Preprocessing.** The *Amazon Fine Food Reviews* dataset, provided by the Stanford Network Analysis Project (SNAP; SNAP (2014)) on Kaggle,[21] comes in a clean format. We group reviews by their `ProductID`. For each review, we concatenate the title and body text to form the covariate, while the response is the reviewer's score/rating (1 to 5 stars). Here's a sample review:

---

**Score:** 4            **Product:** BBQ Pop Chips
**Title:** Delicious!
**Text:** BBQ Pop Chips are a delicious tasting healthier chip than many on the market. They are light and full of flavor. The 3 oz bags are a great size to have. I would recommend them to anyone.

---

[21] https://www.kaggle.com/datasets/snap/amazon-fine-food-reviews

We focus on the top $m = 200$ products with the most reviews for the compound mean estimation of average ratings. This approach mitigates extreme heteroscedasticity across estimators for different problems, which could unduly favor shrinkage-based methods when considering unweighted compound risk. There are a total of 74,913 reviews for all 200 products.

**Fine-tuning `BERT`.** The Bidirectional Encoder Representations from Transformers (`BERT`) model is a widely adopted language model for many NLP tasks including text classification (Devlin et al., 2019). However, pre-training `BERT` from scratch is time-consuming and requires large amounts of data. We therefore use the `bert-base-multilingual-uncased-sentiment` model[22] from Town (2023) as the base model, denoted as `BERT-base`. `BERT-base` is pre-trained on general product reviews (not exclusive to Amazon) in six languages. It achieves 67.5% prediction accuracy on a validation set of 100 products ($\sim$46k reviews).

Then, we further fine-tune it on the held-out review data, that is, reviews outside the top 200 products, for 2 full epochs. The fine-tuning is done using Hugging Face's `transformers` library (Wolf, 2019). After fine-tuning, the `BERT-tuned` model achieves 78.8% accuracy on the same validation set.

### E.3. Spiral Galaxy Fractions (Galaxy Zoo 2)

**Dataset & Preprocessing.** The Galaxy Zoo 2 (GZ2) project[23] contains a large collection of human-annotated classification results for galaxy images from SDSS. However, instead of having a single dataframe, GZ2 has many different tables—each for subsets of the SDSS raw data. We begin with a particular subset of 239,696 images with metadata drawn from Hart et al. (2016). Our data cleaning pipeline is inspired by Lin et al. (2021), which removes missing data and relabels the class name of each galaxy image to a more readable format:

> **Class Names:** Round Elliptical, In-between Elliptical, Cigar-shaped Elliptical, Edge-on Spiral, Barred Spiral, Unbarred Spiral, Irregular, Merger

In the downstream estimation problems, we consider a galaxy "spiral" if it is classified as one of the three classes ending with "Spiral", otherwise "non-spiral". Below we display a few examples of galaxy images. Each image has dimensions of $424 \times 424 \times 3$, where the third dimension represents the three filter channels: g (green), r (red), and i (infrared). The cleaned dataset has 155,951 images in total.

*Figure 5.* Example of spiral & non-spiral galaxy images from Galaxy Zoo 2.

---

[22]https://huggingface.co/nlptown/bert-base-multilingual-uncased-sentiment
[23]https://data.galaxyzoo.org/

The additional SDSS metadata for GZ2[24] contains valuable information that directly partitions the galaxies based on certain attributes, e.g., REDSHIFT_SIMPLE_BIN based on galaxy redshift measurements, and WVT_BIN calculated by weighted Voronoi tessellation. These partitions naturally motivate fine-grained compound mean estimation on this dataset.

After partitioning the images based on WVT_BIN,[25] we consider only the top $m = 100$ partitions based on the cutoff that each problem should have $\geq 150$ images (many partitions have very few galaxy images in them), for the same reason as in the Amazon Review dataset. Finally, we have a total of $\sim$100k images as covariates (either $X_{ij}$ or $\tilde{X}_{ij}$) for our problem.

**Training the Predictor.** We employ the ResNet50 architecture (He et al., 2016), utilizing the pre-trained model from torchvision initially trained on ImageNet (Deng et al., 2009). To tailor the model to our task, we fine-tune it on $\sim$50k images excluded from the top $m$ problems. The model is trained to classify galaxies into eight categories, later condensed into a binary spiral/non-spiral classification for prediction. We use a batch size of 256 and Adam optimizer (Kingma & Ba, 2015) with a learning rate of 1e-3. After 20 epochs, the model achieves 87% training accuracy and 83% test accuracy. Despite these promising results, Table 3 indicates that the predictions still require debiasing for accurate estimation.

### E.4. Benchmarking in real-world datasets

In this appendix we describe the steps to obtain the MSEs and their standard errors for real-world datasets shown in Table 3.

Let $K$ be the number of experiment trials, $T_j$ be the total number of data points for problem $j$, i.e. $\{\dot{X}_{ij}, \dot{Y}_{ij}\}_{i=1}^{T_j}$ represents the "raw data" we have, and $n_j, N_j$ be the desired number of labeled/unlabeled data to simulate, usually calculated through a hyper-parameter splitting ratio (e.g. $N_j = \lfloor r \cdot T_j \rfloor$, $n_j = T_j - N_j$ for $r = 0.8$ in our case).

1. Following evaluation methodology in existing PPI literature, e.g., (Angelopoulos et al., 2023), we first calculate the mean of all responses for each problem and treat it as the pseudo ground-truth, i.e., $\dot{\theta}_j := \frac{1}{T_j} \sum_i \dot{Y}_{ij}$.

2. For each trial $k \in [K]$, we create a random permutation for the raw data, with indices permuted by $\kappa : \mathbb{N} \to \mathbb{N}$, and obtain the labeled and unlabeled datasets for problem $j$ as

$$\{X_{ij}, Y_{ij}\}_{i=1}^{n_j} = \{\dot{X}_{\kappa(i)j}, \dot{Y}_{\kappa(i)j}\}_{i=1}^{n_j}, \quad \{\tilde{X}_{ij}\}_{i=1}^{N_j} = \{\dot{X}_{\kappa(i)j}\}_{i=n_j+1}^{T_j}$$

3. We proceed with using these datasets to obtain the baseline and PAS estimators. Let $\hat{\theta}_j^k$ be an estimator for the $j$-th problem at trial $k$, then our final reported MSE and standard error is calculated as

$$\widehat{\text{MSE}}_K(\hat{\boldsymbol{\theta}}) := \frac{1}{K} \sum_{k=1}^{K} \left( \frac{1}{m} \sum_{j=1}^{m} (\hat{\theta}_j^k - \dot{\theta}_j)^2 \right), \quad \text{SE}_K(\hat{\boldsymbol{\theta}}) := \frac{1}{\sqrt{K}} \sqrt{ \frac{1}{K-1} \sum_{k=1}^{K} \left( \frac{1}{m} \sum_{j=1}^{m} (\hat{\theta}_j^k - \dot{\theta}_j)^2 - \widehat{\text{MSE}}_K(\hat{\boldsymbol{\theta}}) \right) }.$$

Note that the standard error only accounts for uncertainty due to the random splits into labeled and unlabeled datasets.

### E.5. Additional Experiment with Varying Labeled/Total Data Ratio

In Table 3, we report our experiment results on different tasks and predictors, but *fixing the ratio between labeled and total amounts of data* $n_j/(n_j + N_j) = 0.2$ for all problems. To verify the broader applicability of our method, we repeat our experiments across a much wider range of ratios—from 1% to 40%—and report the results in Figure 6. For each ratio, we follow exactly the same data-splitting and benchmarking procedures specified in Appendix E.4.

### E.6. Computational Resources

All the experiments were conducted on a compute cluster with Intel Xeon Silver 4514Y (16 cores) CPU, Nvidia A100 (80GB) GPU, and 64GB of memory. Fine-tuning the BERT-tuned model took 2 hours, and training the ResNet50

---

[24]The column names and their meanings are available at https://data.galaxyzoo.org/data/gz2/gz2sample.txt.

[25]In the Galaxy Zoo 2 dataset, WVT_BIN denotes Voronoi bins constructed based on each galaxy's intrinsic size and absolute magnitude. The motivation and implementation of this binning strategy are detailed in Hart et al. (2016), who justify such partitioning—aligned with our compound mean estimation setup—by noting that spiral arm morphology exhibits systematic dependencies on stellar mass and related intrinsic properties.

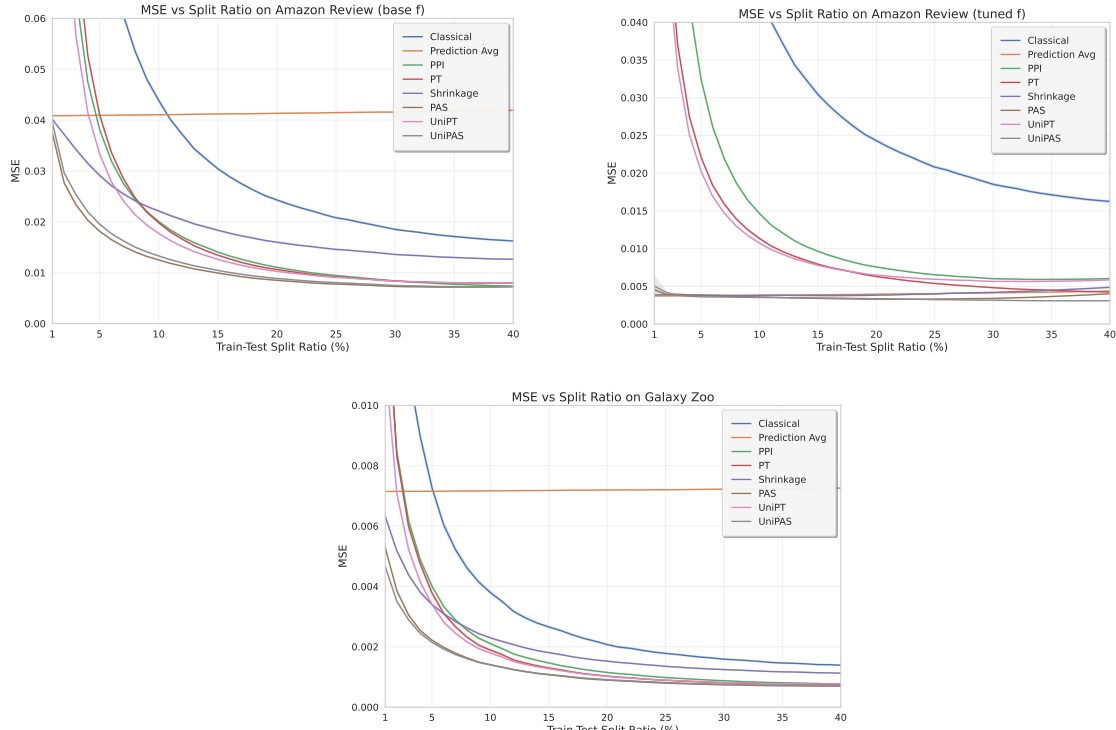

*Figure 6.* Average MSEs for the three real-world datasets when the labeled/unlabeled split ratio varies from 1% to 40%. `UniPT` and `UniPAS` are the two newly added variants of PT and PAS estimators, respectively.

model took 1 hour. All the inferences (predictions) can be done within 10 minutes. The nature of our research problem requires running the prediction only once per dataset, making it fast to benchmark all estimators for $K = 200$ trials using existing predictions.

### E.7. Code Availability

The code for reproducing the experiments is available at https://github.com/listar2000/prediction-powered-adaptive-shrinkage.

## F. Proofs of Theoretical Results

### F.1. Proof of Theorem. 4.1

For each problem $j \in [m]$, we are shrinking the PT estimator $\hat{\theta}_j^{\mathrm{PT}}$ obtained from the first stage toward $\tilde{Z}_j^f$, the prediction mean on the unlabeled data. Conditioning on $\eta_j$, we denote

$$\tilde{\sigma}_j^2 := \mathrm{Var}_{\eta_j}\left[\hat{\theta}_j^{\mathrm{PT}}\right] = \mathrm{Var}_{\eta_j}\left[\hat{\theta}_{j,\lambda_j^*}^{\mathrm{PPI}}\right],$$

$$\tilde{\gamma}_j := \mathrm{Cov}_{\eta_j}\left[\hat{\theta}_j^{\mathrm{PT}}, \tilde{Z}_j^f\right] = \lambda_j^* \mathrm{Var}_{\eta_j}\left[\tilde{Z}_j^f\right],$$

where all the first and second moments of $\hat{\theta}_j^{\mathrm{PT}}$ and $\tilde{Z}_j^f$ exist under the conditions of Assumption 2.2. For each global $\omega \geq 0$, the shrinkage parameter for the $j$-th problem is defined as $\omega_j := \omega/(\omega + \tilde{\sigma}_j^2)$. Then, following the result in Theorem B.1, CURE for $\hat{\theta}_{j,\omega_j}^{\mathrm{PAS}} := \omega_j \hat{\theta}_j^{\mathrm{PT}} + (1 - \omega_j)\tilde{Z}_j^f$,

$$\mathrm{CURE}\left(\hat{\theta}_{j,\omega}^{\mathrm{PAS}}\right) = (2\omega_j - 1)\tilde{\sigma}_j^2 + 2(1 - \omega_j)\tilde{\gamma}_j + \left[(1 - \omega_j)(\hat{\theta}_j^{\mathrm{PT}} - \tilde{Z}_j^f)\right]^2,$$

is an unbiased estimator of the risk, i.e.,

$$\mathbb{E}_{\eta_j}\left[\mathrm{CURE}\left(\hat{\theta}_{j,\omega}^{\mathrm{PAS}}\right)\right] = R(\hat{\theta}_{j,\omega_j}^{\mathrm{PAS}}, \theta_j).$$

Finally, the CURE for the collection of estimators is $\hat{\boldsymbol{\theta}}_{\omega}^{\mathrm{PAS}} := (\hat{\theta}_{1,\omega}^{\mathrm{PAS}}, \ldots, \hat{\theta}_{m,\omega}^{\mathrm{PAS}})^{\mathsf{T}}$

$$\mathrm{CURE}\left(\hat{\boldsymbol{\theta}}_{\omega}^{\mathrm{PAS}}\right) := \frac{1}{m}\sum_{j=1}^{m}\mathrm{CURE}\left(\hat{\theta}_{j,\omega}^{\mathrm{PAS}}\right),$$

which is an unbiased estimator of the compound risk $\mathcal{R}_m(\hat{\boldsymbol{\theta}}_{\omega}^{\mathrm{PAS}}, \boldsymbol{\theta})$ by linearity of the expectation. $\qquad\square$

### F.2. Formal conditions and proof of Proposition 5.1

We aim to prove that CURE converges uniformly to the true squared-error loss $\ell_m(\hat{\boldsymbol{\theta}}_{\omega}^{\mathrm{PAS}}, \boldsymbol{\theta})$ as $m \to \infty$. Specifically, our goal is to establish

$$\sup_{\omega \geq 0}\left|\mathrm{CURE}(\hat{\boldsymbol{\theta}}_{\omega}^{\mathrm{PAS}}) - \ell_m(\hat{\boldsymbol{\theta}}_{\omega}^{\mathrm{PAS}}, \boldsymbol{\theta})\right| \xrightarrow[m\to\infty]{L^1} 0.$$

For this proposition, all the expectation and variance terms without subscript are conditioning on $\boldsymbol{\eta}$. We keep using the notations $\theta_j = \mathbb{E}[\hat{\theta}_j^{\mathrm{PT}}]$, $\mu_j = \mathbb{E}[\tilde{Z}_j^f]$, $\tilde{\sigma}_j^2 = \mathrm{Var}[\hat{\theta}_j^{\mathrm{PT}}]$ and $\tilde{\gamma}_j = \mathrm{Cov}[\hat{\theta}_j^{\mathrm{PT}}, \tilde{Z}_j^f]$. For this proposition, additional assumptions are placed on the data generating process (integrated over $\mathbb{P}_{\eta}$). We first show how they translate to moment conditions on the estimators $\hat{\theta}_j^{\mathrm{PT}}$ and $\tilde{Z}_j^f$.

**Lemma F.1.** *Under the assumptions of Proposition 5.1, and specifically $\mathbb{E}_{\mathbb{P}_{\eta}}\left[f(X_{ij})^4\right] < \infty$ and $\mathbb{E}_{\mathbb{P}_{\eta}}\left[Y_{ij}^4\right] < \infty$, it holds that:*

$$\sup_{j\geq 1}\mathbb{E}_{\mathbb{P}_{\eta}}\left[\left(\hat{\theta}_j^{\mathrm{PT}}\right)^4\right] < \infty, \qquad \sup_{j\geq 1}\mathbb{E}_{\mathbb{P}_{\eta}}\left[\left(\tilde{Z}_j^f\right)^4\right] < \infty,$$

$$\mathbb{E}_{\mathbb{P}_{\eta}}\left[\theta_j^4\right] < \infty, \qquad \mathbb{E}_{\mathbb{P}_{\eta}}\left[\mu_j^4\right] < \infty.$$

*Proof.* By Minkowski's inequality, we have

$$\mathbb{E}_{\mathbb{P}_{\eta}}\left[\left(\hat{\theta}_j^{\mathrm{PT}}\right)^4\right] = \mathbb{E}_{\mathbb{P}_{\eta}}\left[\left(\bar{Y}_j + \lambda_j^*(\tilde{Z}_j^f - \bar{Z}_j^f)\right)^4\right]$$

$$\leq \left(\mathbb{E}_{\mathbb{P}_{\eta}}\left[\bar{Y}_j^4\right]^{1/4} + \lambda_j^*\mathbb{E}_{\mathbb{P}_{\eta}}\left[\left(\tilde{Z}_j^f\right)^4\right]^{1/4} + \lambda_j^*\mathbb{E}_{\mathbb{P}_{\eta}}\left[\left(\bar{Z}_j^f\right)^4\right]^{1/4}\right)^4,$$

so it suffices to bound the fourth moments of $\bar{Y}_j, \tilde{Z}_j, \bar{Z}_j$. We proceed with $\bar{Y}_j$ first. Again, by Minkowski's inequality

$$\mathbb{E}_{\mathbb{P}_{\eta}}\left[\bar{Y}_j^4\right]^{1/4} = \mathbb{E}_{\mathbb{P}_{\eta}}\left[\left(\sum_{i=1}^{n_j} n_j^{-1}Y_{ij}\right)^4\right]^{1/4}$$

$$\leq \sum_{i=1}^{n_j}\mathbb{E}_{\mathbb{P}_{\eta}}\left[\left(n_j^{-1}Y_{ij}\right)^4\right]^{1/4}$$

$$= \sum_{i=1}^{n_j} n_j^{-1}\mathbb{E}_{\mathbb{P}_{\eta}}\left[Y_{ij}^4\right]^{1/4}$$

$$= \mathbb{E}_{\mathbb{P}_{\eta}}\left[Y_{ij}^4\right]^{1/4} < \infty.$$

The given assumption on the data generating process is used in the last line. Although the left-hand side of the inequality may depend on $j$ (via the deterministic $n_j$), the right-hand side does not depend on $j$ (since we are integrating over $\mathbb{P}_{\eta}$) and so we may take a supremum over all $j$ on the left-hand side. The arguments for $\tilde{Z}_j$ and $\bar{Z}_j$ based on the finiteness of

$\mathbb{E}_{\mathbb{P}_\eta}[f(X_{ij})^4]$ are analogous. Next, by Jensen's inequality we have

$$\theta_j^4 = \mathbb{E}_{\eta_j}\left[\hat{\theta}_j^{\text{PT}}\right]^4 \leq \mathbb{E}_{\eta_j}\left[(\hat{\theta}_j^{\text{PT}})^4\right]$$

$$\implies \quad \mathbb{E}_{\mathbb{P}_\eta}\left[\theta_j^4\right] \leq \mathbb{E}_{\mathbb{P}_\eta}\left[(\hat{\theta}_j^{\text{PT}})^4\right] < \infty,$$

and similarly we can also obtain $\mathbb{E}_{\mathbb{P}_\eta}[\mu_j^4] < \infty$. $\qquad\qquad\qquad\qquad\qquad\qquad\qquad\qquad$ $\square$

With Lemma F.1, we will prove Proposition 5.1 via the following steps.

**Step 1: Decompose the difference.** We first decompose both CURE and the loss separately as

$$\text{CURE}(\hat{\boldsymbol{\theta}}_\omega^{\text{PAS}}) = \frac{1}{m}\sum_{j=1}^m \left((2\omega_j - 1)\tilde{\sigma}_j^2 + \left[(1-\omega_j)(\hat{\theta}_j^{\text{PT}} - \tilde{Z}_j^f)\right]^2 + 2(1-\omega_j)\tilde{\gamma}_j\right)$$

$$= \frac{1}{m}\sum_{j=1}^m \underbrace{\left((2\omega_j - 1)\tilde{\sigma}_j^2 + (1-\omega_j)^2(\hat{\theta}_j^{\text{PT}} - \mu_j)^2\right)}_{\mathbb{I}(\omega)}$$

$$+ \frac{1}{m}\sum_{j=1}^m \underbrace{\left(2(1-\omega_j)\tilde{\gamma}_j + 2(1-\omega_j)^2(\mu_j - \tilde{Z}_j^f)(\hat{\theta}_j^{\text{PT}} - \mu_j) + (1-\omega_j)^2(\mu_j - \tilde{Z}_j^f)^2\right)}_{\mathbb{II}(\omega)}$$

$$\ell_m(\hat{\boldsymbol{\theta}}_\omega^{\text{PAS}}, \boldsymbol{\theta}) = \frac{1}{m}\sum_{j=1}^m (\omega_j\hat{\theta}_j^{\text{PT}} + (1-\omega_j)\tilde{Z}_j^f - \theta_j)^2$$

$$= \frac{1}{m}\sum_{j=1}^m \underbrace{(\omega_j\hat{\theta}_j^{\text{PT}} + (1-\omega_j)\mu_j - \theta_j)^2}_{\mathbb{I}^*(\omega)}$$

$$+ \frac{1}{m}\sum_{j=1}^m \underbrace{\left(2(1-\omega_j)(\tilde{Z}_j^f - \mu_j)(\hat{\theta}_j^{\text{PT}} - \theta_j) + 2(1-\omega_j)^2(\mu_j - \tilde{Z}_j^f)(\hat{\theta}_j^{\text{PT}} - \mu_j) + (1-\omega_j)^2(\mu_j - \tilde{Z}_j^f)^2\right)}_{\mathbb{II}^*(\omega)},$$

and we are interested in bounding

$$\sup_{\omega \geq 0}\left|\text{CURE}(\hat{\boldsymbol{\theta}}_\omega^{\text{PAS}}) - \ell_m(\hat{\boldsymbol{\theta}}_\omega^{\text{PAS}}, \boldsymbol{\theta})\right| \leq \sup_{\omega \geq 0}|\mathbb{I}(\omega) - \mathbb{I}^*(\omega)| + \sup_{\omega \geq 0}|\mathbb{II}(\omega) - \mathbb{II}^*(\omega)|. \tag{30}$$

**Step 2: Bounding the first difference** $\Delta_1(\omega) := \mathbb{I}(\omega) - \mathbb{I}^*(\omega)$**.** The proof in this step is directly adapted from **Theorem 5.1** in Xie et al. (2012) and generalizes to non-Gaussian data. With some algebraic manipulation, we can further decompose

$$\Delta_1(\omega) = \frac{1}{m}\sum_{j=1}^m \left((2\omega_j - 1)\tilde{\sigma}_j^2 + (1-\omega_j)^2(\hat{\theta}_j^{\text{PT}} - \mu_j)^2\right)$$

$$- \frac{1}{m}\sum_{j=1}^m (\omega_j\hat{\theta}_j^{\text{PT}} + (1-\omega_j)\mu_j - \theta_j)^2$$

$$= \text{CURE}(\hat{\boldsymbol{\theta}}_\omega^0) - \ell_m(\hat{\boldsymbol{\theta}}_\omega^0, \boldsymbol{\theta}) - \frac{2}{m}\sum_{j=1}^m \mu_j(1-\omega_j)(\hat{\theta}_j^{\text{PT}} - \theta_j)$$

where $\quad \text{CURE}(\hat{\boldsymbol{\theta}}_\omega^0) = \frac{1}{m}\sum_{j=1}^m \left((2\omega_j - 1)\tilde{\sigma}_j^2 + (1-\omega_j)^2(\hat{\theta}_j^{\text{PT}})^2\right), \quad \ell_m(\hat{\boldsymbol{\theta}}_\omega^0, \boldsymbol{\theta}) = \frac{1}{m}\sum_{j=1}^m (\omega_j\hat{\theta}_j^{\text{PT}} - \theta_j)^2,$

corresponds to CURE and the loss of the "shrink-toward-zero" estimator $\hat{\theta}_{j,\omega}^0 := \omega_j \hat{\theta}_j^{\text{PT}}$. We thus have

$$\sup_{\omega \geq 0} |\Delta_1(\omega)| \leq \sup_{\omega \geq 0} \left| \text{CURE}(\hat{\boldsymbol{\theta}}_\omega^0) - \ell_m(\hat{\boldsymbol{\theta}}_\omega^0, \boldsymbol{\theta}) \right| + \frac{2}{m} \sup_{\omega \geq 0} \left| \sum_{j=1}^m \mu_j (1 - \omega_j)(\hat{\theta}_j^{\text{PT}} - \theta_j) \right|. \tag{31}$$

Now, rearrangements of terms gives that

$$\sup_{\omega \geq 0} \left| \text{CURE}(\hat{\boldsymbol{\theta}}_\omega^0) - \ell_m(\hat{\boldsymbol{\theta}}_\omega^0, \boldsymbol{\theta}) \right| = \sup_{\omega \geq 0} \left| \frac{1}{m} \sum_{j=1}^m \left( (\hat{\theta}_j^{\text{PT}})^2 - \tilde{\sigma}_j^2 - \theta_j^2 - 2\omega_j \left( (\hat{\theta}_j^{\text{PT}})^2 - \hat{\theta}_j^{\text{PT}} \theta_j - \tilde{\sigma}_j^2 \right) \right) \right|$$

$$\leq \underbrace{\left| \frac{1}{m} \sum_{j=1}^m (\hat{\theta}_j^{\text{PT}})^2 - \tilde{\sigma}_j^2 - \theta_j^2 \right|}_{(*)} + \underbrace{\sup_{\omega \geq 0} \left| \frac{1}{m} \sum_{j=1}^m 2\omega_j \left( (\hat{\theta}_j^{\text{PT}})^2 - \hat{\theta}_j^{\text{PT}} \theta_j - \tilde{\sigma}_j^2 \right) \right|}_{(**)}.$$

For the first term $(*)$,

$$\mathbb{E}_{\mathbb{P}_\eta} \left[ \mathbb{E}_{\boldsymbol{\eta}} \left[ \left( \frac{1}{m} \sum_{j=1}^m (\hat{\theta}_j^{\text{PT}})^2 - \tilde{\sigma}_j^2 - \theta_j^2 \right)^2 \right] \right] = \frac{1}{m^2} \sum_{j=1}^m \mathbb{E}_{\mathbb{P}_\eta} \left[ \text{Var}_{\eta_j} \left[ (\hat{\theta}_j^{\text{PT}})^2 \right] \right] \leq \frac{1}{m} \sup_j \text{Var}_{\mathbb{P}_\eta} \left[ (\hat{\theta}_j^{\text{PT}})^2 \right].$$

Thus by Jensen's inequality and iterated expectation:

$$\mathbb{E}_{\mathbb{P}_\eta} \left[ \left| \frac{1}{m} \sum_{j=1}^m (\hat{\theta}_j^{\text{PT}})^2 - \tilde{\sigma}_j^2 - \theta_j^2 \right| \right] \leq \left( \frac{1}{m} \sup_j \text{Var}_{\mathbb{P}_\eta} \left[ (\hat{\theta}_j^{\text{PT}})^2 \right] \right)^{1/2}. \tag{32}$$

For the second term $(**)$, we start by arguing conditionally on $\boldsymbol{\eta}$, which implies in particular that we may treat all the $\tilde{\sigma}_j^2$ as fixed. It is thus without loss of generality to assume that $\tilde{\sigma}_1^2 \leq ... \leq \tilde{\sigma}_m^2$ (by first sorting problems according to the value of $\tilde{\sigma}_j^2$). Then, since $\omega_j$ is monotonic function of $\tilde{\sigma}_j^2$ for any fixed $\omega \geq 0$, we have $1 \geq \omega_1 \geq ... \geq \omega_m \geq 0$. The following inequality follows:

$$\sup_{\omega \geq 0} \left| \frac{1}{m} \sum_{j=1}^m 2\omega_j \left( (\hat{\theta}_j^{\text{PT}})^2 - \hat{\theta}_j^{\text{PT}} \theta_j - \tilde{\sigma}_j^2 \right) \right| \leq \max_{1 \geq c_1 \geq ... \geq c_m \geq 0} \left| \frac{2}{m} \sum_{j=1}^m c_j \left( (\hat{\theta}_j^{\text{PT}})^2 - \hat{\theta}_j^{\text{PT}} \theta_j - \tilde{\sigma}_j^2 \right) \right|. \tag{33}$$

The following lemma would help us for handling the RHS of (33) (the same structural form of it will appear repeatedly in subsequent parts of the proof).

**Lemma F.2.** *Let $A_1, \ldots, A_n$ be real numbers. Then*

$$\max_{1 \geq c_1 \geq \cdots \geq c_n \geq 0} \left| \sum_{i=1}^n c_i A_i \right| = \max_{1 \leq k \leq n} \left| \sum_{i=1}^k A_i \right|.$$

*Proof.* Define $S_k = \sum_{i=1}^k A_i$ for $k = 1, \ldots, n$, and let $c_1, \ldots, c_n$ be real numbers satisfying $1 \geq c_1 \geq \cdots \geq c_n \geq 0$. Set $c_{n+1} = 0$. Then we can rewrite

$$\sum_{i=1}^n c_i A_i = \sum_{k=1}^n (c_k - c_{k+1}) \left( \sum_{i=1}^k A_i \right) = \sum_{k=1}^n (c_k - c_{k+1}) S_k.$$

Since $c_k \geq c_{k+1}$, each $\alpha_k := c_k - c_{k+1}$ is nonnegative, and

$$\sum_{k=1}^n \alpha_k = c_1 - c_{n+1} \leq 1.$$

Hence,

$$\left| \sum_{i=1}^n c_i A_i \right| = \left| \sum_{k=1}^n \alpha_k S_k \right| \leq \sum_{k=1}^n \alpha_k |S_k| \leq \left( \max_{1 \leq k \leq n} |S_k| \right) \left( \sum_{k=1}^n \alpha_k \right) \leq \max_{1 \leq k \leq n} |S_k|.$$

This shows

$$\max_{1 \geq c_1 \geq \cdots \geq c_n \geq 0} \left| \sum_{i=1}^n c_i A_i \right| \leq \max_{1 \leq k \leq n} \left| \sum_{i=1}^k A_i \right|.$$

To see that this upper bound can be attained, consider for each $k$ the choice

$$c_1 = c_2 = \cdots = c_k = 1, \quad c_{k+1} = c_{k+2} = \cdots = c_n = 0.$$

Since $1 \geq c_1 \geq \cdots \geq c_n \geq 0$, we have that

$$\left| \sum_{i=1}^n c_i A_i \right| = \left| \sum_{i=1}^k A_i \right| = |S_k|.$$

Taking the maximum over all such $k \in \{1, \ldots, n\}$ matches $\max_{1 \leq k \leq n} |S_k|$. Thus,

$$\max_{1 \geq c_1 \geq \cdots \geq c_n \geq 0} \left| \sum_{i=1}^n c_i A_i \right| = \max_{1 \leq k \leq n} \left| \sum_{i=1}^k A_i \right|,$$

as claimed. $\qquad\square$

With Lemma F.2 in hand, we have

$$\max_{1 \geq c_1 \geq \cdots \geq c_m \geq 0} \left| \frac{2}{m} \sum_{j=1}^m c_j \left( (\hat{\theta}_j^{\mathrm{PT}})^2 - \hat{\theta}_j^{\mathrm{PT}} \theta_j - \tilde{\sigma}_j^2 \right) \right| = \max_{1 \leq k \leq m} \left| \frac{2}{m} \sum_{j=1}^k \left( (\hat{\theta}_j^{\mathrm{PT}})^2 - \hat{\theta}_j^{\mathrm{PT}} \theta_j - \tilde{\sigma}_j^2 \right) \right|.$$

Let $M_k := \sum_{j=1}^k ((\hat{\theta}_j^{\mathrm{PT}})^2 - \hat{\theta}_j^{\mathrm{PT}} \theta_j - \tilde{\sigma}_j^2)$, it is easy to see that $\{M_k\}_{k=1}^m$ forms a martingale conditional on $\boldsymbol{\eta}$. Therefore, by a standard $L^2$ maximal inequality (ref. Theorem 4.4.6 in Durrett (2019)), we have

$$\mathbb{E}_{\boldsymbol{\eta}} \left[ \max_{1 \leq k \leq m} M_k^2 \right] \leq 4 \mathbb{E}_{\boldsymbol{\eta}} \left[ M_m^2 \right] = 4 \sum_{j=1}^m \mathrm{Var}_{\eta_j} \left[ (\hat{\theta}_j^{\mathrm{PT}})^2 - \hat{\theta}_j^{\mathrm{PT}} \theta_j \right], \tag{34}$$

which then implies

$$\mathbb{E}_{\mathbb{P}_\eta} \left[ \left( \sup_{\omega \geq 0} \left| \frac{1}{m} \sum_{j=1}^m 2\omega_j \left( (\hat{\theta}_j^{\mathrm{PT}})^2 - \hat{\theta}_j^{\mathrm{PT}} \theta_j - \tilde{\sigma}_j^2 \right) \right| \right)^2 \right] \leq \frac{4}{m^2} \mathbb{E}_{\mathbb{P}_\eta} \left[ \max_{1 \leq k \leq m} M_k^2 \right]$$

$$= \frac{16}{m^2} \sum_{j=1}^m \mathbb{E}_{\mathbb{P}_\eta} \left[ \mathrm{Var}_{\eta_j} \left[ (\hat{\theta}_j^{\mathrm{PT}})^2 - \hat{\theta}_j^{\mathrm{PT}} \theta_j \right] \right]$$

$$\leq \frac{16}{m} \sup_j \mathrm{Var}_{\mathbb{P}_\eta} \left[ (\hat{\theta}_j^{\mathrm{PT}})^2 - \hat{\theta}_j^{\mathrm{PT}} \theta_j \right]$$

$$\implies \mathbb{E}_{\mathbb{P}_\eta} \left[ \sup_{\omega \geq 0} \left| \frac{1}{m} \sum_{j=1}^m 2\omega_j \left( (\hat{\theta}_j^{\mathrm{PT}})^2 - \hat{\theta}_j^{\mathrm{PT}} \theta_j - \tilde{\sigma}_j^2 \right) \right| \right] \leq \left( \frac{16}{m} \sup_j \mathrm{Var}_{\mathbb{P}_\eta} \left[ (\hat{\theta}_j^{\mathrm{PT}})^2 - \hat{\theta}_j^{\mathrm{PT}} \theta_j \right] \right)^{1/2}. \tag{35}$$

Next, we bound the last expression in (31): $\frac{2}{m} \sup_{\omega \geq 0} \left| \sum_{j=1}^m (1 - \omega_j) \mu_j (\hat{\theta}_j^{\mathrm{PT}} - \theta_j) \right|$. Note that $(1 - \omega_j)$ is also monotonic in $\tilde{\sigma}_j^2$, and if we define $M_k' := \sum_{j=1}^k \mu_j (\hat{\theta}_j^{\mathrm{PT}} - \theta_j)$, then $\{M_k'\}_{k=1}^m$ forms another martingale conditioning on $\boldsymbol{\eta}$. Therefore, following the same argument as (33)–(34) gives

$$\frac{4}{m^2} \mathbb{E}_{\mathbb{P}_\eta} \left[ \sup_{\omega \geq 0} \left| \sum_{j=1}^m (1 - \omega_j) \mu_j (\hat{\theta}_j^{\mathrm{PT}} - \theta_j) \right|^2 \right] \leq \frac{4}{m^2} \mathbb{E}_{\mathbb{P}_\eta} \left[ \max_{1 \leq k \leq m} M_k'^2 \right]$$

$$\leq \frac{16}{m^2} \mathbb{E}_{\mathbb{P}_\eta} \left[ M_m'^2 \right] = \frac{16}{m} \sup_j \mathbb{E}_{\mathbb{P}_\eta} \left[ \mathrm{Var}_{\eta_j} \left[ \hat{\theta}_j^{\mathrm{PT}} \right] \mu_j^2 \right]$$

$$\implies \mathbb{E}_{\mathbb{P}_\eta} \left[ \frac{2}{m} \sup_{\omega \geq 0} \left| \sum_{j=1}^m (1 - \omega_j) \mu_j (\hat{\theta}_j^{\mathrm{PT}} - \theta_j) \right| \right] \leq \left( \frac{16}{m} \sup_j \mathbb{E}_{\mathbb{P}_\eta} \left[ \mathrm{Var}_{\eta_j} \left[ \hat{\theta}_j^{\mathrm{PT}} \right] \mu_j^2 \right] \right)^{1/2}. \tag{36}$$

The upper bounds derived in (32), (35) and (36) establish control on

$$\mathbb{E}_{\mathbb{P}_\eta}\left[\sup_{\omega\geq 0}|\Delta_1(\omega)|\right] \leq \frac{4}{\sqrt{m}}\left(\sup_j \operatorname{Var}_{\mathbb{P}_\eta}\left[(\hat{\theta}_j^{\mathrm{PT}})^2\right]^{1/2} + \sup_j \operatorname{Var}_{\mathbb{P}_\eta}\left[(\hat{\theta}_j^{\mathrm{PT}})^2 - \hat{\theta}_j^{\mathrm{PT}}\theta_j\right] + \sup_j \mathbb{E}_{\mathbb{P}_\eta}\left[\operatorname{Var}_{\eta_j}\left[\hat{\theta}_j^{\mathrm{PT}}\right]\mu_j^2\right]\right),$$

since each term on the right-hand side can be controlled by the fourth-moment conditions established in Lemma F.1, we know that $\Delta_1(\omega)$ converges uniformly to zero.

**Step 3: Bounding the second difference $\Delta_2(\omega) := \mathbb{II}(\omega) - \mathbb{II}^*(\omega)$.** We next cancel out identical terms in the second difference in (30) and get

$$\Delta_2(\omega) = \frac{2}{m}\sum_{j=1}^m (1-\omega_j)\left[\tilde{\gamma}_j - (\tilde{Z}_j^f - \mu_j)(\hat{\theta}_j^{\mathrm{PT}} - \theta_j)\right]. \tag{37}$$

By the same proof logic that has been applied twice above, we now have a function $(1-\omega_j)$ monotonic in $\tilde{\sigma}_j^2$, and a martingale $Q_k := \sum_{j=1}^k \left[\tilde{\gamma}_j - (\tilde{Z}_j^f - \mu_j)(\hat{\theta}_j^{\mathrm{PT}} - \theta_j)\right]$ for $k = 1,\ldots,m$ (recall that $\tilde{\gamma}_j = \operatorname{Cov}_{\eta_j}[\hat{\theta}_j^{\mathrm{PT}}, \tilde{Z}_j^f]$). The steps from (33)–(34) follows, and we have

$$\frac{4}{m^2}\mathbb{E}_{\mathbb{P}_\eta}\left[\left(\sup_{\omega\geq 0}\left|\sum_{j=1}^m (1-\omega_j)\left[\tilde{\gamma}_j - (\tilde{Z}_j^f - \mu_j)(\hat{\theta}_j^{\mathrm{PT}} - \theta_j)\right]\right|\right)^2\right] \leq \frac{16}{m}\sup_j \mathbb{E}_{\mathbb{P}_\eta}\left[\operatorname{Var}_{\eta_j}\left[(\tilde{Z}_j^f - \mu_j)(\hat{\theta}_j^{\mathrm{PT}} - \theta_j)\right]\right],$$

and so,

$$\mathbb{E}_{\mathbb{P}_\eta}\left[\frac{2}{m}\sup_{\omega\geq 0}\left|\sum_{j=1}^m (1-\omega_j)\left[\tilde{\gamma}_j - (\tilde{Z}_j^f - \mu_j)(\hat{\theta}_j^{\mathrm{PT}} - \theta_j)\right]\right|\right] \leq \left(\frac{16}{m}\sup_j \mathbb{E}_{\mathbb{P}_\eta}\left[\operatorname{Var}_{\eta_j}\left[(\tilde{Z}_j^f - \mu_j)(\hat{\theta}_j^{\mathrm{PT}} - \theta_j)\right]\right]\right)^{1/2}.$$

Again, the (fourth-)moment conditions from Lemma F.1 suffice to ensure that

$$\sup_j \mathbb{E}_{\mathbb{P}_\eta}\left[\operatorname{Var}_{\eta_j}\left[(\tilde{Z}_j^f - \mu_j)(\hat{\theta}_j^{\mathrm{PT}} - \theta_j)\right]\right] < \infty,$$

and to establish control of

$$\mathbb{E}_{\mathbb{P}_\eta}\left[\sup_{\omega\geq 0}|\Delta_2(\omega)|\right].$$

**Step 4: Concluding the argument.** Finally, based on Steps 1–3, we have that

$$\mathbb{E}_{\mathbb{P}_\eta}\left[\sup_{\omega\geq 0}\left|\mathrm{CURE}(\hat{\boldsymbol{\theta}}_\omega^{\mathrm{PAS}}) - \ell_m(\hat{\boldsymbol{\theta}}_\omega^{\mathrm{PAS}}, \boldsymbol{\theta})\right|\right] \leq \mathbb{E}_{\mathbb{P}_\eta}\left[\sup_{\omega\geq 0}|\Delta_1(\omega)|\right] + \mathbb{E}_{\mathbb{P}_\eta}\left[\sup_{\omega\geq 0}|\Delta_2(\omega)|\right],$$

and both terms on the right hand side converge to zero by our preceding bounds and the moment assumptions in the statement of the theorem. $\square$

### F.3. Proof of Theorem 5.2

We apply a standard argument used to prove consistency of M-estimators.

Let $\omega_*$ be the oracle choice of $\omega \geq 0$ that minimizes the Bayes risk $\mathcal{B}_m^{\mathbb{P}_\eta}(\hat{\boldsymbol{\theta}}_\omega^{\mathrm{PAS}})$.[26] Notice that by definition of $\hat{\omega}$ as the minimizer of CURE,

$$\mathrm{CURE}(\hat{\boldsymbol{\theta}}_{\hat{\omega}}^{\mathrm{PAS}}) \leq \mathrm{CURE}(\hat{\boldsymbol{\theta}}_{\omega_*}^{\mathrm{PAS}}).$$

Then:

$$\ell_m(\hat{\boldsymbol{\theta}}_{\hat{\omega}}^{\mathrm{PAS}}, \boldsymbol{\theta}) - \ell_m(\hat{\boldsymbol{\theta}}_{\omega_*}^{\mathrm{PAS}}, \boldsymbol{\theta}) \leq 2\sup_{\omega\geq 0}\left|\mathrm{CURE}(\hat{\boldsymbol{\theta}}_\omega^{\mathrm{PAS}}) - \ell_m(\hat{\boldsymbol{\theta}}_\omega^{\mathrm{PAS}}, \boldsymbol{\theta})\right|.$$

---

[26]To streamline the proof, we assume that the infimum is attained by a value $\omega_*$. If the infimum is not attained, the proof still goes through using approximate minimizers.

Taking expectations,

$$\mathcal{B}_m^{\mathbb{P}_\eta}(\hat{\boldsymbol{\theta}}_{\hat{\omega}}^{\mathrm{PAS}}) - \mathcal{B}_m^{\mathbb{P}_\eta}(\hat{\boldsymbol{\theta}}_{\omega_*}^{\mathrm{PAS}}) \leq 2\,\mathbb{E}_{\mathbb{P}_\eta}\left[\sup_{\omega \geq 0}\left|\mathrm{CURE}(\hat{\boldsymbol{\theta}}_\omega^{\mathrm{PAS}}) - \ell_m(\hat{\boldsymbol{\theta}}_\omega^{\mathrm{PAS}}, \boldsymbol{\theta})\right|\right].$$

Noting that the right hand side converges to $0$ as $m \to \infty$, and recalling the definition of $\omega_*$, we prove the desired result

$$\mathcal{B}_m^{\mathbb{P}_\eta}(\hat{\boldsymbol{\theta}}_{\hat{\omega}}^{\mathrm{PAS}}) \leq \inf_{\omega \geq 0} \mathcal{B}_m^{\mathbb{P}_\eta}(\hat{\boldsymbol{\theta}}_\omega^{\mathrm{PAS}}) + o(1).$$

$\square$

## F.4. Proof of Proposition 5.3

We start with the result in Theorem 5.2

$$\mathcal{B}_m^{\mathbb{P}_\eta}(\hat{\boldsymbol{\theta}}_{\hat{\omega}}^{\mathrm{PAS}}, \boldsymbol{\theta}) \leq \inf_{\omega \geq 0} \mathcal{B}_m^{\mathbb{P}_\eta}(\hat{\boldsymbol{\theta}}_\omega^{\mathrm{PAS}}, \boldsymbol{\theta}) + o(1).$$

Now, since we are integrating over $\eta \sim \mathbb{P}_\eta$ for all problems

$$\mathcal{B}_m^{\mathbb{P}_\eta}(\hat{\boldsymbol{\theta}}_\omega^{\mathrm{PAS}}, \boldsymbol{\theta}) = \frac{1}{m}\sum_{j=1}^m \mathbb{E}_{\mathbb{P}_\eta}\left[(\hat{\theta}_{j,\omega}^{\mathrm{PAS}} - \theta_j)^2\right]$$

$$= \mathbb{E}_{\mathbb{P}_\eta}\left[(\hat{\theta}_{j,\omega}^{\mathrm{PAS}} - \theta_j)^2\right],$$

by definition $\hat{\theta}_{j,\omega}^{\mathrm{PAS}} = \omega_j \hat{\theta}_j^{\mathrm{PT}} + (1 - \omega_j)\tilde{Z}_j^f$, where $\omega_j = \omega/(\omega + \tilde{\sigma}^2)$. Therefore

$$\mathbb{E}_{\mathbb{P}_\eta}\left[(\hat{\theta}_{j,\omega}^{\mathrm{PAS}} - \theta_j)^2\right] = \mathbb{E}_{\mathbb{P}_\eta}\left[\left(\omega_j(\hat{\theta}_j^{\mathrm{PT}} - \theta_j) + (1 - \omega_j)(\tilde{Z}_j^f - \theta_j)\right)^2\right]$$

$$= \frac{\omega^2}{(\omega + \tilde{\sigma}^2)^2}\mathbb{E}_{\mathbb{P}_\eta}\left[(\hat{\theta}_j^{\mathrm{PT}} - \theta_j)^2\right] + \frac{\tilde{\sigma}^4}{(\omega + \tilde{\sigma}^2)^2}\mathbb{E}_{\mathbb{P}_\eta}\left[(\tilde{Z}_j^f - \theta_j)^2\right]$$

$$+ 2\frac{\tilde{\sigma}^2\omega}{(\omega + \tilde{\sigma}^2)^2}\mathbb{E}_{\mathbb{P}_\eta}\left[(\hat{\theta}_j^{\mathrm{PT}} - \theta_j)(\tilde{Z}_j^f - \theta_j)\right].$$

By our assumption, second moment terms like $\tilde{\sigma}^2$ and $\tilde{\gamma}$ are now fixed, so we have (by iterated expectation)

$$\mathbb{E}_{\mathbb{P}_\eta}\left[(\hat{\theta}_j^{\mathrm{PT}} - \theta_j)^2\right] = \tilde{\sigma}^2.$$

Noting that $\tilde{\gamma}_j = 0$ since $N_j = \infty$, we have

$$\mathbb{E}_{\mathbb{P}_\eta}\left[(\hat{\theta}_{j,\omega}^{\mathrm{PAS}} - \theta_j)^2\right] = \frac{\omega^2\tilde{\sigma}^2}{(\omega + \tilde{\sigma}^2)^2} + \frac{\tilde{\sigma}^4}{(\omega + \tilde{\sigma}^2)^2}\mathbb{E}_{\mathbb{P}_\eta}\left[(\tilde{Z}_j^f - \theta_j)^2\right].$$

Plugging in $\omega = \mathbb{E}_{\mathbb{P}_\eta}\left[(\tilde{Z}_j^f - \theta_j)^2\right]$ gives

$$\frac{\omega^2\tilde{\sigma}^2}{(\omega + \tilde{\sigma}^2)^2} + \frac{\tilde{\sigma}^4}{(\omega + \tilde{\sigma}^2)^2}\mathbb{E}_{\mathbb{P}_\eta}\left[(\tilde{Z}_j^f - \theta_j)^2\right] = \frac{\tilde{\sigma}^2\mathbb{E}_{\mathbb{P}_\eta}\left[(\tilde{Z}_j^f - \theta_j)^2\right]}{\tilde{\sigma}^2 + \mathbb{E}_{\mathbb{P}_\eta}\left[(\tilde{Z}_j^f - \theta_j)^2\right]}.$$

We finally have

$$\mathcal{B}_m^{\mathbb{P}_\eta}(\hat{\boldsymbol{\theta}}^{\mathrm{PAS}}) \leq \frac{\tilde{\sigma}^2\mathbb{E}_{\mathbb{P}_\eta}\left[(\tilde{Z}_j^f - \theta_j)^2\right]}{\tilde{\sigma}^2 + \mathbb{E}_{\mathbb{P}_\eta}\left[(\tilde{Z}_j^f - \theta_j)^2\right]} + o(1).$$

$\square$

## F.5. Proof of Lemma C.2

We aim to prove that $\mathbb{E}_{\mathbb{P}_\eta}\left[(\hat{\lambda}_{\mathrm{clip}} - \lambda^*_{\mathrm{clip},m})^2\right] \to 0$ as $m \to \infty$. Let $\lambda^*_m$ be the unclipped theoretical optimal global parameter for $m$ problems, and $\hat{\lambda}$ be its unclipped sample-based estimator:

$$\lambda^*_m = \frac{\sum_{j=1}^m n_j^{-1}\gamma_j}{\sum_{j=1}^m \frac{n_j+N_j}{n_j N_j}\tau_j^2} \quad \text{and} \quad \hat{\lambda} = \frac{\sum_{j=1}^m n_j^{-1}\hat{\gamma}_j}{\sum_{j=1}^m \frac{n_j+N_j}{n_j N_j}\hat{\tau}_j^2}.$$

The clipped versions are $\lambda^*_{\mathrm{clip},m} = \mathrm{clip}(\lambda^*_m, [0,1])$ and $\hat{\lambda}_{\mathrm{clip}} = \mathrm{clip}(\hat{\lambda}, [0,1])$.

**Step 1: Convergence in probability of the numerator and the denominator.** Let:

$$N^*_m = \frac{1}{m}\sum_{j=1}^m n_j^{-1}\gamma_j, \qquad\qquad D^*_m = \frac{1}{m}\sum_{j=1}^m \frac{n_j+N_j}{n_j N_j}\tau_j^2, \qquad (38)$$

$$\hat{N}_m = \frac{1}{m}\sum_{j=1}^m n_j^{-1}\hat{\gamma}_j, \qquad\qquad \hat{D}_m = \frac{1}{m}\sum_{j=1}^m \frac{n_j+N_j}{n_j N_j}\hat{\tau}_j^2. \qquad (39)$$

So $\lambda^*_m = N^*_m/D^*_m$ and $\hat{\lambda} = \hat{N}_m/\hat{D}_m$. We denote $\Delta N_m = \hat{N}_m - N^*_m = m^{-1}\sum_{j=1}^m U_j$ where $U_j = n_j^{-1}(\hat{\gamma}_j - \gamma_j)$. Note that $\mathbb{E}_{\mathbb{P}_\eta}[U_j] = 0$, and by assumption 2 and 3 we know that $\mathrm{Var}_{\mathbb{P}_\eta}[U_j] = \mathbb{E}_{\mathbb{P}_\eta}[n_j^{-2}\mathrm{Var}_{\eta_j}[\hat{\gamma}_j]]$ is bounded (say $< V_U < \infty$), we then have

$$\mathbb{E}_{\mathbb{P}_\eta}[\Delta N_m] = 0, \quad \mathrm{Var}_{\mathbb{P}_\eta}[\Delta N_m] \to 0 \quad \text{as} \quad m \to \infty.$$

So $\Delta N_m \xrightarrow{L^2} 0$, which implies $\Delta N_m \xrightarrow{P} 0$. Thus, $\hat{N}_m - N^*_m \xrightarrow{P} 0$. An identical proof will also give us $\hat{D}_m - D^*_m \xrightarrow{P} 0$ for the denominator terms.

**Step 2: Convergence in probability of unclipped ratio.** By assumption 5, we have $D^*_m$ (and $\hat{D}_m$ as well since $\hat{D}_m = D^*_m + o_P(1)$) bounded away from zero in probability. Thus, by standard argument using the Continuous Mapping Theorem, we have

$$\hat{\lambda} - \lambda^*_m = \frac{N^*_m + o_P(1)}{D^*_m + o_P(1)} - \frac{N^*_m}{D^*_m} \xrightarrow{P} 0.$$

for the unclipped ratio.

**Step 2: $L^2$ convergence for the clipped ratio.** The clipping function $g(x) = \mathrm{clip}(x, [0,1])$ is again continuous. Re-applying the Continuous Mapping Theorem gives

$$\hat{\lambda}_{\mathrm{clip}} - \lambda^*_{\mathrm{clip},m} = \mathrm{clip}(\hat{\lambda}, [0,1]) - \mathrm{clip}(\lambda^*_m, [0,1]) \xrightarrow{P} 0.$$

Finally, the clipping also makes sure that $|\hat{\lambda}_{\mathrm{clip}} - \lambda^*_{\mathrm{clip},m}| \leq 1$. This then leads to the stronger convergence result $\hat{\lambda}_{\mathrm{clip}} - \lambda^*_{\mathrm{clip},m} \xrightarrow{L^2} 0$ and completes the proof. $\qquad\square$

## F.6. Proof of Theorem C.3

Denote $V_j(\lambda) = \mathrm{Var}_{\eta_j}[\hat{\theta}_{j,\lambda}] = \frac{\sigma_j^2}{n_j} - \frac{2\lambda}{n_j}\gamma_j + \lambda^2\frac{n_j+N_j}{n_j N_j}\tau_j^2$. Let $S_m(\lambda) = \sum_{j=1}^m V_j(\lambda)$. The average sum of variances is $V_m(\lambda) = \frac{1}{m}S_m(\lambda)$. We can express $V_m(\lambda)$ as a quadratic function of $\lambda$: $V_m(\lambda) = C_m - 2B_m\lambda + A_m\lambda^2$, where

$$A_m = \frac{1}{m}\sum_{j=1}^m \frac{n_j+N_j}{n_j N_j}\tau_j^2 \quad (= D^*_m \text{ from Equation (38)})$$

$$B_m = \frac{1}{m}\sum_{j=1}^m \frac{\gamma_j}{n_j} \quad (= N^*_m \text{ from Equation (38)})$$

$$C_m = \frac{1}{m}\sum_{j=1}^m \frac{\sigma_j^2}{n_j}.$$

The minimizer of $V_m(\lambda)$ over $\lambda \in \mathbb{R}$ is $\lambda_u = B_m/A_m$ (assuming $A_m > 0$, which holds if not all $\tau_j^2 = 0$). Since $V_m(\lambda)$ is a quadratic function in $\lambda$ with a positive leading coefficient $A_m$ (i.e., an upward-opening parabola), its minimum over the closed interval $[0, 1]$ is achieved at $\lambda^*_{\text{clip},m} = \text{clip}(\lambda_u, [0, 1]) = \text{clip}(B_m/A_m, [0, 1])$.

We want to show that $V_m(\hat{\lambda}_{\text{clip}}) - V_m(\lambda^*_{\text{clip},m}) \xrightarrow{P} 0$. The derivative of $V_m(\lambda)$ is $V'_m(\lambda) = 2A_m\lambda - 2B_m$. By the Mean Value Theorem, for some $\xi_m$ between $\hat{\lambda}_{\text{clip}}$ and $\lambda^*_{\text{clip},m}$:

$$V_m(\hat{\lambda}_{\text{clip}}) - V_m(\lambda^*_{\text{clip},m}) = V'_m(\xi_m)(\hat{\lambda}_{\text{clip}} - \lambda^*_{\text{clip},m})$$

$$= (2A_m\xi_m - 2B_m)(\hat{\lambda}_{\text{clip}} - \lambda^*_{\text{clip},m})$$

Since $\hat{\lambda}_{\text{clip}}$ and $\lambda^*_{\text{clip},m}$ are both in $[0, 1]$, $\xi_m$ is also in $[0, 1]$. The terms $A_m = D_m^*$ and $B_m = N_m^*$ are averages of $m$ independent terms with uniformly bounded variances (under Assumption 4 of Lemma C.2). Thus, by Chebyshev's inequality, $A_m = O_P(1)$ and $B_m = O_P(1)$ (i.e., they are bounded in probability). Since $\xi_m \in [0, 1]$, the term $(2A_m\xi_m - 2B_m)$ is also $O_P(1)$. Let $\Delta\lambda_m = \hat{\lambda}_{\text{clip}} - \lambda^*_{\text{clip},m}$. From Lemma C.2, we have $\Delta\lambda_m \xrightarrow{P} 0$. Therefore,

$$V_m(\hat{\lambda}_{\text{clip}}) - V_m(\lambda^*_{\text{clip},m}) = (2A_m\xi_m - 2B_m)\Delta\lambda_m \xrightarrow{P} 0.$$

The $L^1$ convergence then follows from the fact that This establishes the asymptotic variance optimality of the `UniPT` estimator. $\square$

### F.7. Proof of Proposition C.5

**Organization of the proof.** We provide a high-level sketch of the proof in Figure 7. Our main goal is to establish that $\widehat{\text{CURE}} - \ell_m \xrightarrow{L^1} 0$ uniformly in $\omega$ as $m \to \infty$. To achieve this, we introduce two intermediate estimators—CURE$'$ and CURE$''$—that serve as bridges between $\widehat{\text{CURE}}$ and $\ell_m$. The proof proceeds by showing that each consecutive pair of estimators is asymptotically close, which allows us to conclude the overall result via repeated applications of the triangle inequality.

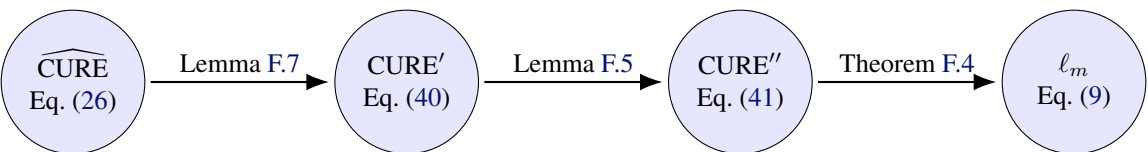

*Figure 7.* A visual sketch of the proof. Each node represents a variant of the original CURE (i.e. a risk estimator conditioned on $\omega$); each arrow then represents an asymptotic ($m \to \infty$) closeness result between the estimators.

We first define two intermediate forms of CURE, denoted as CURE$'$ and CURE$''$ respectively, between the original CURE (Eq. 24) that requires full knowledge about second moments and the sample estimate-based $\widehat{\text{CURE}}$ defined in Equation (26):

$$\text{CURE}'(\hat{\boldsymbol{\theta}}_\omega^{\text{UPAS}}) = \frac{1}{m}\sum_{j=1}^{m}\left[(2\omega_j^\circ - 1)\dot{\sigma}_j^2 + 2(1 - \omega_j^\circ)\dot{\gamma}_j + (1 - \omega_j^\circ)^2(\hat{\theta}_j^{\text{UPT}} - \bar{Z}_j^f)^2\right], \quad \omega_j^\circ := \frac{\omega}{\omega + \mathring{\sigma}_j^2}, \qquad (40)$$

with $\mathring{\sigma}_j^2$ being the variance target defined in Equation (27).[27] In other words, we can treat $\mathring{\sigma}_j^2$ as a fixed (but unknown) plug-in value so $\omega_j^\circ$ is non-random for all $\omega > 0$. Next we define CURE$''$ by replacing the $\dot{\sigma}_j^2$ and $\dot{\gamma}_j$ terms in CURE$'$—which in turn depend on the estimated UniPT parameter $\hat{\lambda}_{\text{clip}}$—with variants that depend on $\lambda^*_{\text{clip},m}$ instead:

$$\text{CURE}''(\hat{\boldsymbol{\theta}}_\omega^{\text{UPAS}}) = \frac{1}{m}\sum_{j=1}^{m}\left[(2\omega_j^\circ - 1)\ddot{\sigma}_j^2 + 2(1 - \omega_j^\circ)\ddot{\gamma}_j + (1 - \omega_j^\circ)^2(\hat{\theta}_j^{\text{UPT}} - \bar{Z}_j^f)^2\right], \qquad (41)$$

$$\text{where} \quad \ddot{\sigma}_j^2 = \frac{\hat{\sigma}_j^2}{n_j} + \frac{N_j + n_j}{N_j n_j}(\lambda^*_{\text{clip},m})^2\hat{\tau}_j^2 - \frac{2}{n_j}\lambda^*_{\text{clip},m}\hat{\gamma}_j, \quad \ddot{\gamma}_j = \lambda^*_{\text{clip},m}\frac{\hat{\tau}_j^2}{N_j}.$$

CURE$''$ possesses the following desirable properties:

---

[27]We omit the dependency on $m$ by using the shorthand $\mathring{\sigma}_j^2 \equiv \mathring{\sigma}_{j,m}^2$ introduced in (27).

**Lemma F.3.** $\mathrm{CURE}''(\hat{\boldsymbol{\theta}}_\omega^{\mathrm{UPAS}})$ *is an unbiased estimator of* $\mathcal{R}_m(\hat{\boldsymbol{\theta}}_\omega^{\mathrm{UPAS}})$ *conditioning on* $\eta_j$.

*Proof.* This is a direct result of Theorem B.1 and $\ddot{\sigma}_j^2, \ddot{\gamma}_j$ being unbiased estimators by construction. □

**Theorem F.4** (Asymptotic Consistency of $\mathrm{CURE}''$). *Under the assumption of Proposition C.5,*

$$\mathbb{E}_{\mathbb{P}_\eta}\left[\sup_{\omega \geq 0} |\mathrm{CURE}''(\hat{\boldsymbol{\theta}}_\omega^{\mathrm{UPAS}}) - l_m(\hat{\boldsymbol{\theta}}_\omega^{\mathrm{UPAS}}, \boldsymbol{\theta})|\right] \xrightarrow{m \to \infty} 0.$$

*Proof.* For simplicity, hereafter we drop the notations' dependencies on $\hat{\boldsymbol{\theta}}_\omega^{\mathrm{UPAS}}$. By triangle inequality,

$$\mathbb{E}_{\mathbb{P}_\eta}\left[\sup_{\omega \geq 0} |\mathrm{CURE}'' - l_m|\right] \leq \mathbb{E}_{\mathbb{P}_\eta}\left[\sup_{\omega \geq 0} |\mathrm{CURE} - l_m|\right] + \mathbb{E}_{\mathbb{P}_\eta}\left[\sup_{\omega \geq 0} |\mathrm{CURE}'' - \mathrm{CURE}|\right], \tag{42}$$

and we remark here that the CURE referred in Equation (42) differs slightly from the CURE for PAS in Theorem 4.1 as we replace $\tilde{\sigma}_j^2$ with $\mathring{\sigma}_j^2$ in the definition of its $\omega_j$ (same as $\omega_j^\circ$).[28] But since $\mathring{\sigma}_j^2$ is a fixed plug-in value, we can still use Theorem 4.1 to bound the first term on the RHS of (42), so the only task here is to bound the second term as well.

Let $A_j = \ddot{\sigma}_j^2 - \tilde{\sigma}_j^2$ and $B_j = \ddot{\gamma}_j - \tilde{\gamma}_j$. By assumption, we have $\mathbb{E}_{\eta_j}[A_j] = 0$, $\mathbb{E}_{\eta_j}[B_j] = 0$, $\mathbb{E}_{\eta_j}[A_j^2] \leq V_\sigma < \infty$ and $\mathbb{E}_{\eta_j}[B_j^2] \leq V_\gamma < \infty$. Let $c_j(\omega) = 2\omega_j - 1$ and $d_j(\omega) = 2(1 - \omega_j)$, we have

$$\mathrm{CURE}'' - \mathrm{CURE} = \frac{1}{m}\sum_{j=1}^m (\mathrm{CURE}_j'' - \mathrm{CURE}_j) = \frac{1}{m}\sum_{j=1}^m (c_j(\omega)A_j + d_j(\omega)B_j).$$

Now we can apply triangle inequality again. Since both $c_j(\omega)$ and $d_j(\omega)$ are monotone functions of $\omega$ (and supported on $[0,1]$), we can use the same strategies (constructing martingale and applying maximal inequality) as in the proof of Proposition 5.1 Let $M_k' = \sum_{j=1}^k A_j$, we have

$$\left|\sum_{j=1}^m c_j(\omega)A_j\right| = \left|\sum_{k=1}^{m-1} (c_k(\omega) - c_{k+1}(\omega))M_k' + c_m(\omega)M_m'\right|$$
$$\leq (1 + |c_m(\omega)|) \max_{1 \leq k \leq m} |M_k'| \leq 2 \max_{1 \leq k \leq m} |M_k'| \tag{43}$$

Now taking expectation over $\mathbb{P}_\eta$ and use the maximal inequality gives

$$\mathbb{E}_{\mathbb{P}_\eta}\left[\left(\max_{1 \leq k \leq m} |M_k'|\right)^2\right] \leq 4\sum_{j=1}^m \mathbb{E}_{\mathbb{P}_\eta}[A_j^2] \leq 4mV_\sigma$$

Finally, by Jensen's inequality, we have

$$\mathbb{E}_{\mathbb{P}_\eta}\left[\sup_{\omega \geq 0} \left|\frac{1}{m}\sum_{j=1}^m c_j(\omega)A_j\right|\right] \leq \mathbb{E}_{\mathbb{P}_\eta}\left[\frac{1}{m}\sup_{\omega \geq 0}\left|\sum_{j=1}^m c_j(\omega)A_j\right|\right]$$

$$\leq \mathbb{E}_{\mathbb{P}_\eta}\left[\frac{2}{m}\max_{1 \leq k \leq m} |M_k'|\right] = \frac{2}{m}\mathbb{E}_{\mathbb{P}_\eta}\left[\max_{1 \leq k \leq m} |M_k'|\right] \leq \frac{2}{m}2\sqrt{mV_\sigma} = \frac{4\sqrt{V_\sigma}}{\sqrt{m}} \to 0$$

Similar to Equation (43), we can show that

$$\left|\sum_{j=1}^m d_j(\omega)B_j\right| \leq 2 \max_{1 \leq k \leq m} |N_k'|$$

---

[28]In other parts of this CURE, however, we keep using $\tilde{\sigma}_j^2 := \mathrm{Var}\left[\hat{\theta}_{\lambda_{\mathrm{clip},m}^*}^{\mathrm{UPT}}\right]$ and $\tilde{\gamma}_j := \mathrm{Cov}\left[\hat{\theta}_{j,\lambda_{\mathrm{clip},m}^*}^{\mathrm{UPT}}, \tilde{Z}_j^f\right]$, which are also the limits of $\ddot{\sigma}_j^2$ and $\ddot{\gamma}_j$ as $m \to \infty$.

where $N_k' = \sum_{j=1}^{k} B_j$. Again by applying maximal inequality,

$$\mathbb{E}_{\mathbb{P}_\eta}\left[\left(\max_{1\le k\le m}|N_k'|\right)^2\right] \le 4mV_\gamma.$$

Finally

$$\mathbb{E}_{\mathbb{P}_\eta}\left[\sup_{\omega\ge 0}\left|\frac{1}{m}\sum_{j=1}^{m}d_j(\omega)B_j\right|\right] \le \mathbb{E}_{\mathbb{P}_\eta}\left[\frac{2}{m}\max_{1\le k\le m}|N_k'|\right] = \frac{2}{m}2\sqrt{mV_\gamma} = \frac{4\sqrt{V_\gamma}}{\sqrt{m}} \to 0$$

as $m\to\infty$. Combining everything together gives

$$\mathbb{E}_{\mathbb{P}_\eta}\left[\sup_{\omega\ge 0}|\text{CURE}'' - \text{CURE}|\right] \le \mathbb{E}_{\mathbb{P}_\eta}\left[\sup_{\omega\ge 0}\left|\frac{1}{m}\sum c_j A_j\right|\right] + \mathbb{E}_{\mathbb{P}_\eta}\left[\sup_{\omega\ge 0}\left|\frac{1}{m}\sum d_j B_j\right|\right]$$
$$\le O(1/\sqrt{m}) + O(1/\sqrt{m}) \to 0$$

$\square$

CURE$''$ retains the nice asymptotic properties of CURE, but it consists of $\ddot{\sigma}_j^2$ and $\ddot{\gamma}_j$ that we cannot evaluate from data (due to the unknown $\lambda^*_{\text{clip},m}$). Our next lemma derives the asymptotic closeness between CURE$''$ and CURE$'$, where the definition of the latter quantity is one step closer to the fully data-driven $\widehat{\text{CURE}}$.

**Lemma F.5** (Closeness between CURE$'$ and CURE$''$). *Under the assumption of Proposition C.5, we have*

$$\mathbb{E}_{\mathbb{P}_\eta}\left[\sup_{\omega\ge 0}|\text{CURE}'(\hat{\boldsymbol{\theta}}_\omega^{\text{UPAS}}) - \text{CURE}''(\hat{\boldsymbol{\theta}}_\omega^{\text{UPAS}})|\right] \xrightarrow{m\to\infty} 0$$

*Proof.* By construction, CURE$'$ and CURE$''$ only differ in their use of $\dot{\sigma}_j^2$ versus $\ddot{\sigma}_j^2$ (similarly for $\dot{\gamma}_j$ v.s. $\ddot{\gamma}_j$). We can thus decompose their difference as

$$\sup_{\omega>0}\left|\text{CURE}' - \text{CURE}''\right| = \sup_{\omega>0}\left|\frac{1}{m}\sum_{j=1}^{m}\left[(2\omega_j^\circ - 1)(\dot{\sigma}_j^2 - \ddot{\sigma}_j^2) + 2(1-\omega_j^\circ)(\dot{\gamma}_j - \ddot{\gamma}_j)\right]\right|.$$
$$\le \frac{1}{m}\sum_{j=1}^{m}\sup_{\omega>0}\left|(2\omega_j^\circ - 1)(\dot{\sigma}_j^2 - \ddot{\sigma}_j^2)\right| + \frac{1}{m}\sum_{j=1}^{m}\sup_{\omega>0}\left|2(1-\omega_j^\circ)(\dot{\gamma}_j - \ddot{\gamma}_j)\right|$$
$$\le \frac{1}{m}\sum_{j=1}^{m}\left|\dot{\sigma}_j^2 - \ddot{\sigma}_j^2\right| + \frac{2}{m}\sum_{j=1}^{m}|\dot{\gamma}_j - \ddot{\gamma}_j| \tag{44}$$

since $\sup_\omega|2\omega_j^\circ - 1|\le 1$ and $\sup_\omega|2(1-\omega_j^\circ)|\le 2$. Since we have $\hat{\lambda}_{\text{clip}} \to \lambda^*_{\text{clip},m}$ (and $\hat{\lambda}^2_{\text{clip}} \to (\lambda^*_{\text{clip},m})^2$)[29] in $L^2$, and by our fourth-moment assumptions we have $\mathbb{E}_{\mathbb{P}_\eta}[\hat{\sigma}_j^2]$, $\mathbb{E}_{\mathbb{P}_\eta}[\hat{\tau}_j^2]$, and $\mathbb{E}_{\mathbb{P}_\eta}[\hat{\gamma}_j]$ all finite, by construction we also have

$$\sup_j \mathbb{E}_{\mathbb{P}_\eta}\left[(\dot{\sigma}_j^2 - \ddot{\sigma}_j^2)^2\right] \xrightarrow{m\to\infty} 0 \quad \text{and} \quad \sup_j \mathbb{E}_{\mathbb{P}_\eta}\left[(\dot{\gamma}_j - \ddot{\gamma}_j)^2\right] \xrightarrow{m\to\infty} 0.$$

These uniform $L^2$ convergence conditions are sufficient to make sure the expectation of Equation (44) converges to 0, and we thus prove our lemma. $\square$

At this point, we note that the only intractable piece in CURE$'$ is the unknown variance target $\mathring{\sigma}_j^2$, which is used for constructing the weights $\omega_j^\circ$. $\widehat{\text{CURE}}$ then operationalize CURE$'$ by replacing $\mathring{\sigma}_j^2$ with the sample-based $\breve{\sigma}_j^2$. Our next lemma shows that $\breve{\sigma}_j^2 \to \mathring{\sigma}_j^2$ uniformly in $L^2$ as $m\to\infty$, which then leads to our final closeness result between $\widehat{\text{CURE}}$ and CURE$'$.

---

[29]We defer the proof of this convergence result to Lemma F.6.

**Lemma F.6** (Uniform $L^2$ convergence of $\check{\sigma}_j^2$). *Under the assumption of Proposition C.5, we have*

$$\sup_j \mathbb{E}_{\mathbb{P}_\eta}\left[(\check{\sigma}_j^2 - \mathring{\sigma}_j^2)^2\right] \xrightarrow{m \to \infty} 0$$

*Proof.* We can decompose the difference as

$$\check{\sigma}_j^2 - \mathring{\sigma}_j^2 = \underbrace{\frac{1}{n_j}(\bar{\sigma}^2 - \mu_{\sigma^2}) + \frac{N_j + n_j}{N_j n_j}\hat{\lambda}_{\text{clip}}^2(\bar{\tau}^2 - \mu_{\tau^2}) - \frac{2}{n_j}\hat{\lambda}_{\text{clip}}(\bar{\gamma} - \mu_\gamma)}_{(*)}$$

$$+ \underbrace{\frac{N_j + n_j}{N_j n_j}\mu_{\tau^2}(\hat{\lambda}_{\text{clip}}^2 - (\lambda_{\text{clip},m}^*)^2) - \frac{2}{n_j}\mu_\gamma(\hat{\lambda}_{\text{clip}} - \lambda_{\text{clip},m}^*)}_{(**)}.$$

**Bounding $(*)$:** Using the inequality $(a + b + c)^2 \leq 3(a^2 + b^2 + c^2)$,

$$\mathbb{E}_{\mathbb{P}_\eta}[(*)^2] \leq 3\left(c_1^2 \mathbb{E}_{\mathbb{P}_\eta}[(\bar{\sigma}^2 - \mu_{\sigma^2})^2] + c_2^2 \mathbb{E}_{\mathbb{P}_\eta}[(\bar{\tau}^2 - \mu_{\tau^2})^2] + c_3^2 \mathbb{E}_{\mathbb{P}_\eta}[(\bar{\gamma} - \mu_\gamma)^2]\right) \tag{45}$$

where $c_1, c_2, c_3$ are some bounded terms involving $n_j, N_j, \hat{\lambda}_{\text{clip}}$ (note that $\hat{\lambda}_{\text{clip}}$ is clipped and bounded). Now, by our assumptions (esp. the finiteness of fourth-moments) and the Weak Law of Large Numbers, as $m \to \infty$ we have

$$\bar{\sigma}^2 \xrightarrow{P} \mathbb{E}_{\mathbb{P}_\eta}[\sigma_j^2] =: \mu_{\sigma^2}, \quad \bar{\tau}^2 \xrightarrow{P} \mathbb{E}_{\mathbb{P}_\eta}[\tau_j^2] =: \mu_{\tau^2}, \quad \bar{\gamma} \xrightarrow{P} \mathbb{E}_{\mathbb{P}_\eta}[\gamma_j] =: \mu_\gamma,$$

then by our assumption (again by finite fourth moments in data generating process),

$$\mathbb{E}_{\mathbb{P}_\eta}[(\bar{\sigma}^2 - \mu_{\sigma^2})^2] = \text{Var}_{\mathbb{P}_\eta}[\bar{\sigma}^2] = \frac{1}{m}\text{Var}_{\mathbb{P}_\eta}[\hat{\sigma}_j^2] = O(1/m),$$

similarly, we know that the other two terms in the RHS of Equation (45) are also $O(1/m)$. Moreover, since all the problems are iid, these rates remain valid even if we take the supremum over $j \in [m]$. We thus have

$$\sup_j \mathbb{E}_{\mathbb{P}_\eta}[(*)^2] = O(1/m) = o(1).$$

**Bounding $(**)$:** Similarly, we use the inequality $(a + b)^2 \leq 2a^2 + 2b^2$ to get

$$\mathbb{E}_{\mathbb{P}_\eta}[(**)^2] \leq 2\left(k_1^2 \mathbb{E}_{\mathbb{P}_\eta}\left[\left((\hat{\lambda}_{\text{clip}}^2 - (\lambda_{\text{clip},m}^*)^2)^2\right)\right] + k_2^2 \mathbb{E}_{\mathbb{P}_\eta}\left[(\hat{\lambda}_{\text{clip}} - \lambda_{\text{clip},m}^*)^2\right]\right) \tag{46}$$

where $k_1, k_2$ are some bounded terms involving $n_j, N_j, \mu_{\tau^2}, \mu_\gamma$. By Proposition C.2, we already know that $\hat{\lambda}_{\text{clip}} \to \lambda_{\text{clip},m}^*$ in $L^2$. Further, since both $\hat{\lambda}_{\text{clip}}$ and $\lambda_{\text{clip},m}^*$ are bounded within $[0, 1]$, thus

$$\left((\hat{\lambda}_{\text{clip}}^2 - (\lambda_{\text{clip},m}^*)^2)\right)^2 = (\hat{\lambda}_{\text{clip}} - \lambda_{\text{clip},m}^*)^2(\hat{\lambda}_{\text{clip}} + \lambda_{\text{clip},m}^*)^2$$

$$\leq 2(\hat{\lambda}_{\text{clip}} - \lambda_{\text{clip},m}^*)^2$$

so we know that $\hat{\lambda}_{\text{clip}}^2 \to (\lambda_{\text{clip},m}^*)^2$ in $L^2$ as well. We thus have $\sup_j \mathbb{E}_{\mathbb{P}_\eta}[(**)^2] = o(1)$.

**Putting $(*)$ and $(**)$ together.** Since $(\check{\sigma}_j^2 - \mathring{\sigma}_j^2)^2 \leq 2(*)^2 + 2(**)^2$, the above results show us that

$$\sup_j \mathbb{E}_{\mathbb{P}_\eta}\left[(\check{\sigma}_j^2 - \mathring{\sigma}_j^2)^2\right] \leq 2\sup_j \mathbb{E}_{\mathbb{P}_\eta}[(*)^2] + 2\sup_j \mathbb{E}_{\mathbb{P}_\eta}[(**)^2] = o(1)$$

and we obtain the uniform convergence in $L^2$. $\quad\square$

With Lemma F.6, we can derive an asymptotic closeness result between CURE$'$ and the new $\widehat{\text{CURE}}$ that we can actually calculate from data.

**Lemma F.7** (Closeness between $\widehat{\text{CURE}}$ and $\text{CURE}'$). *Under the assumption of Proposition C.5, we have*

$$\mathbb{E}_{\mathbb{P}_\eta}\left[\sup_{\omega\geq 0}|\widehat{\text{CURE}}(\hat{\boldsymbol{\theta}}_\omega^{\text{UPAS}}) - \text{CURE}'(\hat{\boldsymbol{\theta}}_\omega^{\text{UPAS}})|\right] \xrightarrow{m\to\infty} 0$$

*Proof.* Denote $D_j(\omega) = \widehat{\text{CURE}}_j - \text{CURE}'_j$ as the difference on the $j$-th problem. We can further decompose

$$D_j(\omega) = \underbrace{2\Delta\omega_j\dot{\sigma}_j^2}_{Q_j(\omega)}\underbrace{-2\Delta\omega_j\dot{\gamma}_j}_{R_j(\omega)}\underbrace{-\Delta\omega_j(2-\hat{\omega}_j-\omega_j^\circ)(\hat{\theta}_j-\bar{Z}_j^f)^2}_{S_j(\omega)},$$

$$\text{with}\quad \Delta\omega_j := \hat{\omega}_j - \omega_j^\circ = \frac{\omega(\mathring{\sigma}_j^2 - \check{\sigma}_j^2)}{(\omega + \check{\sigma}_j^2)(\omega + \mathring{\sigma}_j^2)}.$$

Going term by term, for $Q_j(\omega)$, we have that for all $\omega > 0$

$$|Q_j(\omega)| = \left|2\frac{\omega(\mathring{\sigma}_j^2 - \check{\sigma}_j^2)}{(\omega + \check{\sigma}_j^2)(\omega + \mathring{\sigma}_j^2)}\dot{\sigma}_j^2\right| \leq \frac{2}{\delta}|\mathring{\sigma}_j^2 - \check{\sigma}_j^2||\dot{\sigma}_j^2|,$$

since $\frac{\omega}{\omega + \check{\sigma}_j^2} \leq 1$ and $\omega + \mathring{\sigma}_j^2 \geq \delta \geq 0$ (by our assumption in Proposition C.5). Therefore,

$$\mathbb{E}_{\mathbb{P}_\eta}\left[\sup_{\omega\geq 0}\left|\frac{1}{m}\sum_{j=1}^m Q_j(\omega)\right|\right] \leq \mathbb{E}_{\mathbb{P}_\eta}\left[\frac{1}{m}\sum_{j=1}^m \sup_{\omega\geq 0}|Q_j(\omega)|\right]$$

$$\leq \mathbb{E}_{\mathbb{P}_\eta}\left[\frac{1}{m}\sum_{j=1}^m \frac{2}{\delta}|\mathring{\sigma}_j^2 - \check{\sigma}_j^2||\dot{\sigma}_j^2|\right] \leq \frac{2}{m\delta}\sum_{j=1}^m \sqrt{\mathbb{E}_{\mathbb{P}_\eta}\left[(\mathring{\sigma}_j^2 - \check{\sigma}_j^2)^2\right]\mathbb{E}_{\mathbb{P}_\eta}[(\dot{\sigma}_j^2)^2]}, \quad (47)$$

where the last inequality follows from Cauchy-Schwarz. Handling $R_j(\omega)$ similarly, we have

$$\mathbb{E}_{\mathbb{P}_\eta}\left[\sup_{\omega\geq 0}\left|\frac{1}{m}\sum_{j=1}^m R_j(\omega)\right|\right] \leq \frac{2}{m\delta}\sum_{j=1}^m \sqrt{\mathbb{E}_{\mathbb{P}_\eta}\left[(\mathring{\sigma}_j^2 - \check{\sigma}_j^2)^2\right]\mathbb{E}_{\mathbb{P}_\eta}[\dot{\gamma}_j^2]}. \quad (48)$$

Finally, we have

$$|S_j(\omega)| = \frac{\omega|\mathring{\sigma}_j^2 - \check{\sigma}_j^2|}{(\omega + \check{\sigma}_j^2)(\omega + \mathring{\sigma}_j^2)}|2 - \hat{\omega}_j - \omega_j^\circ|(\hat{\theta}_j - \bar{Z}_j^f)^2.$$

Using $\frac{\omega}{\omega + \check{\sigma}_j^2} \leq 1$, $\frac{1}{\omega + \mathring{\sigma}_j^2} \leq \frac{1}{\delta}$, and $|2 - \hat{\omega}_j - \omega_j^\circ| \leq 2$,

$$\sup_\omega |S_j(\omega)| \leq \frac{2}{\delta}|\mathring{\sigma}_j^2 - \check{\sigma}_j^2|(\hat{\theta}_j - \bar{Z}_j^f)^2,$$

which gives

$$\mathbb{E}_{\mathbb{P}_\eta}\left[\sup_{\omega\geq 0}\left|\frac{1}{m}\sum_{j=1}^m S_j(\omega)\right|\right] \leq \frac{2}{m\delta}\sum_{j=1}^m \sqrt{\mathbb{E}_{\mathbb{P}_\eta}\left[(\mathring{\sigma}_j^2 - \check{\sigma}_j^2)^2\right]\mathbb{E}_{\mathbb{P}_\eta}[(\hat{\theta}_j - \bar{Z}_j^f)^4]}. \quad (49)$$

Now, by the uniform $L^2$ convergence of $\mathring{\sigma}_j^2$ given by Lemma F.6, as well as the fourth-moment assumptions that guarantee $\mathbb{E}_{\mathbb{P}_\eta}[(\dot{\sigma}_j^2)^2] < \infty$, $\mathbb{E}_{\mathbb{P}_\eta}[\dot{\gamma}_j^2] < \infty$ and $\mathbb{E}_{\mathbb{P}_\eta}[(\hat{\theta}_j - \bar{Z}_j^f)^4] < \infty$, we immediately see that (47, 48, 49) are all $o(1)$ terms. Finally

$$\mathbb{E}_{\mathbb{P}_\eta}\left[\sup_{\omega\geq 0}|\widehat{\text{CURE}}(\hat{\boldsymbol{\theta}}_\omega^{\text{UPAS}}) - \text{CURE}'(\hat{\boldsymbol{\theta}}_\omega^{\text{UPAS}})|\right] \leq \mathbb{E}_{\mathbb{P}_\eta}\left[\sup_{\omega\geq 0}\left|\frac{1}{m}\sum_{j=1}^m Q_j(\omega)\right|\right] + \mathbb{E}_{\mathbb{P}_\eta}\left[\sup_{\omega\geq 0}\left|\frac{1}{m}\sum_{j=1}^m R_j(\omega)\right|\right]$$

$$+ \mathbb{E}_{\mathbb{P}_\eta}\left[\sup_{\omega\geq 0}\left|\frac{1}{m}\sum_{j=1}^m S_j(\omega)\right|\right] \xrightarrow{m\to\infty} 0.$$

$\square$

Finally, combining the results from Theorem F.4, Lemma F.5 and Lemma F.7 via triangle inequality, we obtain the final result through

$$
\begin{aligned}
\mathbb{E}_{\mathbb{P}_\eta}\left[\sup_{\omega\geq 0}|\widehat{\mathrm{CURE}}(\hat{\boldsymbol{\theta}}_\omega^{\mathrm{UPAS}}) - l_m(\hat{\boldsymbol{\theta}}_\omega^{\mathrm{UPAS}}, \boldsymbol{\theta})|\right] &\leq \mathbb{E}_{\mathbb{P}_\eta}\left[\sup_{\omega\geq 0}|\mathrm{CURE}''(\hat{\boldsymbol{\theta}}_\omega^{\mathrm{UPAS}}) - l_m(\hat{\boldsymbol{\theta}}_\omega^{\mathrm{UPAS}}, \boldsymbol{\theta})|\right] \\
&\quad + \mathbb{E}_{\mathbb{P}_\eta}\left[\sup_{\omega\geq 0}|\mathrm{CURE}'(\hat{\boldsymbol{\theta}}_\omega^{\mathrm{UPAS}}) - \mathrm{CURE}''(\hat{\boldsymbol{\theta}}_\omega^{\mathrm{UPAS}})|\right] \\
&\quad + \mathbb{E}_{\mathbb{P}_\eta}\left[\sup_{\omega\geq 0}|\widehat{\mathrm{CURE}}(\hat{\boldsymbol{\theta}}_\omega^{\mathrm{UPAS}}) - \mathrm{CURE}'(\hat{\boldsymbol{\theta}}_\omega^{\mathrm{UPAS}})|\right] \\
&\xrightarrow{m\to\infty} 0.
\end{aligned}
$$

$\square$

