# OpenReview forum: "Prediction-Powered Adaptive Shrinkage Estimation"
_ICML.cc/2025/Conference — ICML 2025 poster_

### Official Review · Reviewer_RfuV · 2025-03-08

**Overall Recommendation:** 4

**Summary:**

This paper proposes a "shrinkage estimator" for estimating multiple problems at once using PPI (prediction powered inference). They discuss the various methods by which one can reduce variance in such an estimator and how their method takes advantage of each, demonstrating theoretically that they can get better estimators by optimizing the CURE (correlation based risk estimate) which they introduce. Empirically they show on synthetic and real datasets that this improves on multi-estimation over both classical and PPI baselines.

#########
UPDATE AFTER REBUTTAL:
I remain positive about this paper - keeping my score at 4.

**Claims And Evidence:**

I find the claims + evidence shown in this paper to be convincing. The authors provide both theoretical and empirical evidence for their claim, which is that PAS is an improved estimator over PPI in the multi-prediction setup. I find the background and intuition given in sections 2-3 to be quite helpful.

**Essential References Not Discussed:**

n/a

**Experimental Designs Or Analyses:**

fairly standard setup and showing improved risk on two metrics

**Methods And Evaluation Criteria:**

Yes, the evaluation setup is fairly standard for PPI and adapted to their specific setting (multi-prediction).

It would be nice to see experiments shown on different sizes of labelled data to demonstrate at what value of n we start seeing improvements from PAS.

**Other Comments Or Suggestions:**

L204: “cl” in classical is underlined?

L170 right: typo on “unlabeled”

Relatedly to the "sharing information" point, I would be curious to know if there's any type of covariate shift type assumption that you think is helping here- for instance, it is roughly (not exactly) satisfied in the synthetic example Ex 2.2.

**Other Strengths And Weaknesses:**

Main critique: I would love a little more intuition about what “parallel problems” are supposed to be, in particular early on in exposition and then tied to the examples in 2.2 and methods in 4.2. I was a little unclear on how "borrowing information across problems" is done (as stated on line 42). I can see there is shrinkage going on but that is calculated on what seems to be a mostly problem-by-problem level as in (15). So a little more explanation here would help my understanding of what's going on - for instance I don't know see why we need the machinery from Eq 5 of defining the meta-distribution over problem parameters, it doesn't seem there's any assumption made on how problems are connected or distributed.

Quick note on the assumption of correlation/covariance statistics being known - Var(Y) and corr(Y, f) can be tough to estimate is labelled data is small. Would be good to discuss how this is implicitly an assumption that labelled data is big enough, and/or what regime of n you expect this estimator to work well in.

**Questions For Authors:**

n/a

**Relation To Broader Scientific Literature:**

connection to existing literature is well laid out in this paper - discusses very clearly in Section 3 exactly how various methods from the PPI and Bayesian space connect to this work

**Theoretical Claims:**

I did not check closely but I did look at the proofs in Appendix A.

---

> ### Author Rebuttal · Authors · 2025-03-31
>
> We thank the reviewer for considering our idea convincing and thoughtful comments.
>
> * **Sharing information across problems:** We agree with the reviewer that further motivation of why and how it is possible to share information across problems will improve our exposition. In our revision we will add further motivation. Below is an alternative attempt to further explain and elaborate on Prop 5.3. Suppose as in that proposition that $n_j$ is the same across all problems, $N_j=\infty$, and that second moments $\rho_j,\tau_j,\sigma_j$ are identical across all problems. We make these assumptions throughout the remainder of our response. Then we could ask: What is the best convex combination of $\hat{\theta}_j^{PT}$ and $\tilde{Z}_j^f$ in the following sense:
>
>     $$\omega_j^*\in\mathrm{argmin}_{\omega_j}\mathbb E[(\theta_j-(\omega_j\hat{\theta}_j^{PT}+(1-\omega_j)\tilde{Z}_j^f)^2].$$
>
>     By direct calculation (since the RHS is a convex quadratic in $\omega_j$) we find that:
>     $$\omega_j^*=\frac{\mathbb E[(\theta_j-\tilde{Z}_j^f)^2]}{\mathbb E[(\theta_j-\tilde Z_j^f)^2]+\tilde\sigma^2}.$$
>     This implies the following intuitive result: the larger the MSE $\mathbb E[(\theta_j-\tilde{Z}_j^f)^2]$, the less weight we should assign to $\tilde{Z}_j^f$. Note that if we have a single problem $m=1$, then if $n_j$ is sufficiently large, we could estimate $\tilde{\sigma}^2$ accurately. However, we cannot estimate $\mathbb E[(\theta_1-\tilde{Z}_1^f)^2]$ accurately since we only have a single $\theta_1$ (a single problem). At best, we can compute an unbiased estimate of this quantity since:
>     $\mathbb E[(\hat{\theta}_1^{PT}-\tilde{Z}_1^f)^2]-\tilde{\sigma}^2=\mathbb E[(\theta_1-\tilde{Z}_1^f)^2].$
>
>     Now suppose we have multiple problems, then we can now learn how good the ML predictor is for estimating the $\theta_j$ by sharing information across problems. To wit, as $m \to \infty$
>
>     $$\frac{1}{m}\sum_{j=1}^m(\hat{\theta}_j^{PT}-\tilde{Z}_j^f)^2-\tilde\sigma^2\stackrel{\mathbb P}{\to}\mathbb E[(\theta_j-\tilde Z_j^f)^2],$$
>
>     where we emphasize that the mean squared on the RHS also integrates with respect to the **meta-distribution** $\mathbb P_{\eta}$ that models the distribution of the $\theta_j$. Thus we can also estimate the optimal $\omega_j^*$. In words: **sharing information across problems allows us to learn how good the ML predictor is** and to then decide how much to shrink toward it. Our implementation shares information in a similar way but is more involved due to the heteroscedasticity across problems (not all second moments are identical) and also for heterogeneity in $n_j$, $N_j$.
>
> * **Role of meta-distribution:** See above. We note that we could state our result in a frequentist setting wherein all parameters are deterministic, similar to the classical result of [James-Stein](https://en.wikipedia.org/wiki/James%E2%80%93Stein_estimator). We preferred to state results in terms of a meta-distribution, since we thought this would be a more familiar setup for audience at ICML.
>
> *   **Assumption on known second moments:**
>     We agree that our assumption on second-moment parameters $(\sigma^2_j, \gamma_j)$ are not easily satisfied in practice. In Sec 2 of paper, such treatment is more of a theoretical convenience, and sample-based estimates are used for real-world datasets. Therefore, it is true that an implicit requirement for our theory of PAS to work well in many practical settings is that the sample-based estimates are good enough. While our regime ($m\to\infty$, $n_j,N_j$ finite) cannot yield asymptotic result on these sample-based estimates, we make the following remarks:
>
>     1. In practice, PAS works well even when $n_j$ is very small (and so the estimates are noisy). To highlight this point further and motivated by your suggestion, we reran our real data examples with different labeled/unlabeled splits, going from 1%-40%. PAS still dominates other baselines even when only 1% of the data is labeled (e.g. $n_j < 10$ for Amazon Review). [[Link to the plots]](https://doi.org/10.6084/m9.figshare.28694657.v1)
>
>     2. In our response to Reviewer Tcxk, we propose a new variant of PAS called UniPAS, whose asymptotic guarantee does not require knowledge of second moments. UniPAS also has competitive empirical performance with PAS.
>
> *   **Other comments:**
>     *   L204: The underline is intentional to denote the abbreviation "cl" for classical estimator. We can clarify this.
>     *   L170: Thanks for catching the typo; we will fix it.
>     *   Covariate Shift: We interpret this as potential differences in $P_{\eta_j}(X_{ij})$ across problems $j$. In the synthetic model, the mean of $X_{ij}$ varies with $\eta_j$, but that is not an assumption that PAS makes. What makes "information sharing" more effective, as mentioned above, is when $(\theta_j-\tilde{Z}_j^f)^2$ is small on average across problems. If the question refers to train/test covariate shift within problems, this violates the PPI assumption.

---

> > ### Comment · Reviewer_RfuV · 2025-04-02
> >
> > Thanks for the rebuttal. I find these clarifications helpful - the connection of E[theta - Z] to information shrinking as well as to the distributional assumptions piece are both useful for me. The experiments on small n and with UniPAS are also nice! I'm already positive about this paper and continue to be so.

---

### Official Review · Reviewer_qLFS · 2025-03-10

**Overall Recommendation:** 3

**Summary:**

This paper proposes a method for adaptively combining ML predictions with gold-standard labels, to estimate a multivariate parameter (e.g., the mean across several partitions of the data) with small mean-squared error.  The paper builds upon the PPI++ estimator, while proposing to additionally perform global shrinkage using an adaptively chosen shrinkage parameter.  Experimental results show that the approach approves over the classical estimator in MSE more often than a series of baseline methods.

**Update after rebuttal**: I have reviewed the response and will maintain my score.

**Claims And Evidence:**

The claimed contributions in the introduction are (in my opinion) well-scoped and well-supported, clearly distinguishing asymptotic claims from finite-sample claims, and justifying the benefits of the method with empirical evidence.

**Essential References Not Discussed:**

I believe much of the relevant related work is cited, but I would appreciate more discussion of how the proposed approach relates to some of the (cited) alternatives, e.g., Fisch et al. 2024 (who similarly considered stratified PPI++ across different subpopulations) and Rosenman et al. 2023 (who similarly consider shrinkage towards a biased predictor in a related causal inference problem).  For instance, does the "Shrinkage" baseline correspond to the method of Rosenman et al. 2023 or some variant of that method?

**Experimental Designs Or Analyses:**

As stated in "methods and evaluation criteria", the evaluation appears sound to me, mainly focusing on the real data analysis which I believe presents the most robust empirical evidence for the method.  Given that the focus of this paper is on estimation (and not inference / uncertainty quantification), it is reasonable to measure performance by MSE, and I appreciated the extra inclusion of the "% improved" metric, since (as the authors note) MSE across an entire vector can be improved while sacrificing performance on some dimensions.

**Methods And Evaluation Criteria:**

The method certainly makes sense, and has a clear intuitive basis.  Regarding the evaluation criteria, the theoretical development is mainly (in my view) for intuition, given some limitations (e.g., knowledge of certain parameters, see "other strengths and weaknesses").  So I view the empirical evaluation as the main "evaluation" component, where the evaluation approach appears sound to me---I particularly appreciated the use of the "percentage of problems improved" metric as a thoughtful counter to typical concerns with shrinkage-type estimators.

**Other Comments Or Suggestions:**

As a minor note, I found the use of distinct notation in Section 3 and Assumption 2.1 to be confusing, and it was a little hard for me to see the connections between the two.

**Other Strengths And Weaknesses:**

The paper is very clearly written, the synthetic experiments are a nice intuition pump, and the real-data experiments were compelling in my view.

However, a few points of clarifications / "weaknesses":
1. How is the PPI++ baseline implemented in the experiments?  As currently presented, I imagine that PPI++ is implemented (separately?) for each dimension of the target parameter to estimate.  However, PPI++ has more general formulations than mean estimation, e.g., one could view the present problem as a more general M-estimation problem, or even the task of learning a generalized linear model with a fixed coefficient for each stratum, both of which are covered in Sections 4 and 3 respectively of [1].  I'm not sure that actually results in a different estimator from the formulation presented here, but it would be helpful to clarify.
2. As noted in Section 2.2., all of the theoretical work starts from the assumption that $\tau_j, \sigma_j, \rho_j$ are known.  However, a major challenge in this area of research is that these must be estimated from the (same) limited data that we have for estimating $E[Y_{ij}]$.  Typically this deficit is handled somewhat crudely by appealing to asymptotics (i.e., convergence of similar terms in probability to their true values), but I don't see any of that here.  Of course, the ultimate validation is empirical, but I'm curious how the authors would address this weakness in the theoretical development:  For instance, do Proposition 5.1 and Theorem 5.2 still hold as both $m$ (number of tasks) and $n$ (number of samples) tend to infinity?
3. I didn't follow the analog between the main method derived in Section 4 and the exposition in Section 3.  For instance, it seems in Section 3 that $\phi$ and $\theta$ are known to be correlated with the same mean, and in any case we should expect that $E(\theta \mid \psi)$ is a better predictor of $\theta$ than $E(\theta)$.  Moving from Equation (12) to Equation (16), we use $\tilde{Z}_j^f$ in place of $E[\theta \mid \psi]$ by "analogy", but I don't understand the conceptual connection, given that e.g., $\tilde{Z}_j^f$ could be arbitrarily biased, while $E(\theta \mid \psi)$ is not.

[1] A. N. Angelopoulos, J. C. Duchi, and T. Zrnic. Ppi++: efficient prediction-powered inference. arXiv preprint (2311.01453v2), 11 2023. URL: http://arxiv.org/abs/2311.01453v2, arXiv:2311.01453v2.

**Questions For Authors:**

My questions are essentially derivative from my comments above, but to recap (in priority order):
1. How is the PPI++ baseline implemented in the experiments, and in particular, how does the "multivariate" form of PPI++ compare to the PT baseline shown here?  Is the multivariate form of PPI++ essentially the same, but with only a single fixed parameter lambda instead of a per-coordinate parameter?
2. Where do existing shrinkage methods fit into the baselines shown in the experiments?  For instance, does the "Shrinkage" baseline correspond to the method of Rosenman et al. 2023 or some variant of that method?
3. As noted in Section 2.2., all of the theoretical work starts from the assumption that $\tau_j, \sigma_j, \rho_j$ are known. For instance, do Proposition 5.1 and Theorem 5.2 still hold as both $m$ (number of tasks) and $n$ (number of samples) tend to infinity, i.e., is there at least an asymptotic argument that this does not matter?
4. Could you explain the conceptual reason why the analog in Equation (16) of $\tilde{Z}_j^f$ to $E(\theta \mid \psi)$ makes sense?  I think I get the intuition at a very rough level (it's additional "prior" information), but see my comments above in "other strengths and weaknesses"

**Relation To Broader Scientific Literature:**

As noted in part by the authors, there is a long line of work on combining imputed labels (which may be biased) with gold-standard labels. PPI++ is one such idea, where PPI constructs an unbiased estimator using imputed and gold-standard labels, while "power tuning" is added in PPI++ to improve efficiency by estimating the optimal degree of reliance on imputed labels.  Shrinkage is another idea, which can be shown to provably improve estimation (in a particular sense, namely MSE across the entire vector of parameters) when there are 3 or more parameters to estimate, an idea going back to the James-Stein estimator.  This shrinkage idea has also been applied in (other) areas of combining "potentially biased data" with "unbiased data", such as in causal inference, where the "biased" data is observational and the "unbiased" data experimental (see Rosenman et al. 2023, cited among other shrinkage-style estimators in this work).

In my understanding, this paper brings together these two lines of work with perhaps an additional twist (e.g., adopting a particular Empirical Bayes perspective on shrinkage, which I am less familiar with), and demonstrates that they work well together empirically, while giving some theoretical intuition.

**Theoretical Claims:**

Unfortunately, I did not check the proofs in the supplement due to a lack of time.

---

> ### Author Rebuttal · Authors · 2025-03-31
>
> We thank the reviewer for a very helpful report, which has helped us improve on this work.
>
> * We agree that we should expand on connections to related work, which will be added as a new section in the appendix after our revision. Briefly:
>     - Fisch et al. 2024 (StratPPI): The starting point is similar to ours with multiple parameters $\theta_1,...,\theta_m$. However, in StratPPI, these parameters are not of interest per-say. Instead there exists known weights $w_1,...,w_m$ such that the parameter of true interest is $\theta := \sum_{j} w_j \theta_j$. The ultimate goal is to come up with an unbiased and low variance estimator of $\theta$.
>     - Rosenman et al (2023): When their stratum weights are the same, their estimator corresponds to our shrinkage baseline up to differences in the one-dimensional family of shrinkage weights (our Eq. (15)).
>     - PPI++: Suppose $n_j=n, N_j=N$ for all problems, then indeed we can cast our setting into PPI++ with a parameter vector. Moreover, PT is then using the same $\lambda$ for all problems (more of which below).
> * Previously, we were comparing to PPI++ applied separately to each problem (with its own $\lambda_j$). PPI++ itself requires asymptotics with $n_j,N_j\to\infty$. It would be possible to extend to asymptotics with $m,n\to\infty$; however, based on the reviewer comments we now have a more compelling methodological alternative: if we only seek to compare ourselves to PPI++ with a single $\lambda$, then we can learn the optimal $\lambda$ in asymptotics with $n$ fixed, $m\to\infty$ (a result complementary to the PPI++ paper which takes $m$ fixed and $n\to\infty$). In closed form, we have
>     $$\lambda^*:=\frac{\sum_{j=1}^mn_j^{-1}\gamma_j}{\sum_{j=1}^m\frac{n_j+N_j}{n_jN_j}\tau_j^2}$$
>     which coincides with the optimal weight selected by PPI++. We further clip $\lambda^*$ to be within $[0, 1]$.
>
>     Without knowing $\tau_j, \sigma_j, \gamma_j$, we can still plug in their sample-based estimates (see Point 2 for Reviewer Tcxk) and obtain $\hat\lambda$ (after clipping it as well), which has the property that $\hat\lambda - \lambda^* \overset{L^2}{\to} 0$ as $m\to\infty$ (this is different from the per-problem case when $n_j \to \infty$ is needed). We denote this method as UniPT.
> * Taking UniPT as a starting point, we develop UniPAS. This has an asymptotic guarantee ($n_j, N_j$ fixed, $m\to\infty$) analogous to PAS replacing PT with UniPT (that is, we try to be asymptotically at least as good as PPI++ with a single $\lambda$, while PAS was trying to at least as good as PPI++ with a separate-per problem $\lambda$). The upshot is that UniPAS **does not** require knowledge  of $\tau_j, \sigma_j, \gamma_j$. In empirical results UniPAS is competitive with PAS (although PAS is slightly better).
> * In addition to including two new methods (UniPT, UniPAS), we have reran our real data examples with labeled/unlabeled split ratios ranging from 1% to 40%. [[Link to plots]](https://doi.org/10.6084/m9.figshare.28694657.v1)
> * On the analogy of $Z_j^f$ and $\mathbb E[\theta\mid\psi]$: We agree that the analogy is somewhat loose and we will provide more details in the revision. Our goal is to provide a heuristic motivation for the one-dimensional parameterized family of weights $\omega_j(\cdot)$ (whose ultimate success is judged by the empirical results).  To elaborate on this: in Eq. (12) of Sec 3, we find that the best weights take the form:
>     $$\frac{\omega}{\omega+\sigma_{\varepsilon}^2},$$
>     for $\omega = \mathbb E[(\theta - \mathbb E[\theta \mid \phi])^2] = \mathbb E[\mathrm{Var}[\theta \mid \phi]]$. If we instead take the best convex combination of $\hat{\theta}^{cl}-\xi$ and $\mathbb E[\theta]$ (instead of $\mathbb E[\theta\mid\phi]$), the optimal weights again take the form above, now with $\omega=\mathbb E[(\theta-\mathbb E[\theta])^2] = \mathrm{Var}(\theta)$. Now suppose that we ask for the best convex combination (not necessarily a Bayes predictor) between $\hat{\theta}^{cl}-\xi$ and $h(\phi)$ where $h(\cdot)$ is some fixed function. Then we can show that the optimal convex combination is given by the form above with
>     $$
>     \omega:=\mathbb E[(\theta-h(\phi))^2] =  \mathbb E[\mathrm{Var}[\theta\mid\phi]] + \mathbb E[(h(\phi)-\mathbb E[\theta\mid\phi])^2].
>     $$
>     (The above expression is interesting as it forces us to inflate "$\omega$", i.e., to shrink less toward $h(\phi)$ in a way that depends on how close $h(\phi)$ is to $\mathbb E[\theta\mid\phi]$.) The takeaway is that for many possible predictors, the optimal weights have the same parameterized form up to the single parameter $\omega$ that varies according to the quality of the predictor. This motivates our one-dimensional family of weights. Once this family has been motivated, we learn $\omega$ in (15) in a way that does not depend on the above analogy at all by minimizing CURE.

---

> > ### Comment · Reviewer_qLFS · 2025-04-07
> >
> > I appreciate the clarifications and new comparisons (which I believe should go into the main paper), and I'm glad my review was helpful for improving upon the work.  I will retain my score, but I'm still positive on the paper (somewhere between a 3 and 4)

---

### Official Review · Reviewer_JtAw · 2025-03-12

**Overall Recommendation:** 4

**Summary:**

The paper proposes the Prediction-Powered Adaptive Shrinkage (PAS) method to enhance estimation accuracy for multiple means. PAS integrates Prediction-Powered Inference (PPI) with empirical Bayes shrinkage, first debiasing noisy machine learning (ML) predictions within each task and then leveraging information across tasks for improved estimation. Theoretically, the authors establish the asymptotic optimality of the tuning strategy and prove that PAS achieves a lower asymptotic mean squared error (MSE) than any baseline estimator. Experimental results on synthetic and real-world datasets demonstrate that PAS consistently outperforms existing methods.

**Claims And Evidence:**

Please see my comments for each category below.

**Essential References Not Discussed:**

NA

**Experimental Designs Or Analyses:**

The numerical experiments demonstrate the superior performance of the proposed method compared to the baselines. However, given that PAS's performance critically depends on the quality of the predictor, it would be interesting to analyze its sensitivity to heavily biased predictors—for example, by intentionally using a biased predictor to observe its impact. Additionally, since PAS involves power tuning and adaptive shrinkage, the computational cost is expected to be high. A detailed comparison of computational efficiency, such as running time benchmarks, would be helpful to assess the trade-off between accuracy improvement and computational expense.

**Methods And Evaluation Criteria:**

The proposed approach leverages Prediction-Powered Inference (PPI) and empirical Bayes methods to enhance statistical estimation, which I find to be a novel contribution. The theoretical guarantee of asymptotic optimality and the strong performance in numerical experiments further support its effectiveness.

However, the proposed approach's performance is expected to depend heavily on the quality of the predictor used. If I understand the mechanism correctly, PAS does not have a built-in mechanism to correct or exclude misleading predictors, potentially impacting its effectiveness. Additionally, the computational complexity may be high due to the power tuning and adaptive shrinkage steps, which require optimization and could introduce additional overhead.

**Other Comments Or Suggestions:**

NA

**Other Strengths And Weaknesses:**

NA

**Questions For Authors:**

How does the violation of the assumption of boundness of finite moment affect the optimality?

How does PAS perform when the ML model is biased, thereby the estimation is incorrect? If there is a way to screen and exclude those poor estimations?

If one of the estimations is nearly optimal (close to the true value), is the shrinkage method over-shrink it?

**Relation To Broader Scientific Literature:**

This paper could contribute to estimation, particularly for parallel ML tasks, which look for accurate estimation.

**Theoretical Claims:**

Although the asymptotic optimality of PAS provides a strong theoretical guarantee, one of my main concerns is the assumption of finite moments. The authors state that “PAS inherits the flexibility of PPI in working with any black-box predictive model and makes minimal distributional assumptions about the data.” However, the boundedness of moments may not always hold.

---

> ### Author Rebuttal · Authors · 2025-03-31
>
> We thank the reviewer for acknowledging the novelty and contribution in our paper, as well as the many constructive comments. We hope our response below addresses all the concerns directly.
>
> ### **Background on PPI**
>
> The core idea of Prediction-Powered Inference (PPI) is as follows. Given existing black-box ML predictors $f$, how can we use them to enhance classical statistical procedures and improve their efficiency? The focus in this literature is on employing safeguards so that if the ML predictor is good, then statistical gains can be large, while if the ML predictor is bad, then one still retains some form of statistical guarantees (such as consistency). The focus on this literature is not on how to train the best possible ML models (most results are agnostic to that), but rather how to wrap around an existing arbitrary $f$ to improve statistical procedures. In this paper, we follow this established perspective in the PPI literature.
>
>
> Here, *arbitrary* typically means that $f$ can be of any functional form or a total black-box (e.g., API calls to LLMs), and should thus be considered given and fixed. While adapting to the quality of $f$ is important, an “arbitrarily bad” $f$ is rarely considered in practice, since prediction-powered methods are generally deployed only when $f$ is expected to provide at least some signal about the response.
> Nevertheless, PAS **does** include mechanisms to adapt to both poor and high-quality predictors. To demonstrate this, we revisit the synthetic model and consider the following predictors:
> 1. a (newly added) very poor predictor $f_r$, which outputs a $Unif[0, 2]$ value for all inputs,
> 2. the near-optimal predictor $f_1(x) = x^2$ from the paper (in fact, $f_1$ closely approximates $E[Y_{ij} | X_{ij} = x]$).
>
> **Table 1: MSE (± se) $\times 10^{-3}$ in synthetic model ($K=200$ replicates)**
>
> ||Classical|Prediction Avg|PPI|PAS|
> |-|-|-|-|-|
> |$f_r$|3.14 ± 0.03|549.17 ± 2.57|24.05 ± 0.22|3.14 ± 0.03|
> |$f_1$|3.14 ± 0.03|0.27 ± 0.00|2.69 ± 0.03|0.27 ± 0.00|
>
> In the extreme case where $f_r$ is very biased and provides no useful information, PAS  effectively defaults to the classical estimator, showing robustness against poor predictions. On the other hand, with a very good $f_1$, PAS tracks the predictive mean closely and perform best. Importantly, no over-shrinkage is observed. In words, PAS learns how good the predictor is in a data-driven way.
>
> ### **Bounded moment assumptions**
> We appreciate the reviewer's question regarding the finite moment assumptions. We first remark that these assumptions are satisfied when the response and prediction are bounded (as in the Amazon and Galaxy datasets), or when their joint distribution is reasonably well-behaved (as in our synthetic setting).
>
> We note that *some* moment assumptions are standard across the PPI literature. Specifically, finite **second moments** of the joint model $(Y_{ij}, f(X_{ij}))$ are generally needed for basic procedures like power tuning. These are typically viewed as mild assumptions, although we admit that they preclude heavy-tailed distributions.
>
> Our work extends PPI to compound mean estimation using shrinkage principles, particularly risk minimization via an unbiased estimate (CURE). To prove the asymptotic optimality of PAS, we require finite fourth moments. This is a technical condition used to control the variance of the risk estimate itself.
>
> Finally, addressing the reviewer's question about violations: if the fundamental second moment assumption fails, the variance-reduction premise of PPI itself becomes ill-defined. If only the fourth moment assumption is violated, PAS may lose its formal asymptotic guarantee to outperform power tuning and the predictive mean. However, CURE remains an unbiased risk estimate provided that second moments exist.
>
> ### **A light-weight approach**
>
> A key strength of PAS is its low computational overhead compared to both classical and PPI estimators. Although PAS uses both power tuning and adaptive shrinkage, the first stage yields a closed-form expression for the optimal tuning parameter $\lambda_j^*$; the second stage involves optimizing CURE over a one-dimensional parameter space, and each evaluation is inexpensive since CURE has an analytic form as well. This makes the optimization very fast in practice.
>
> We first precompute all ML predictions. After this step, all estimators (including PAS) are very fast to compute. Below we benchmark the runtimes for three estimators on the Amazon Review dataset with the precomputed ML predictions. In each trial, the task is to estimate the mean product ratings for $m = 200$ products. We then record the time to construct each estimator by taking the average of 100 repeated constructions. The table reports the mean and max time (in milliseconds) to construct each estimator across 10 such trials.
>
> |Estimator|Mean Time (ms)|Max Time (ms)|
> |-|-|-|
> |Classical|7.4|8.4|
> |PT|21.2|22.0|
> |PAS|34.5|35.4|

---

### Official Review · Reviewer_Tcxk · 2025-03-16

**Overall Recommendation:** 4

**Summary:**

Paper proposes prediction-powered adaptive shrinkage (PAS), an extension of prediction-powered inference (PPI) that uses empirical Bayes ideas to further reduce estimates when multiple related estimation problems are solved together. The paper is well written and well thought out, with good theoretical and empirical results.

**Claims And Evidence:**

The claims are well supported by both theory and empirical results. The paper is well written and explains the ideas clearly with good intuition building examples.

**Essential References Not Discussed:**

None.

**Experimental Designs Or Analyses:**

The experimental design looks sensible, with clear separation of datasets used to estimate the different quantities needed (e.g. the Bert fine-tuning). It is nice to see the method working across both textual and image domains with different predictors etc.

**Methods And Evaluation Criteria:**

Yes. Evaluation on both synthetic and real world examples demonstrate the efficacy of the method.

**Other Comments Or Suggestions:**

None.

**Other Strengths And Weaknesses:**

None.

**Questions For Authors:**

1. What assumptions or methods might be required to strengthen the asymptotic results to a finite m one?
2. It would be good to more explicitly work out what happens if the variance parameters tau_j, sigma_j, rho_j are unknown and need to be estimated, and the impact of this additional estimation step on the resulting method and theoretical results.
3. Please elaborate on how these variance parameters are estimated from data in the empirical results (particularly the Amazon and Galaxy Zoo datasets).
4. It would be nice to look at performance of the method and baselines for smaller n_j/N_j, and also for a wider range of the label/unlabeled split beyond 20-80. The method should be able to work with a wider range.

**Relation To Broader Scientific Literature:**

The relation to broader literature is clearly set out through the paper.

**Theoretical Claims:**

Theoretical claims are asymptotic in nature, holding as the number of problems m->infty.

They also assume the variance parameters tau_j, sigma_j, rho_j are known. While paper refers to this as of secondary concern, and referred to existing papers that took the same approach, this would seem to me to be quite a significant assumption and would be good to work out implications of having to estimate these from data on the theoretical developments.

---

> ### Author Rebuttal · Authors · 2025-03-31
>
> We thank the reviewer for the careful assessment. The thoughtful questions have helped us improve our work. For better exposition, we slightly reordered our responses to the questions.
>
> 1. **Finite $m$ results.** Our current theoretical analysis permits for finite sample bounds. We will track these results more explicitly in the revision. For instance, the $o(1)$ term in Prop 5.1 is actually $O(1/\sqrt{m})$.
>
> 2. **How are variance parameters estimated in practice?** For real-world datasets, we use sample-based estimates for the variance parameters. We estimate $\sigma^2_j$ and $\gamma_j$ with the standard unbiased estimators
>
>     $$
>     \hat{\sigma}^2_j = \frac{1}{n_j-1} \sum_{i=1}^{n_j} (Y_{ij} - \bar Y_j)^2, \quad \hat\gamma_j = \frac{1}{n_j-1}\sum_{i=1}^{n_j}(Y_{ij}-\bar Y_j)(f(X_{ij}) -\bar{Z}_j^f)
>     $$
>
>     using labeled data. For the prediction variance $\tau^2_j$, we use predictions in both labeled/unlabeled data:
>     $$
>     \hat\tau_j^2 = \frac{1}{n_j + N_j -1}\bigg(\sum_{i=1}^{n_j}(f(X_{ij}) - \hat Z_j^f)^2 + \sum_{i=1}^{N_j}(f(\tilde X_{ij}) - \hat{Z}_j^f)^2 \bigg)
>     $$
>
>     where $\hat Z_j^f = \frac{1}{n_j + N_j -1}\left(\sum_{i=1}^{n_j}f(X_{ij}) + \sum_{i=1}^{N_j}f(\tilde X_{ij})\right)$.
>
> 3. **Impact of estimating variance parameters (with Point 2).**
>     We agree with the reviewer that our current theory does not account for estimation errors due to plugging in sample-based estimates of the variance parameters. One possible remedy would be to consider asymptotics in which all of $n_j, N_j, m \to \infty$. However, we prefer asymptotics with $n_j, N_j$ fixed and $m \to \infty$ to represent the regime in which we have a lot of very noisy/difficult individual problems and also to distinguish results from standard setup in the PPI literature that keeps $m$ fixed and takes $n_j, N_j \to \infty$. We note PAS with plug-in variance estimates empirically works well even for very small $n_j$, see point 4 below.
>
>     Motivated by the reviewer's comment, we now consider two further methods: UniPT (that is, power-turning with the same $\lambda$ for all problems, see response to reviewer qLFS) and UniPAS, which is similar to PAS but uses the same power-tuning parameter across problems. We can prove that UniPAS has asymptotic ($m \to \infty$, $n_j,N_j$ fixed) risk less or equal to that of UniPT, PPI, the classical estimator, and the prediction mean. **The result for UniPAS accounts for the data-based estimation of the variance parameters**. Here is a sketch of UniPAS:
>
>     - We start with UniPT by estimating a single power-tuning parameter $\hat{\lambda}$ (as in our response to reviewer qLFS) that has the property that $\hat{\lambda} - \lambda^* \to 0$ in $L^2$ as $m \to \infty$ (with $n_j,N_j$ fixed) where $\lambda^*$ is the optimal single tuning parameter. Call the resulting estimator $\hat{\theta}_j^{UniPT}$.
>
>     - We come up with stable **working** estimates of $Var[\hat{\theta}_j^{UniPT}]$ by **pretending** $\sigma_j^2, \tau_j^2, \gamma_j$ are the same across all $j$ (but sample sizes can vary). The common values can be estimated accurately by averaging the estimates in Point 2 over all $m$. Plugging these in, we derive working estimates $\bar{\sigma}_j^2$ of $Var[\hat{\theta}_j^{UniPT}]$. Then we consider the one-dimensional parameterized family of weights
>         $$ \bar{\omega}_j(\omega) = \frac{\omega}{\omega + \bar{\sigma}_j^2}.$$
>         This family retains the property that for $\omega=0$ it recovers prediction mean and for $\omega=\infty$ it recovers UniPT.
>
>     - Pretend momentarily that $\bar{\sigma}_j$ and $\hat{\lambda}$ are deterministic. (This is not actually needed; by steps 1 and 2 above asymptotically these converge to a deterministic limit.) Suppose we consider the family of estimators
>
>         $$
>         \bar{\omega}_j(\omega)\hat{\theta}_j^{UniPT} + (1-\bar{\omega}_j(\omega))\tilde Z_j^f.
>         $$
>
>         An unbiased estimate of risk is given by CURE,
>         $$\frac{1}{m}\sum_{j=1}^m (2\bar{\omega}_j(\omega) - 1)Var[\hat{\theta}_j^{UniPT}] + 2(1 - \bar{\omega}_j(\omega)) Cov[\hat{\theta}_j^{UniPT}, \tilde Z_j^f] +(1 - \bar{\omega}_j(\omega))^2\big(\hat{\theta}_j^{UniPT} - \tilde Z_j^f\big)^2,$$
>         which is analogous to the expression of CURE for PAS. Now here comes the punchline. If above we replace $Var[\hat{\theta}_j^{UniPT}]$ and $Cov[\hat{\theta}_j^{UniPT}, \tilde Z_j^f]$ by unbiased estimates (analogous to Point 2), then we still retain an unbiased estimate of risk. Moreover, since we are averaging over $m$ with $m \to \infty$, we can establish asymptotic uniform consistency of our objective to the true risk, and thus we can establish asymptotic optimality for UniPAS.
>
> 4. **Different $n_j/N_j$ and split ratios.**  Following the reviewer's suggestion we reran our real data analyses with the new methods and different labeled/unlabeled splits, going from 1%-40%. [[Link to the plots]](https://doi.org/10.6084/m9.figshare.28694657.v1)

---

### Decision · Program_Chairs · 2025-05-01

**Decision:**

Accept (poster)

**Comment:**

The paper presents an extension of a recent statistical approach called Prediction-Powered Inference.
The extension enables to non-trivially exploit this approach in the context of several parallel and possibly related tasks.

All reviewers were very positive regarding the contribution, novelty and importance of the problem.
So I am happy to accept this paper.

* The reviewers have made several useful suggestions for improving the presentation and providing intuition, please take these into consideration